# RNF144B negatively regulates antiviral immunity by targeting MDA5 for autophagic degradation

Guoxiu Li[1,2,3,4], Jing Zhang [ID][1,3,4✉], Zhixun Zhao[1,3], Jian Wang[1,3], Jiaoyang Li[1,3], Weihong Xu[1,3], Zhanding Cui[1,3], Pu Sun[1,3], Hong Yuan[1,3], Tao Wang[1,3], Kun Li[1,3], Xingwen Bai[1,3], Xueqing Ma[1,3], Pinghua Li[1,3], Yuanfang Fu[1,3], Yimei Cao [ID][1,3], Huifang Bao[1,3], Dong Li[1,3], Zaixin Liu[1,3], Ning Zhu[1,3], Lijie Tang [ID][2✉] & Zengjun Lu [ID][1,3✉]

## Abstract

**As a RIG-I-like receptor, MDA5 plays a critical role in antiviral innate immunity by acting as a cytoplasmic double-stranded RNA sensor capable of initiating type I interferon pathways. Here, we show that RNF144B specifically interacts with MDA5 and promotes K27/K33-linked polyubiquitination of MDA5 at lysine 23 and lysine 43, which promotes autophagic degradation of MDA5 by p62. Rnf144b deficiency greatly promotes IFN production and inhibits EMCV replication in vivo. Importantly, *Rnf144b*⁻/⁻ mice has a significantly higher overall survival rate than wild-type mice upon EMCV infection. Collectively, our results identify RNF144B as a negative regulator of innate antiviral response by targeting CARDs of MDA5 and mediating autophagic degradation of MDA5.**

**Keywords** MDA5; E3 Ligase; RNF144B; Innate Immunity; Type I Interferon
**Subject Categories** Immunology; Microbiology, Virology & Host Pathogen Interaction; Post-translational Modifications & Proteolysis

## Introduction

The ability to distinguish self from non-self is the essential function of the immune system. The innate immune system is the first line of host defense against invading microorganisms. The RIG-I-like receptors (RLRs) detect viral RNA ligands or processed self RNA in the cytoplasm and induce antiviral innate immune responses (Loo and Gale, 2011; Takeuchi and Akira, 2008). To date, three RLR members have been characterized: RIG-I (retinoic acid-inducible gene I), MDA5 (melanoma differentiation-associated factor 5), and LGP2 (laboratory of genetics and physiology 2) (Saito et al, 2007).

RIG-I and MDA5 contain three distinct domains: tandem caspase-recruitment domains (CARDs) at their N terminal, which mediate downstream signaling; a central DExD/H helicase domain with an ATP-binding motif; and a C-terminal region. Although there are some sequence, structural, and functional resemblances between RIG-I and MDA5, they have unique conformational and functional properties and are regulated by different post-translational modifications. RIG-I and MDA5 sense different types of RNAs. RIG-I primarily detects viral 5-diphosphate (5′pp) or 5′-triphosphate (5′ppp) of double-stranded RNA (dsRNA) and short dsRNA (Goubau et al, 2014; Hornung et al, 2006; Luo et al, 2011; Schlee et al, 2009). MDA5 preferably recognizes relatively long dsRNA in the genome of dsRNA viruses or dsRNA replication intermediates of viruses (Kato et al, 2008). Endogenous retroelements, such as Alu elements also activate MDA5 signaling (Ahmad et al, 2018). Upon binding to their ligands, RIG-I and MDA5 undergo conformational changes and recruit MAVS (also known as VISA, CARDIF, and IPS-1) through their CARDs. MAVS localizes to the outer mitochondrial membrane and multimerizes into a prion-like filament structure, which acts as a central platform for the assembly of a virus-induced complex that activates TAK1-IKK and TBK1/IKKε kinases. Subsequently, transcription factors NF-κB and IRF3 are activated to drive type I interferon (IFN) production and antiviral gene expression (Rehwinkel and Gack, 2020).

RLR signaling is regulated by multiple mechanisms to prevent aberrant production of interferon that could result in the development of immune toxicity or autoimmune disorders. Post-translational modifications, particularly ubiquitination of key components, are critical for the regulation of the RLR signaling pathway. K63-linked ubiquitination of RIG-I and MDA5 facilitates their activation. K48-linked ubiquitination promotes proteasomal degradation of RIG-I and MDA5 (Lang et al, 2017). TRIM40 and Parkin also catalyze K48-linked ubiquitination of MDA5 (Bu et al, 2020; Zhao et al, 2017).

In addition to the proteasome pathway, the autophagic pathway can also mediate protein degradation. Autophagy is an evolutionarily conserved biological process that contributes to cytoplasmic quality control, cellular metabolism, and innate and adaptive immunity (Clarke and Simon, 2019; Deretic, 2021; Deretic et al, 2013; Herzig and Shaw, 2018; Levine et al, 2011; Liu and Sabatini, 2020; Ma et al, 2013; Mariño et al, 2014; Morishita and Mizushima, 2019). Autophagy is an essential protective mechanism that helps cells fend off external threats, including pathogens such as viruses and bacteria, and internal sources of inflammation, including

[1]State Key Laboratory for Animal Disease Control and Prevention, College of Veterinary Medicine, Lanzhou University, Lanzhou Veterinary Research Institute, Chinese Academy of Agricultural Sciences, Lanzhou 730000, China. [2]Department of Preventive Veterinary Medicine, College of Veterinary Medicine, Northeast Agricultural University, Harbin, Heilongjiang, China. [3]Gansu Province Research Center for Basic Disciplines of Pathogen Biology, Lanzhou 730046, China. [4]These authors contributed equally: Guoxiu Li, Jing Zhang. ✉E-mail: zhangjing@caas.cn; tanglijie@neau.edu.cn; luzengjun@caas.cn

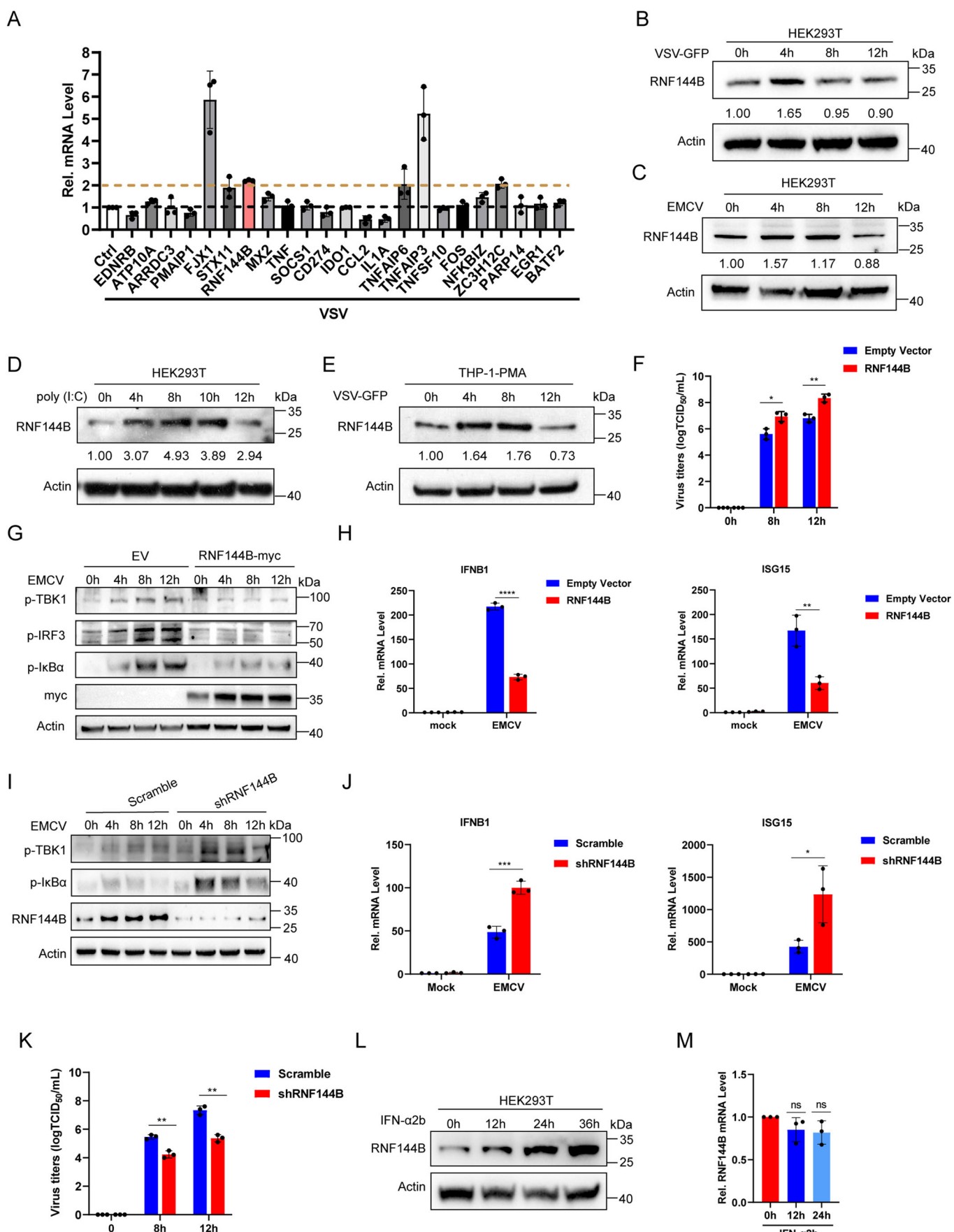

◄ **Figure 1.  Identification of RNF144B as a negative regulator of RLR-signaling.**

(A) Real-time QPCR analysis confirming the expression levels of VSV upregulated or downregulated genes. (B–D) HEK293T cells were treated with VSV-GFP (MOI = 0.1), EMCV (MOI = 1), poly (I:C) (1 µg/mL) for indicated time points. Cell lysates were used for immunoblot analysis with the indicated antibodies. (E) THP-1 cells were treated with PMA, then infected with VSV-GFP (MOI = 0.1) for indicated time points. Cell lysates were used for immunoblot analysis with the indicated antibodies. (F) HEK293T cells were transfected with empty vector or RNF144B-Myc plasmid for 18 h, and cells were infected with EMCV (MOI = 1) for indicated time points. The cell supernatant was harvested and analyzed by TCID$_{50}$ (50% Tissue Culture Infective Dose) assay. $n = 3$ biological replicates. Statistical significance was determined by two-tailed unpaired Student's t-test. *$P = 0.02250$, **$P = 0.00850$. (G) HEK293T cells were transfected with empty vector or RNF144B-Myc plasmid. After 18 h, cells were infected with EMCV for the indicated time points. Cells were harvested and protein extracts were analyzed by immunoblot using the indicated antibodies. (H) HEK293T cells were transfected with empty vector or RNF144B-Myc plasmid. After 18 h, cells were infected with EMCV for the indicated time points and subjected to qPCR analysis for IFNB1 and ISG15 mRNA level. $n = 3$ biological replicates. Statistical significance was determined by two-tailed unpaired Student's t-test. ****$P = 0.000010$, **$P = 0.00550$. (I) Control or RNF144B KD HEK293T cells were infected with EMCV (MOI = 1) for indicated time points. Cell lysates were used for immunoblot analysis with the indicated antibodies. (J) Control or RNF144B KD HEK293T cells were infected with EMCV (MOI = 1) for indicated time points and subjected to qPCR analysis for IFNB1 and ISG15 mRNA level. $n = 3$ biological replicates. Statistical significance was determined by two-tailed unpaired Student's t-test. ***$P = 0.00100$, *$P = 0.03590$. (K) Control or RNF144B KD HEK293T cells were infected with EMCV (MOI = 1) for indicated time points. The cell supernatant was harvested and analyzed by TCID$_{50}$. $n = 3$ biological replicates. Statistical significance was determined by two-tailed unpaired Student's t-test. **$P = 0.00190$ (8 h), **$P = 0.00100$ (12 h). (L) HEK293T cells were treated with IFN-α2b for indicated time points and subjected to immunoblot analysis of RNF144B expression. (M) HEK293T cells were treated with IFN-α2b for indicated time points and subjected to qPCR analysis of RNF144B mRNA level. $n = 3$ biological replicates. Statistical significance was determined by two-tailed unpaired Student's t-test. "ns" indicates no significant difference. Data information: Data shown are representative of at least three biological replicates, with each data point representing a biological experiment. Error bars are presented as mean ± SD. Statistical significance was determined by Student's t-test. Source data are available online for this figure.

molecular aggregates and damaged organelles. It is coordinated by a series of autophagy-associated (ATG) proteins and forms double-membrane structures called phagosomes that involve the sequestration of cytoplasmic components. The phagophores mature into autophagosomes, which fuse with the lysosomes, leading to the degradation of their cargo by acidic hydrolases. Selective autophagy is highly specific when choosing their cargo. Ubiquitination on the autophagic cargo is utilized as a degradation signal that is recognized by broad-spectrum selective autophagy receptors termed SLRs (sequestosome-like receptors: p62, NBR1, OPTN, NDP52, TAX1BP1, etc.) (Birgisdottir et al, 2013). CCDC50 is a novel autophagic adapter specifically recognizes K63-polyubiquitinated RIG-I/MDA5 and delivers them for autophagic degradation (Hou et al, 2021).

The discovery and identification of unknown RLR-mediated antiviral innate immune regulatory factors will enrich the knowledge of RLR signaling regulation and further guide antiviral practice. In this study, we analyzed RNA-seq data from host cells infected with Porcine reproductive and respiratory syndrome virus, Foot-and-mouth disease virus, and Seneca virus A. We found that E3 ligase RNF144B negatively regulates IFN production and facilitates EMCV replication in vitro. Further study revealed that RNF144B catalyzed the K27 and K33 polyubiquitination of MDA5 on lysine 23 and lysine 43 and promoted its autophagic degradation by p62. Moreover, $Rnf144b^{-/-}$ mice exhibited a higher survival rate upon EMCV infection compared with wild-type mice.

## Results

### Identification of RNF144B as a negative regulator of RLR-signaling

In order to identify host factors that regulate the antiviral response to RNA viruses, we analyzed RNA-Seq data from host cells infected with Porcine reproductive and respiratory syndrome virus, Foot-and-mouth disease virus, and Seneca virus A (Dataset EV1). Our findings revealed that 20 genes were upregulated following all three viral infections (Fig. EV1A). Notably, some of these genes are known

regulators of the host antiviral immune response, including MX2, SOCS1, CD274, IDO1, CCL2, IL1A, TNFAIP3, TNFSF10, NFKBIZ, ZC3H12C, PARP14, and BATF2 (Ahmed et al, 2015; Huang et al, 2019; Li et al, 2021; Liu et al, 2013; Liu et al, 2020; Qin et al, 2019; Raghu et al, 2017; Tang et al, 2021; Wang et al, 2020; Yi et al, 2018). We further detected the mRNA expression levels of these 20 genes after Vesicular Stomatitis Virus (VSV) infection by qPCR. The results showed that five genes were upregulated more than 2-fold, namely FJX1, RNF144B, TNFAIP3, TNFAIP6, and ZC3H12C (Fig. 1A). TNFAIP3 and TNFAIP6 have been reported to regulate innate immune response to viral infection (Shembade and Harhaj, 2012; Yu et al, 2016). To explore if they could regulate VSV replication, we transfected siRNAs targeting FJX1, RNF144B, and ZC3H12C into HEK293T cells (Fig. EV1B–D). The results showed that knockdown of ZC3H12C and FJX1 had no significant effect on VSV replication (Fig. EV1E). However, knockdown of RNF144B impaired VSV replication (Fig. EV1F). Furthermore, we also observed that knockdown of RNF144B suppressed EMCV replication (Fig. EV1G).

The protein expression level of RNF144B exhibited a marked increase during the initial stages of VSV and EMCV infections, as well as poly (I:C) transfection in 293T cells, and then gradually returned to baseline levels (Fig. 1B–D). A similar trend was observed in THP-1 cells (Fig. 1E). To explore the physiological role of RNF144B, we introduced RNF144B into 293T cells and observed that its overexpression augmented both viruses copy numbers and virus titers of EMCV (Figs. 1F and EV1H). Furthermore, we investigated whether RNF144B affects innate immune signaling in response to RNA virus infection. The overexpression of RNF144B substantially reduced the phosphorylation of TBK1, IRF3, and IκBα in 293T cells during EMCV infection (Fig. 1G). Moreover, RNF144B overexpression impeded IFN production and ISG induction induced by EMCV infection (Fig. 1H). Conversely, knockdown of RNF144B in 293T cells led to a significant increase in IFN production, ISG induction, and levels of phosphorylated IκBα and TBK1 upon EMCV infection (Fig. 1I,J). In addition, the knockdown of RNF144B impaired EMCV replication (Fig. 1K). Moreover, the transfection of poly (I:C) induced higher levels of IFNB1 and ISG15 expression in cells with RNF144B knockdown (Fig. EV1I–M). To investigate whether RNF144B is induced by

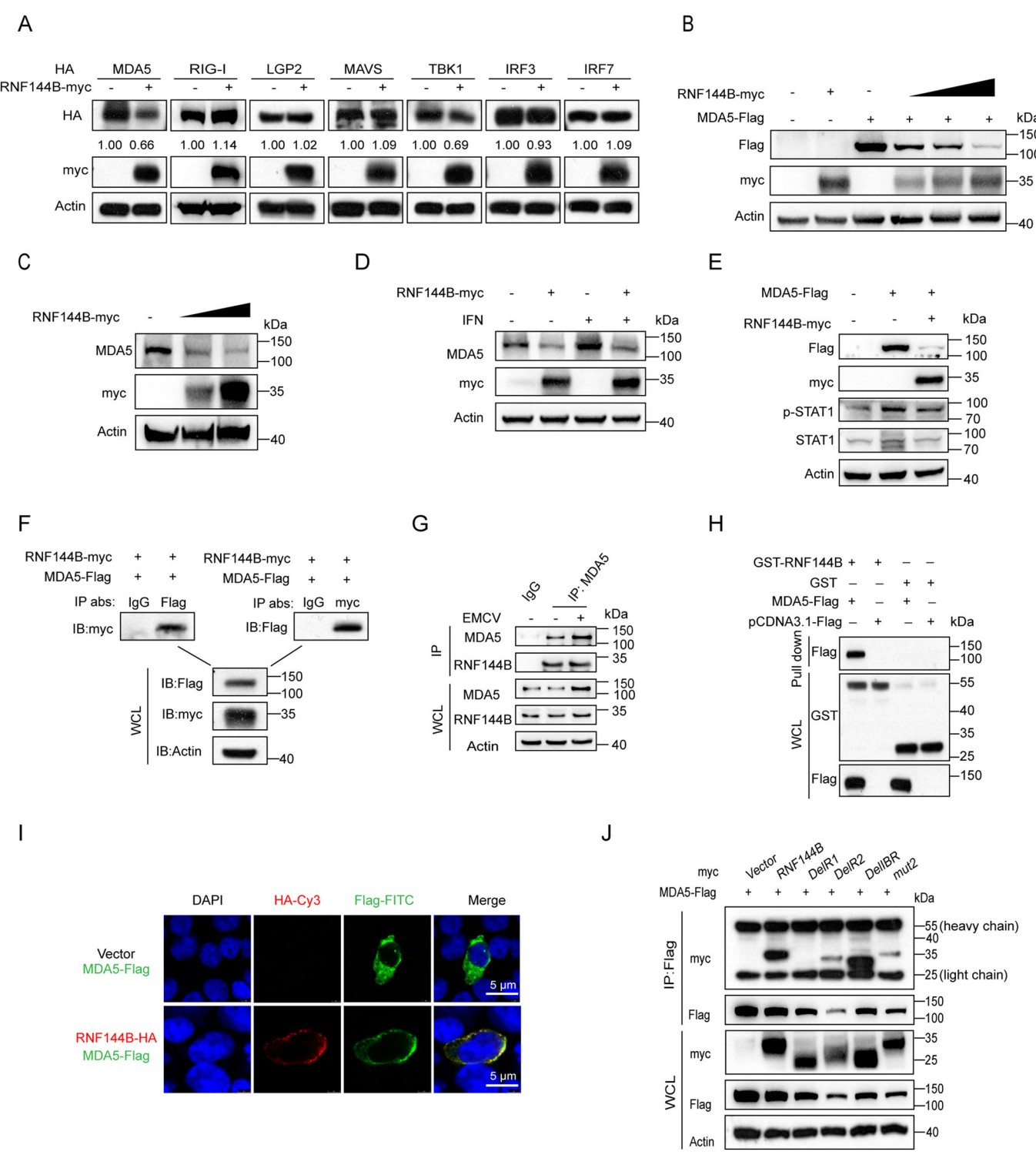

IFN, we treated 293T cells with IFN-α and assessed the mRNA and protein levels of RNF144B. The findings revealed that the protein level of RNF144B increased in a time-dependent manner upon IFN-α treatment, whereas the mRNA expression level remained unaffected (Fig. 1L,M). Transfection of poly(I:C) and virus infection slightly induced the expression of RNF144B mRNA expression levels (~2-fold) (Figs. 1A and EV1N). These results

suggested that RNF144B is not a typical ISG, the protein level increases may be attributed to post-translational regulation.

## RNF144B facilitates the degradation of MDA5

To elucidate the molecular mechanisms through which RNF144B dampens RLR signaling, we conducted co-transfections of

◀

Figure 2.   RNF144B facilitates degradation of MDA5.

(A) HEK293T cells were co-transfected with empty vector or RNF144B-Myc plasmid, and HA-tagged RIG-I, MDA5, LGP2, TBK1, MAVS, IRF3, and IRF7 plasmid, respectively. 18 h post-transfection, cell lysates were analyzed by immunoblot using the indicated antibodies. (B) HEK293T cells were transfected with plasmids encoding RNF144B (0, 0.5, 1, 2 µg) and MDA5-Flag (2 µg). 18 h post-transfection, cell lysates were used for immunoblot analysis with the indicated antibodies. (C) HEK293T cells were transfected with plasmid encoding RNF144B (0, 1.5, 3 µg). 18 h post-transfection, protein extracts were used for immunoblot analysis of the endogenous MDA5 protein level. (D) A549 cells were transfected with empty vector or RNF144B-Myc plasmid. 12 h post-transfection, cells were treated with interferon, then, protein extracts were used for immunoblot analysis of the endogenous MDA5 protein level. (E) HEK293T cells were transfected with plasmids encoding RNF144B and MDA5-Flag. 18 h post-transfection, cell lysates were used for immunoblot analysis with the indicated antibodies. (F) HEK293T cells were transfected with plasmids encoding Myc-tagged RNF144B and Flag-tagged MDA5. 18 h post-transfection, cell lysates were harvested and immunoprecipitated using anti-Flag and anti-myc antibodies, followed by immunoblots using the indicated antibodies. (G) A549 cells were infected with EMCV (MOI = 2) for indicated time points, cell lysates were harvested and immunoprecipitated using anti-MDA5 antibody, followed by immunoblots using the indicated antibodies. (H) GST pulldown assay of GST, GST-RNF144B (12 µg) and MDA5-Flag. (I) Confocal microscopy analysis of HEK293T cells co-transfected with plasmids of MDA5-Flag (green) with RNF144B-HA (red). Scale bar, 5 µm. (J) HEK293T cells were co-transfected with wild-type or mutants of RNF144B-Myc and MDA5-Flag. 18 h post-transfection, cell lysates were harvested and immunoprecipitated using anti-Flag antibody, followed by immunoblots using the indicated antibodies. Source data are available online for this figure.

RNF144B with MDA5, RIG-I, LGP2, MAVS, TBK1, IRF3, and IRF7 in HEK293T cells. The results revealed that the overexpression of RNF144B resulted in a significant decrease in the protein levels of MDA5 and a slight decrease in the expression of TBK1, whereas there was no significant effect on other molecules (Fig. 2A). We observed that RNF144B promoted the degradation of MDA5 in a dose-dependent manner in HEK293T cells (Fig. 2B). Upon ectopic expression of RNF144B, we noted the promotion of both exogenous and endogenous MDA5 degradation (Figs. 2B,C and EV2A). However, the overexpression of RNF144B did not promote RIG-I degradation (Fig. EV2B). It slightly degraded TBK1 as previously reported (Fig. EV2C) (Zhang et al, 2019). To test whether RNF144B degrades IFN induced MDA5 protein levels, we treated 293T cells with IFN and observed that it also degrades IFN induced MDA5 (Fig. 2D). Next, we investigated the impact of RNF144B on MDA5 overexpression-induced IFN signaling. Our findings demonstrated that RNF144B impaired MDA5-induced STAT1 phosphorylation and mRNA expression levels of STAT1 and ISG15 (Figs. 2E and EV2D). Subsequent Co-immunoprecipitation (Co-IP) results demonstrated the interaction between RNF144B and MDA5 (Fig. 2F). This interaction was consistently observed regardless of EMCV infection (Fig. 2G). Furthermore, GST pull-down assays underscored a direct interaction between RNF144B and MDA5 in vitro (Fig. 2H). Immunofluorescence staining confirmed co-localization between RNF144B and MDA5 in the cytoplasm (Fig. 2I). RNF144B comprises two N-terminal RING (Really Interesting New Gene) finger domains, an IBR (In Between RING) domain, and a CTD domain (Fig. EV2E). Domain mapping analysis revealed that the RING1 domain of RNF144B played a pivotal role in its interaction with MDA5 (Fig. 2J).

## RNF144B promotes K27- and K33-linked ubiquitination of MDA5

As RNF144B functions as an E3 ubiquitin ligase, we investigated its effects on the ubiquitination of MDA5. In HEK293T cells, we co-transfected Flag-tagged MDA5, myc-tagged RNF144B, and HA-tagged ubiquitin to perform in vivo ubiquitin assay. The immunoprecipitation result demonstrated that RNF144B facilitated the ubiquitination of MDA5 in a dose-dependent manner (Fig. 3A). To pinpoint the specific residues in RNF144B responsible for catalyzing MDA5 ubiquitination, we searched the conserved cysteine residues within the RING domain. There are 12 cysteine

residues in each of the RING domain (Fig. EV2F). Subsequently, we generated a series of single-point mutations in RNF144B, wherein the cysteine residue was mutated to alanine. Then, we performed in vivo ubiquitin assay with these mutants. The findings indicated that mutants containing C48A, C80A, and C219A of RNF144B significantly diminished its ability to catalyze MDA5 ubiquitination (Fig. 3B). In addition, these mutants exhibited reduced proficiency in degrading MDA5. Subsequently, we generated a plasmid with a three-point mutation (C48/80/219A) and observed the complete disappearance of MDA5 ubiquitination (Fig. 3C). Importantly, overexpression of this mutant did not affect viral copy numbers, virus titers of EMCV and phosphorylation levels of p-TBK1, p-IRF3, and p-IκBα (Figs. 3D,E and EV2G). These results collectively suggest that amino acids C48, C80, and C219 of RNF144B plays a pivotal role in its E3 ubiquitin ligase activity on the ubiquitination of MDA5.

We then proceeded to examine the types of polyubiquitin chains conjugated to MDA5 by RNF144B. To accomplish this, we utilized ubiquitin mutants in which all lysine residues except one were simultaneously mutated to arginine (K-O), or each of the seven lysine residues was individually mutated to arginine (K-R) (Fig. EV2H). These mutants were subjected to testing for their ability to be conjugated to MDA5 by RNF144B. The results unveiled the mutation of lysine 27 (K27R-Ub) or lysine 33 (K33R-Ub) to arginine markedly reduced its capacity to be conjugated to MDA5 by RNF144B (Fig. 3F). In contrast, that RNF144B-mediated MDA5 polyubiquitination was detectable in the presence of K27O-Ub or K33O-Ub, but not with other Ub mutants (Fig. 3G). Moreover, the ubiquitination of MDA5 was completely abolished when transfected with K27/33R-Ub double mutants (Fig. 3H). These findings conclusively indicate that RNF144B induces K27- and K33-linked polyubiquitination of MDA5. Moreover, In Rnf144b KO cells, EMCV infection induced MDA5 ubiquitination was significantly decreased compared to WT cells (Fig. EV2I). The immunoprecipitation results demonstrated that RNF144B enhanced the ubiquitination of endogenous MDA5 (Fig. EV2J).

To determine which domain of MDA5 is ubiquitinated by RNF144B, we transfected vectors expressing His-tagged ubiquitin, Myc-tagged RNF144B and HA-tagged full-length (WT) MDA5 or truncated MDA5 that contained CARDs, ATP and CTD domain in HEK293T cells (Fig. EV2K). Immunoprecipitation analysis demonstrated that the ubiquitination of WT or CARDs of MDA5 was

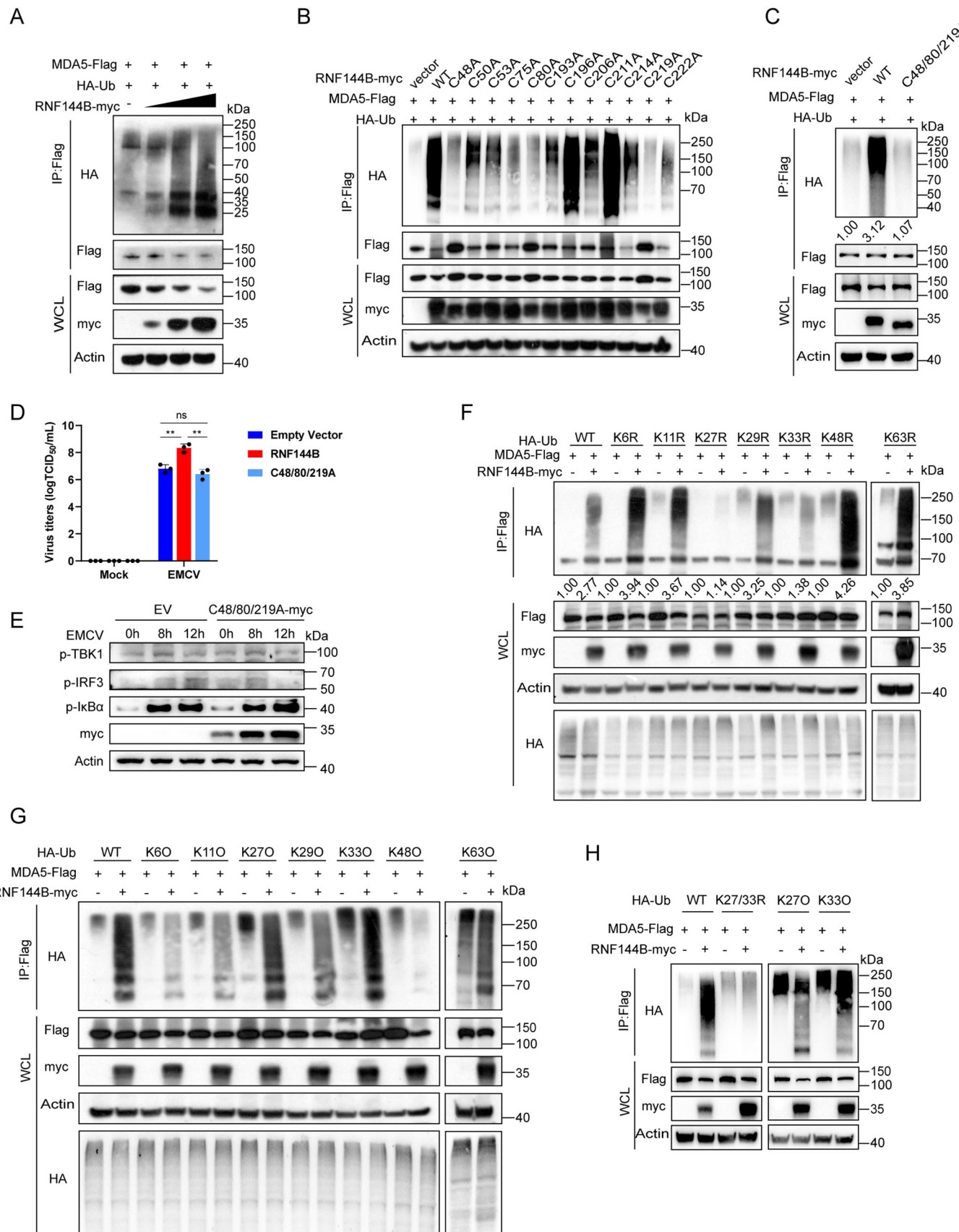

**Figure 3.  RNF144B promotes K27- and K33-linked ubiquitination of MDA5.**

(A) HEK293T cells were co-transfected with HA-Ub, MDA5-Flag, along with empty vector or RNF144B-myc, cell lysates were subjected to denature-immunoprecipitation with anti-Flag antibody followed by immunoblot analysis with indicated antibodies. (B, C) HEK293T cells co-transfected with HA-Ub, MDA5-Flag, along with empty vector, wild-type RNF144B-myc or RNF144B-myc mutants. Cell lysates were subjected to denature-immunoprecipitation with anti-Flag antibody followed by immunoblot analysis with indicated antibodies. (D) HEK293T cells were transfected with empty vector, RNF144B-myc or C48/80/219A-Myc plasmid. After 18 h, cells were infected with EMCV (MOI = 1). $n = 3$ biological replicates. The cell supernatant was harvested and analyzed by TCID$_{50}$ (50% Tissue Culture Infective Dose) assay. Statistical significance was determined by two-tailed unpaired Student's t-test. "ns" indicates no significant difference, **$P = 0.0034$ (RNF144B), **$P = 0.00210$ (C48/80/219 A). (E) HEK293T cells were transfected with empty vector or C48/80/219A-Myc plasmid. After 18 h, cells were infected with EMCV (MOI = 1) for the indicated time points. Cells were harvested and protein extracts were analyzed by immunoblot using the indicated antibodies. (F, G) HEK293T cells co-transfected with HA-Ub (WT and its mutants), MDA5-Flag, and empty vector or RNF144B-Myc plasmids and maintained in the presence of CQ (20 μM) for 18 h. Cell lysates were subjected to denature-immunoprecipitation with anti-Flag antibody followed by immunoblot analysis with indicated antibodies. (H) HEK293T cells co-transfected with HA-Ub (WT and its mutants), MDA5-Flag, and empty vector or RNF144B-Myc plasmids for 18 h. Cell lysates were subjected to denature-immunoprecipitation with anti-Flag antibody followed by immunoblot analysis with indicated antibodies. Data information: Data shown are representative of at least three biological replicates, with each data point representing a biological experiment. Error bars are presented as mean ± SD. Statistical significance was determined by Student's t-test. Source data are available online for this figure.

strongly enhanced by overexpression of RNF144B (Fig. 4A,B). However, the ubiquitination of ATP and CTD of domain had no significant effect on overexpression of RNF144B (Fig. 4C,D). To determine the ubiquitination site of MDA5 by RNF144B, we constructed single point mutations in which all lysines in the CARDs domain of MDA5 were mutated to arginine (Fig. 4E). Immunoprecipitation analysis of ubiquitin showed that the ubiquitination of K23R and K43R mutants by RNF144B was significantly reduced (Fig. 4F). Further, a mutant containing both K23R and K43R mutations were constructed. RNF144B-mediated ubiquitination of this mutant was significantly reduced (Fig. 4G). Flag-tagged wild-type and mutant MDA5 were co-transfected with RNF144B into 293T cells, and Western blot analysis showed that K23R and K43R mutations could not be degraded by RNF144B (Fig. 4H), and overexpression of RNF144B did not lead to degradation of the K23/43 R double mutant (Fig. 4I). These results collectively suggest that amino acids K23 and K43 of MDA5 are essential for its ubiquitination and degradation. Moreover, it has been reported that sumoylation at K43 and K865 by TRIM38 suppresses its K48-linked polyubiquitination and degradation (Hu et al, 2017). To investigate whether sumoylation affects K27/K33 ubiquitination of MDA5, we co-transfected TRIM38 and RNF144B and found that TRIM38 overexpression attenuated RNF144B-induced degradation of MDA5 (Fig. EV2L). Taken together, RNF144B-mediated ubiquitination and TRIM38-mediated sumoylation at the K43 site of MDA5 may compete exclusively with each other to regulate MDA5 stability.

## RNF144B promotes p62-mediated selective autophagic degradation of MDA5

The ubiquitin-proteasome and autophagy-lysosome pathways represent the classic protein degradation mechanisms in eukaryotic cells. Consequently, we employed proteasome inhibitor MG132, autophagy inhibitor CQ, and apoptosis inhibitor Z-VAD to elucidate which pathway was involved in the degradation of MDA5 facilitated by RNF144B. The results unequivocally demonstrated that RNF144B-mediated MDA5 degradation was impeded by the autophagy inhibitor CQ, while having no discernible effect when treated with MG132 and Z-VAD (Fig. 5A). To confirm these findings, we used alternative autophagy inhibitors, 3-MA and the autolysosome inhibitor Baf-A1, to block autophagic degradation of MDA5. The results revealed that the degradation of MDA5

facilitated by RNF144B was reverted upon incubation with 3-MA, Baf-A1, and CQ (Fig. 5B). In summary, these findings collectively suggest that RNF144B promoted the degradation of MDA5 through the autophagy pathway.

Autophagy adapter proteins, such as SQSTM1, NBR1, NDP52, and OPTN, act as intermediaries that bridge cargo substrates to autophagosomes for selective degradation. In a quest to identify the autophagy adapter responsible for recognizing ubiquitinated MDA5 and facilitating its degradation, we conducted Co-IP assays involving MDA5 and these autophagy adapters. The results revealed that MDA5 exhibited interactions with p62, Tollip, and CCDC50, while showing no affinity for other receptors like NBR1, OPTN, and BNIP3L (Fig. 5C). To investigate whether MDA5 interacts with p62, Tollip, and CCDC50 during viral infection, we conducted Co-IP analyses before and post-EMCV infection. Interestingly, we observed that both p62, Tollip and CCDC50 interacted with MDA5 under normal physiological conditions, EMCV infection does not affect their interaction (Fig. 5D). Subsequently, we established cell lines knockdown of p62 and Tollip (Fig. 5E,F). Notably, the degradation of MDA5 mediated by RNF144B was reverted in p62-knockdown cells, whereas it remained unaffected in Tollip-knockdown cells (Fig. 5G,H). In addition, the degrading effect of RNF144B on MDA5 could be partially reversed in cells subjected to *siRNA* knockdown of CCDC50 (Figs. 5I and EV3A). Furthermore, we observed co-localization between MDA5 and p62/Tollip/CCDC50 (Figs. 5J and EV3B). To determine whether p62 recognize K27 and K33 ubiquitinated MDA5, we co-transfected HA-tagged p62, Flag-tagged MDA5 and K27O/K33O/K27R/K33R ubiquitin. Immuno-flourance assay showed that p62 co-localized with MDA5-K27/K33 puncta, but not with MDA5-K27/K33R (Fig. 5K). In conclusion, these results collectively indicate that p62 serves as an autophagy receptor, recognizing K27- and K33-polyubiquitinated MDA5 and facilitating its autophagic degradation.

## RNF144B inhibits antiviral responses in vitro and in vivo

In order to elucidate the regulatory role of RNF144B in RLR-signaling-mediated antiviral responses in vivo, we employed the CRISPR/Cas9 system to generate *Rnf144b* knockout (KO) mice. These mice were engineered to delete exon 2 and exon 3 (Fig. EV3C). The successful knockout of *Rnf144b* was verified through PCR analysis of tail genome DNA and by immunoblot

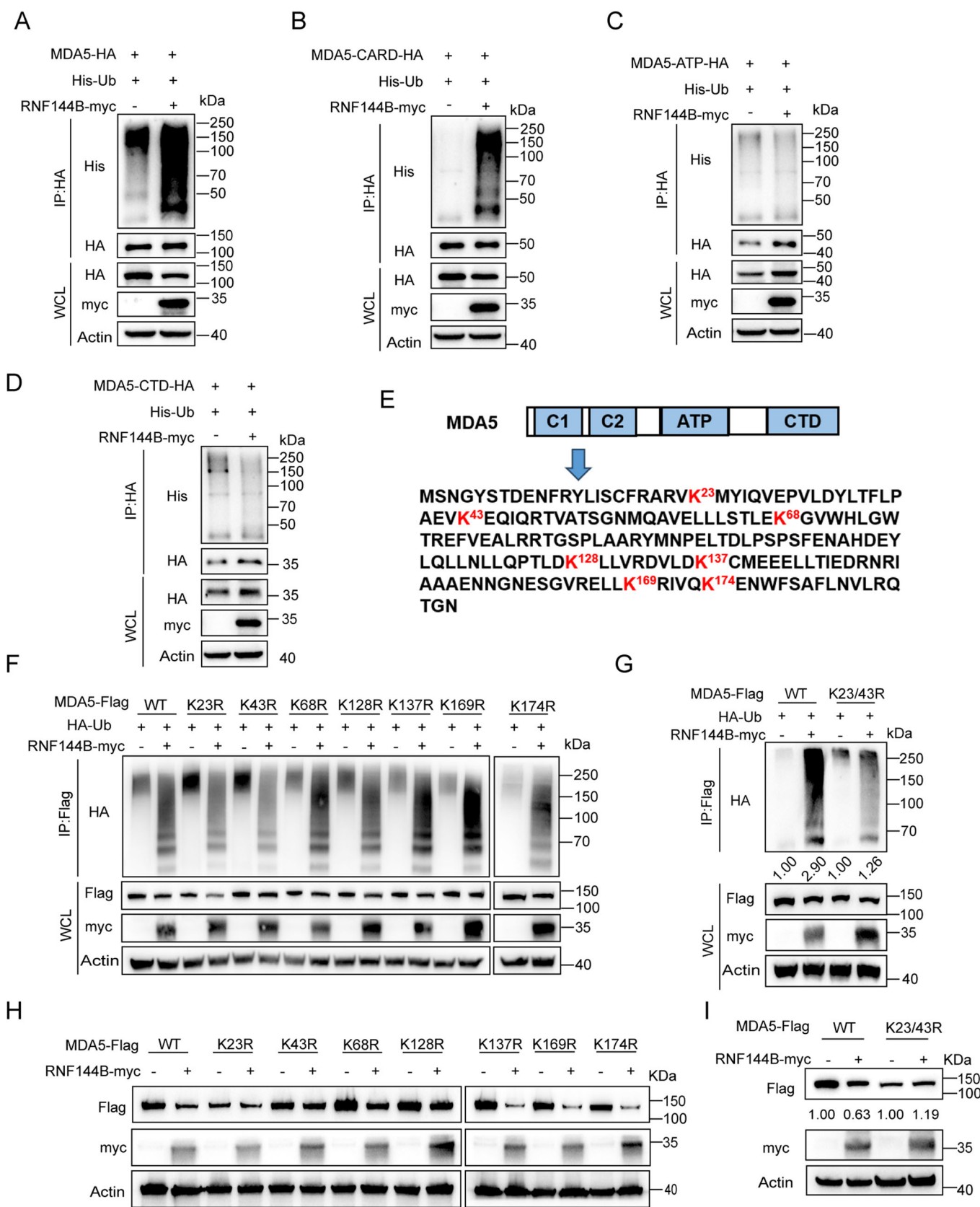

**Figure 4. K23 and K43 of MDA5 are essential for its ubiquitination by RNF144B.**

(A–D) HEK293T cells were co-transfected with His-Ub, MDA5-HA (wild-type and mutants), along with empty vector or RNF144B-myc. Cell lysates were subjected to denature-immunoprecipitation with anti-Flag antibody followed by immunoblot analysis with indicated antibodies. (E) The model of point mutant plasmid of MDA5. (F, G) HEK293T cells were co-transfected with HA-Ub, MDA5-Flag, along with empty vector and RNF144B-myc plasmids and maintained in the presence of CQ (20 μM) for 18 h. Cell lysates were subjected to denature-immunoprecipitation with anti-Flag antibody followed by immunoblot analysis with indicated antibodies. (H, I) HEK293T cells were co-transfected MDA5-Flag (wild-type and mutants) and empty vector or RNF144B-myc. 18 h post-transfection, cell lysates were used for immunoblot analysis with the indicated antibodies. Source data are available online for this figure.

analysis conducted on lysis of mice heart (Fig. EV3D). Importantly, $Rnf144b^{-/-}$ mice exhibited normal growth and development, displaying no discernible physical or behavioral abnormalities. The proportions of lymphoid and myeloid cells within the spleen cells of $Rnf144b^{-/-}$ mice closely resembled those of their wild-type littermates, suggesting that $Rnf144b$ might not play a pivotal role in the development of myeloid cells (Fig. EV3E,F). Subsequent findings revealed a notable inhibition of EMCV replication in $Rnf144b^{-/-}$ MEFs, BMDCs, and BMDMs when compared to their wild-type counterparts (Figs. 6A,B and EV4A–C). Furthermore, the deficiency of $Rnf144b$ potentiated the EMCV-induced phosphorylation of TBK1 and IκBα in MEFs and BMDMs (Figs. 6C and EV4D). In addition, mRNA expression levels of Ifnb1 and pro-inflammatory cytokines were elevated in MEFs, BMDMs, and BMDCs from $Rnf144b$ KO mice compared to their WT counterparts (Figs. 6D and EV4E,F). Intriguingly, in $Rnf144b^{-/-}$ MEFs, the protein levels of MDA5 were upregulated more significantly after EMCV infection compared to wild-type MEFs (Fig. 6C). In addition, MDA5 degradation kinetics were faster in wild-type MEFs than in $Rnf144b$ knockout MEFs in the presence of CHX, a protein synthesis inhibitor (Fig. 6E). However, the protein level of RIG-I was not increased in $Rnf144b$ knockout MEFs (Fig. EV4G).

To delve into the function of RNF144B in host antiviral innate responses in vivo, we subjected $Rnf144b^{-/-}$ mice to intraperitoneal injection with EMCV at a non-lethal dose ($5×10^6$ PFU/mouse). Notably, EMCV replication in heart and brain tissues of $Rnf144b^{-/-}$ mice was significantly reduced compared to control littermates (Fig. 6F). Correspondingly, the mRNA expression levels of IFN-β and the pro-inflammatory cytokines IL-6 and TNF-α in the brain of $Rnf144b^{-/-}$ mice were markedly higher than those in WT mice (Fig. 6G). ELISA results further supported these findings by revealing significantly elevated levels of IFN-β in the serum of $Rnf144b^{-/-}$ mice as compared to wild-type mice (Fig. 6H). To assess the impact of $Rnf144b$ on the survival of mice following EMCV infection, we infected mice through intraperitoneal injection with a higher titer of EMCV at a lethal dose ($6 × 10^7$ PFU/mouse). $Rnf144b^{-/-}$ mice exhibited a notably higher overall survival rate when contrasted with wild-type mice (Fig. 6I). We tested basal Ifnb1 and ISGs (Isg15, Ifit1, Ifit2, Ifit3, Mx1) mRNA expression levels in WT and $Rnf144b$ KO cells. The results showed that basal level of Ifnb1 and ISGs was slightly higher in $Rnf144b$ KO cells (Fig. EV5A). In addition, STAT1 protein level was increased in $Rnf144b$ KO cells at basal state (Fig. EV5B). These results indicating that RNF144B also moderately regulated tonic IFN signaling. To investigate whether RNF144B regulates IFN signaling in response to IFN treatment, we analyzed the phosphorylation levels of STAT1 and the transcription levels of ISGs. The results showed that the levels of p-STAT1 and ISG transcription were elevated in Rnf144b

KO cells following IFN treatment (Fig. EV5C,D). In addition, the p-TBK1 level was increased in Rnf144b KO cells following viral infection. This increase was reversed by MDA5 knockdown, indicating that RNF144B's regulation of TBK1 is dependent on MDA5 (Fig. EV5E). Taken together, these results demonstrated RNF144B is involved in regulating both tonic and ligand activation of IFN signaling.

To investigate whether RNF144B specifically regulated MDA5 signaling, we assessed the response of $Rnf144b$ KO cells to ligands for other signaling pathways, including short poly (I:C), CL097 (TLR7 ligand), (TLR4 ligand), and poly (dA: dT). The findings indicated that high molecular weight poly (I:C) notably induced TBK1 phosphorylation in $Rnf144b$ KO cells, while short poly (I:C) induced TBK1 phosphorylation to a lesser extent, and other ligands did not exhibit significant differences (Fig. EV6A,B). In addition, RNF144B deficiency led to increased expression of type I IFN and other pro-inflammatory cytokines upon transfection with long poly (I:C) and slight increase of short poly (I:C), and no observable respond to poly (dA:dT) (Fig. EV6C–E). Furthermore, VSV replication was impaired in $Rnf144b$ KO cells, whereas HSV replication remained unaffected (Fig. EV6F,G). These results demonstrate that $Rnf144b$ deficiency enhances immune responses against viral infection, suggesting a physiological function of RNF144B in the negative regulation of EMCV induced IFN production and antiviral innate immunity.

## Discussion

RLR signaling plays an essential role in recognizing virus-derived RNAs in the cytoplasm. Activation of RLR signaling results in an antiviral response necessary to suppress the spread of viruses. Limiting excessive activation of RLR signaling is critical to protect the host from an unbalanced response and inflammatory injury.

Ubiquitination mediated degradation is one of the regulatory mechanisms that repress RIG-I and MDA5 activity have been described. However, whether cellular E3 ubiquitin ligases are involved in regulating autophagic degradation of MDA5 remains to be elucidated. In this study, we demonstrated that RNF144B interacts with MDA5 through the RING1 domain, and the cysteines C48, C80, and C219 of RNF144B play a pivotal role in its E3 ubiquitin ligase activity and the ubiquitination of MDA5. E3 ligase RNF144B promotes K27 and K33-linked ubiquitination of MDA5. Autophagic adapter p62 recognized K27- and K33-polyubiquitinated MDA5 and facilitated its autophagic degradation. $Rnf144b^{-/-}$ mice showed that $Rnf144b$ deficiency enhanced EMCV-induced IFN-β production in vivo. $Rnf144b^{-/-}$ mice exhibited a notably higher overall survival rate when compared with wild-type mice.

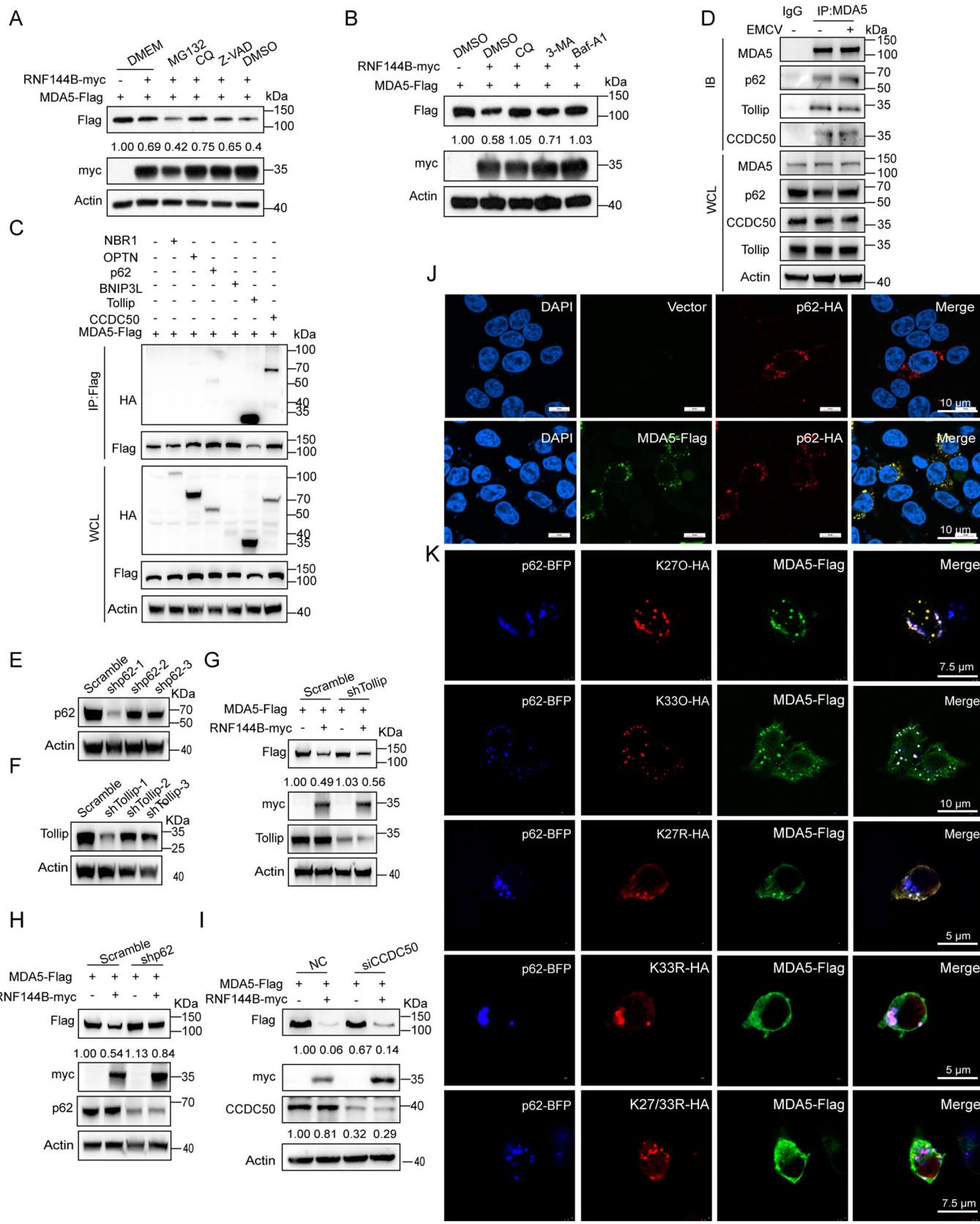

**Figure 5.  RNF144B promotes p62-mediated selective autophagic degradation of MDA5.**

(A) HEK293T cells were co-transfected with empty vector or RNF144B-Myc and MDA5-Flag for 12 h. The cells were treated with MG132 (10 μM) for 4 h, together with DMSO, CQ (20 μM), Z-VAD (10 μM) for 6 h, respectively. Cell lysates were used for immunoblot analysis with the indicated antibodies. (B) HEK293T cells were co-transfected with empty vector or RNF144B-Myc and MDA5-Flag for 12 h. The cells were treated with DMSO, 3MA (2.5 mM), CQ (20 μM), Baf-A1 (10 nM) for 6 h, respectively. Cell lysates were used for immunoblot analysis with the indicated antibodies. (C) HEK293T cells were co-transfected with MDA5-Flag and either NBR1, OPTN, BNIP3L, p62, Tollip, CCDC50, respectively. Cell lysates were analyzed by immunoprecipitation with anti-Flag antibody followed by immunoblot analysis with indicated antibodies. (D) A549 cells were infected with EMCV (MOI = 1) for indicated time points. Cell lysates were analyzed by immunoprecipitation followed by immunoblot analysis with indicated antibodies. (E, F) Immunoblot analysis of p62 and Tollip expression in HEK293T cells stably expressing *shRNA* against p62 and Tollip, respectively. (G) Tollip knockdown cells were transfected with MDA5-Flag and empty vector or RNF144B-myc. Cell lysates were analyzed by immunoblot analysis with indicated antibodies. (H) p62 knockdown cells were transfected with MDA5-Flag and empty vector or RNF144B-myc. Cell lysates were used for immunoblot analysis with the indicated antibodies. (I) HEK293T cells were transfected siCCDC50, along with MDA5-Flag and empty vector or RNF144B-myc. Cell lysates were used for immunoblot analysis with the indicated antibodies. (J) Confocal microscopy analysis of A549 cells co-transfected with MDA5-Flag (green) and p62-HA (red). Scale bar, 10 μm. (K) Confocal microscopy analysis of A549 cells transfected with MDA5-Flag (green), p62-BFP (blue), K27O, K33O, K27R, K33R, and K27R/K33R mutants (red). Source data are available online for this figure.

K48-linked ubiquitination promotes proteasomal degradation of RIG-I and MDA5. E3 ligases RNF125, C-Cbl, TRIM40, RNF122, and Parkin can mediate the K48-linked ubiquitination of RIG-I (Arimoto et al, 2007; Chen et al, 2013; Wang et al, 2016). TRIM40 and Parkin also catalyze K48-linked ubiquitination of MDA5 (Bu et al, 2020). Ubiquitination-induced selective autophagy is another pathway for protein degradation besides degradation via proteasome (Ji et al, 2022). Duck Tembusu virus nonstructural protein 2B has been reported to promote the autophagic degradation of MDA5 (Wu et al, 2022). K27-mediated autophagic degradation of proteins plays a key role in the regulation of innate immunity (Tracz and Bialek, 2021). For example, K27-linked auto-ubiquitination of TRIM23 is essential for its activation of TBK1. It facilitates TBK1 dimerization and ability to phosphorylate the selective autophagy receptor p62 (Sparrer et al, 2017). RNF178 promotes K27-linked polyubiquitination of MAVS at K7, which serves as a recognition signal for autophagic degradation (Yang et al, 2022). To date, K33-linked ubiquitination remains the least studied of all Ub linkage types and some discoveries strongly interlink K33 ubiquitination and autophagy (Nibe et al, 2018). In this study, we observed that RNF144B mediates K27 and K33-linked polyubiquitination of MDA5 at K23 and K43, which acts as a recognition signal for autophagic receptor p62. Previous report demonstrated that TRIM40 mediates the K27- and K48-linked polyubiquitination of MDA5 at K23, K43, and K68 (Zhao et al, 2017). Our findings, along with prior studies, suggest that ubiquitination at the K23 and K43 sites of MDA5 is crucial for its degradation. Moreover, sumoylation at K43/K865 of MDA5 by TRIM38 prevented its degradation (Hu et al, 2017). Our results showed that overexpression of TRIM38 attenuated the degradation of MDA5 induced by RNF144B (Fig. EV2L). These results suggest that RNF144B-mediated ubiquitination and TRIM38-mediated sumoylation at the K43 site of MDA5 may exclusively compete with each other to regulate the stability of MDA5. Various types of ubiquitin chain modifications lead to its degradation through distinct pathways.

RNF144B is also named p53RFP (p53-inducible RING-finger protein) (Huang et al, 2006), IBRDC2 (in-between ring finger domain-containing protein 2) (Benard et al, 2010), and PIR2 (p73-Induced Ring Protein 2) (Zhou et al, 2018). The expression of RNF144B can be induced by p53 family proteins such as p53, p63, and p73. There is some inconsistency regarding its role in regulating cell proliferation and survival. It was first identified as

a p53-induced gene and induces p53-dependent apoptosis (Huang et al, 2006). RNF144B regulates p21 and p63 stability by promoting their degradation (Conforti et al, 2013). Upon DNA damage, RNF144B expression is induced by p73 and enhances the ubiquitin-mediated degradation of the p73-dominant negative isoform deltaNP73 (Sayan et al, 2010). As a result, the inhibitory effect of p73 on apoptosis after DNA damage is alleviated. On the other hand, in certain cancer cell lines, it functions as an oncogene, promoting cell proliferation (Zhou et al, 2018). The first evidence that RNF144B is closely related to innate immunity was its induction by LPS in human macrophages. Mechanistically, RNF144B is induced by LPS in a MyD88-dependent NF-κB activation and negatively regulates LPS-induced IFN production by interacting with TBK1 (Zhang et al, 2019). RNF144B overexpression reduces TBK1 phosphorylation through association with the TBK1 scaffold/dimerization domain (SDD) and inhibits its K63-linked polyubiquitination, independent of its E3 ligase activity (Zhang et al, 2019). However, the role of RNF144B in antiviral immunity has not been explored. In this study, we found that RNF144B expression was induced upon infection with EMCV (Fig. 1C); RNF144B impaired IRF3 activation and IFN-β production upon infection with EMCV (Figs. 1G and EV1H); *Rnf144b* deficiency or knockdown enhanced antiviral response of EMCV both in vitro and in vivo (Figs. 1I,J, EV4C,D and 6I), and RNF144B overexpression promoted EMCV replication (Fig. 1F). These results indicate that RNF144B is a negative regulator of IFN production upon EMCV infection. Our findings revealed that the MDA5 ligand (high molecular weight poly I:C) induced a high level of IFN signaling in *Rnf144b* KO cells, while the RIG-I ligand induced a lesser extent of signaling. Conversely, the IFN signaling induced by other ligands, such as TLR4, TLR7, and cGAS, showed no difference compared to that in wild-type cells (Fig. EV6A–E). In addition, RNF144B did not significantly inhibit the replication of DNA virus HSV (Fig. EV6G). These results indicated that RNF144B specifically regulated RNA virus induced IFN signaling. Furthermore, we found that knockout of RNF144B inhibited VSV replication (Fig. EV6F); however, it did not affect RIG-I degradation (Figs. EV2B and EV4G). This observation suggests that RNF144B may influence the functional activity of RIG-I without altering its protein level. Since RIG-I activity is known to be regulated by various post-translational modification, particularly ubiquitination (Chiang and Gack, 2017; Gack et al, 2007; Liu et al, 2016; Oshiumi et al, 2010; Sanchez et al, 2016), it is plausible that

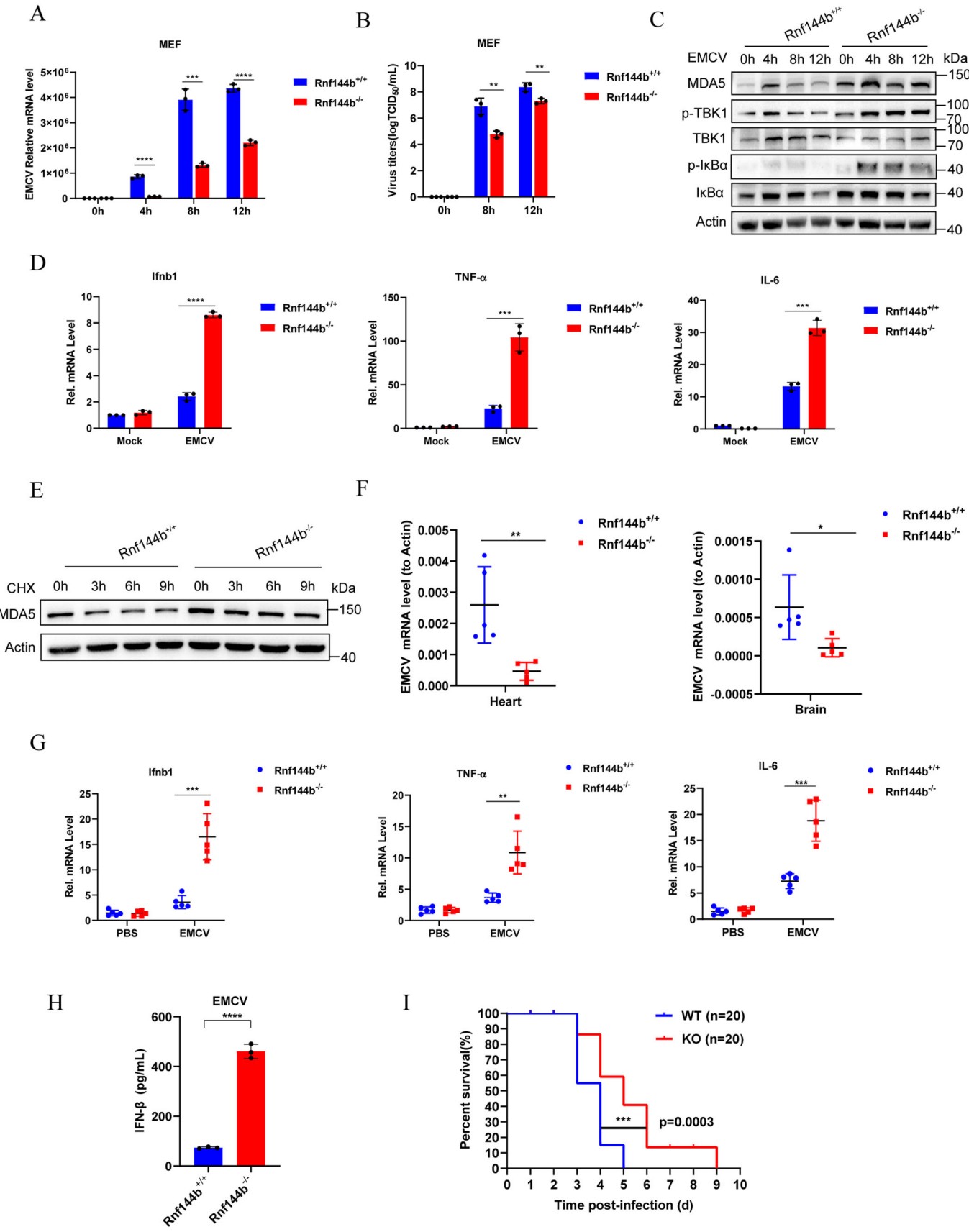

Figure 6.   RNF144B inhibits antiviral responses in vitro and in vivo.

(A) *Rnf144b⁺/⁺* and *Rnf144b⁻/⁻* MEFs were infected with EMCV (MOI = 1) for indicated time points, and then cell lysates were analyzed for EMCV replication level by qPCR. n = 3 biological replicates. Statistical significance was determined by two-tailed unpaired Student's t-test. ****P = 0.000039 (4 h), ***P = 0.00050 (8 h), ****P = 0.000040 (12 h). (B) *Rnf144b⁺/⁺* and *Rnf144b⁻/⁻* MEFs were infected with EMCV (MOI = 1) for indicated time points. The cell supernatant was harvested and analyzed by TCID₅₀ assay. n = 3 biological replicates. Statistical significance was determined by two-tailed unpaired Student's t-test. **P = 0.00540 (8 h), **P = 0.00820 (12 h). (C) Western blot analysis of the indicated signaling proteins in MEFs from *Rnf144b⁺/⁺* or *Rnf144b⁻/⁻* mice infected with EMCV (MOI = 1) for the indicated time periods. (D) *Rnf144b⁺/⁺* and *Rnf144b⁻/⁻* MEFs were infected with EMCV (MOI = 1). After 12 h, cell lysates were analyzed for Ifnb1, TNF-α, IL-6 mRNA level by qPCR. n = 3 biological replicates. Statistical significance was determined by two-tailed unpaired Student's t-test. ****P = 0.000009 (Ifnb1), ***P = 0.00090 (TNF-α), ***P = 0.00030 (IL-6). (E) *Rnf144b⁺/⁺* and *Rnf144b⁻/⁻* MEFs were infected with EMCV (MOI = 1). 12 h post-infection, cells were treated by CHX (30 μM) for indicated time points. Protein extracts were used for immunoblot analysis of the endogenous MDA5 protein level. (F, G) *Rnf144b⁺/⁺* and *Rnf144b⁻/⁻* mice were intraperitoneally injected EMCV (5 × 10⁶ PFU) for 48 h (n = 5 per group). qPCR analysis of EMCV replication level in heart and brain, and mRNA level of Ifnb1, TNF-α, IL-6 in brain (G). Statistical significance was determined by two-tailed unpaired Student's t-test. **P = 0.00530 (heart), *P = 0.02620 (brain), ***P = 0.00030 (Ifnb1), **P = 0.00170 (TNF-α), ***P = 0.00030 (IL-6). (H) ELISA of IFN-β production in serum of *Rnf144b⁺/⁺* and *Rnf144b⁻/⁻* mice that were intraperitoneally injected with EMCV (5 × 10⁶ PFU) for 12 h (n = 3 per group). Statistical significance was determined by two-tailed unpaired Student's t-test. ****P = 0.000020. (I) Survival of *Rnf144b⁺/⁺* and *Rnf144b⁻/⁻* mice (n = 20 per group) intraperitoneally infected with EMCV (6 × 10⁷ PFU). Significance was tested using log-rank (Mantel-Cox) test, ***P = 0.00030. Data information: Data shown are representative of at least three biological replicates, with each data point representing a biological experiment. Error bars are presented as mean ± SD. Statistical significance was determined by Student's t-test. Source data are available online for this figure.

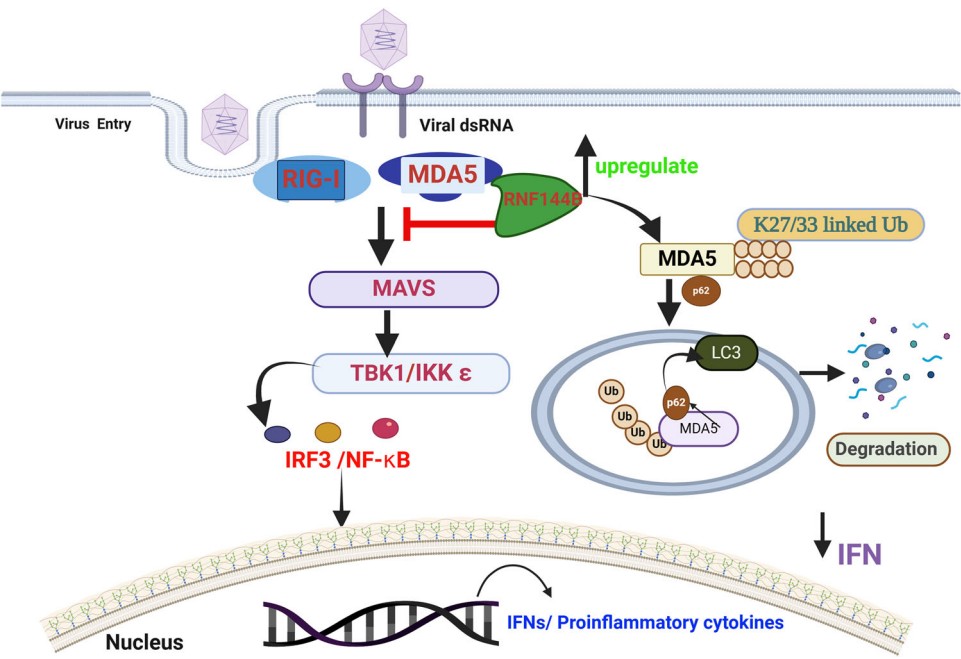

Figure 7.   Working model of RNF144B in negatively regulating antiviral immunity.

Upon viral infection, RNF144B expression is induced by viral infection and it can catalyze K27 and K33 polyubiquitinated of MDA5 at residues K23 and K43. Polyubiquitinated MDA5 is recognized by autophagic receptor p62 and delivered to autophagosomes for degradation.

RNF144B modulates RIG-I activity by influencing its polyubiquitination. Given that TBK1 is essential for IFN signaling induced by both VSV and HSV-1, yet RNF144B deficiency impacts VSV replication without affecting HSV-1 replication, it suggests that RNF144B may have additional targets. We are currently investigating this possibility in our lab. Previous studies have demonstrated that RNF144B negatively regulates LPS-induced TBK1 activation, independent of its E3 ubiquitin ligase activity. However, in our study, we found that RNF144B's regulation of EMCV replication and the interferon response is dependent on its E3 ubiquitin ligase activity. Specifically, in Rnf144b KO cells with MDA5 knockdown, the Rnf144b KO-induced increase in p-TBK1 levels was reversed (Fig. EV5E). This result suggested that the elevated

phosphorylation of TBK1 upon viral infection in Rnf144b KO cells was dependent on MDA5. These results indicate that RNF144B's regulation of MDA5, rather than TBK1, plays a key role in impairing EMCV-induced IFN signaling. Moreover, we found that Rnf144b deficiency slightly increased the expression of IFN and ISGs in the basal state, suggesting that it not only impairs ligand-induced IFN activation but also moderately suppresses tonic IFN activation. However, the equal susceptibility of Rnf144b KO cells and wild-type cells to HSV infection suggests that the impact on tonic signaling is minimal.

Autophagy is a quality control pathway in eukaryotic cells through which cells degrade their own components. The specificity of selective autophagic degradation depends on autophagic

adapters or receptors that deliver the cargo substrates and the lipidated protein LC3 to the autophagosome (Selective autophagy mediated by autophagic adapter proteins) (Shaid et al, 2013). Several cargo receptors have been identified, with p62 being the most widely studied. Mechanistically, the UBA domain of p62 recognizes and binds the polyubiquitin chain on the substrates, while its LIR motif associates with the LIR docking site (LDS) on LC3. It has been previously reported that free ISG15 interacts with RIG-I and acts as a signal for LRRC25/p62-mediated autophagic degradation of RIG-I to inhibit RIG-I-mediated signaling (Du et al, 2018). In this study, we found that p62 can specifically recognize the K27/K33-polyubiquitinated MDA5 and bridge it to the autophagosome for degradation (Fig. 5). Co-immunoprecipitation results showed that MDA5 also interacted with Tollip and CCDC50 (Fig. 5C). However, knockdown of Tollip by shRNA did not affect RNF144B promoted MDA5 degradation (Fig. 5G). CCDC50 knockdown partially restored RNF144B-mediated MDA5 degradation (Fig. 5I). These results indicated that CCDC50 also involved in the degradation of MDA5.

Based on these data, we propose a working model to illustrate how RNF144B negatively regulates RLR-mediated type I IFN production by facilitating the autophagic degradation of MDA5 (Fig. 7). Upon EMCV infection, the expression of RNF144B is induced. It interacts with MDA5 and catalyzes the K27 and K33 polyubiquitination of MDA5 at residue K23 and K43. As an autophagic adapter, p62 recognizes ubiquitinated MDA5 and delivers it to the autophagosome for degradation. Thus, RNF144B negatively regulates MDA5-triggered type I IFN induction pathway.

In summary, our results shed new light on the function of RNF144B in the regulation of MDA5-mediated type I IFN signaling. RNF144B catalyzes the K27/K33 ubiquitination of MDA5 and promotes its autophagic degradation through p62. The discovery of RNF144B as a negative regulator in the detection of RNA viruses provides insight into the mechanism of regulation of the cellular antiviral innate response.

# Methods

### Reagents and tools table

| Reagent/Resource | Reference or Source | Identifier or Catalog Number |
|---|---|---|
| **Experimental Models** | | |
| C57BL/6J (*M. musculus*) | Cyagen | KOCMP-218215-Rnf144b-B6J-VA |
| **Recombinant DNA** | | |
| pCMV-CCDC50(human)-3×HA | MIAOLING BIOLOGY | Cat #P35677 |
| pCMV-TagBFP-SQSTM1(human) | MIAOLING BIOLOGY | Cat #P48929 |
| **Antibodies** | | |
| Anti-Myc | Cell Signaling Technology | Cat# 2276 |
| Anti-p-TBK1 | Cell Signaling Technology | Cat# 5483 |
| Anti-TBK1 | Cell Signaling Technology | Cat# 3504 |

| Reagent/Resource | Reference or Source | Identifier or Catalog Number |
|---|---|---|
| Anti-p-IRF3 | Cell Signaling Technology | Cat# 4947 |
| Anti-IRF3 | Cell Signaling Technology | Cat# 11904 |
| Anti-IκBα | Cell Signaling Technology | Cat# 4814 |
| Anti-p-IκBα | Cell Signaling Technology | Cat# 2859 |
| Anti-mouse IgG, HRP-linked Antibody | Cell Signaling Technology | Cat# 7076 |
| Anti-rabbit IgG, HRP-linked Antibody | Cell Signaling Technology | Cat# 7074 |
| Goat Anti-Rabbit IgG H&L (Cy3 ®) | Abcam | Cat# ab6939 |
| Goat Anti-Mouse IgG H&L (FITC) | Abcam | Cat# ab6785 |
| Anti-RNF144B | ABSMART | Cat#MG470696M |
| Anti-GST | ABSMART | Cat# 324184 |
| Anti-β-Actin | Sigma-Aldrich | Cat# A5441 |
| Anti-Flag tag | Sigma-Aldrich | Cat# F1804 |
| Anti-MDA5 | Proteintech | Cat# 21775-1-AP |
| Anti-RNF144B | Proteintech | Cat# 26306-1-AP |
| Anti-p62 | Proteintech | Cat# 18420-1-AP |
| Anti-Tollip | Proteintech | Cat# 11315-1-AP |
| Anti-CCDC50 | Proteintech | Cat# 68369-1-Ig |
| **Oligonucleotides and other sequence-based reagents** | | |
| shRNA sequences or siRNA sequence | This study | Appendix Table S1 |
| RT-PCR primers | This study | Appendix Table S2 |
| PCR primers | This study | Appendix Table S3 |
| **Chemicals, Enzymes and other reagents** | | |
| 5X PrimeScript RT Master Mix | TAKARA | Cat#RR036A |
| TB Green® Premix Ex Taq™ II | TAKARA | Cat#RR820A |
| Z-VAD | MCE | HY-16658B |
| CQ | MCE | HY-17589A |
| MG132 | MCE | HY-13259 |
| Bafilomycin A1 | MCE | HY-100558 |
| 3-MA | MCE | HY-19312 |
| GM-CSF | Novoprotein | CK02 |
| M-CSF | Novoprotein | CB34 |
| poly(I:C) | Invivogen | tlrl-pic |
| Puromycin | Solarbio | P8230 |
| Pierce™ GST Protein Interaction Pull-Down Kit | ThemoFisher | Cat#21516 |
| Mut Express II Fast Mutagenesis Kit V2 | Vazyme | Cat# C214 |
| Total RNA Extraction Kit | Omega | Cat#R6834-02 |
| EndoFree Maxi Plasmid Kit | TIANGEN | Cat#DP117 |
| TIANamp Genomic DNA Kit | TIANGEN | Cat#DP304 |

| Reagent/Resource | Reference or Source | Identifier or Catalog Number |
|---|---|---|
| Mouse Interferon β (IFN-β/IFNB) ELISA Kit | CUSABIO | Cat#E04945m |
| **Tools** | | |
| GraphPad Prism 8.0.2 | https://www.graphpad.com/ | |
| ImageJ_v1.8.0 | https://imagej.net/ij/ | |

## Reagents and viruses

Poly(I:C) was purchased from Invivogen and used at a final concentration of 1 μg/mL. Puromycin was purchased from Solarbio and used at a final concentration of 2 μg/mL. The collagenase type I was purchased from Merck. 3-MA, MG132, Z-VAD, bafilomycin A1 and chloroquine were purchased from MCE. GM-CSF and M-CSF was purchased from Novoprotein. Protein A/G Magnetic Beads was purchased from MCE. The EMCV was kindly provided by Prof. Zhiyong Li (Wenzhou Medical university, School of Basic Medical Sciences).

## Biosafety

All experiments with EMCV, VSV, and HSV were performed in biosafety laboratory level 2 (BSL-2) and approved by Lanzhou Veterinary Research Institute (LVRI), Chinese Academy of Agricultural Sciences (CAAS).

## Cell culture

The HEK293T, HeLa, and A549 cell lines were cultured in Dulbecco's modified Eagle's medium (DMEM). THP-1 cells were cultured in RMPI-1640 medium. The media were supplied with 10% FBS and 1/100 p/s. Mouse peritoneal macrophages were obtained from female C57BL/6J mice (6–8 weeks old) that were injected intraperitoneally with 3% thioglycolate broth. Three days later, mouse peritoneal macrophages were harvested and incubated for 2 h to remove un-adhered cells. BMDMs and BMDCs were isolated from the femurs of wild-type (WT) and $Rnf144b^{-/-}$ mice. GM-CSF (20 ng/ml, Peprotech) and M-CSF (10 ng/ml, Peprotech) were added to the culture medium to induce differentiation of BMDCs and BMDMs. Primary MEFs were prepared from E14.5 embryos derived from WT and $Rnf144b^{-/-}$ mice. All cells were cultured in a 37 °C incubator with 5% $CO_2$ in their respective culture media containing 10% FBS, 100 U/ml penicillin, and 100 μg/ml streptomycin.

## Mice

CRISPR/Cas9-mediated genome editing was used to generate $Rnf144b^{-/-}$ mice (Cyagen Biosciences, China). In brief, purified Cas9 protein purchased from NEB and guide RNAs (5′-GAATACTTTCAATCGGGAATTGG-3′; 5′-GCATCGAATACTTTATCACGAGG-3′) were injected into fertilized eggs. Subsequently, the eggs were transplanted into pseudo-pregnant mice. The genome of F0 mice was amplified with PCR primers (F:5′-AGAGCCAGATTTTGGATTCTGC-3′; R:5′-TTCTTTCTGACTAGGGCTGG

TACT-3′) and sequenced. The chimeras were crossed with wild-type C57BL/6 mice to obtain $Rnf144b^{+/-}$ mice. The F1 $Rnf144b^{+/-}$ mice were crossed with wild-type C57BL/6 mice for a minimum of three generations. Mice were genotyped by PCR analysis followed by sequencing, and the resulting $Rnf144b^{+/-}$ mice were crossed to generate $Rnf144b^{+/+}$ and $Rnf144b^{-/-}$ mice. Age-matched and sex-matched $Rnf144b^{+/+}$ and $Rnf144b^{-/-}$ littermates were randomly divided into groups for in vivo studies. All mice were housed in the specific pathogen-free animal facility at Lanzhou Veterinary Research Institute, and all animal experiments were conducted in accordance with protocols approved by the Animal Ethics Committee of Lanzhou Veterinary Research Institute. All animals were strictly handled in accordance with the Good Animal Practice of the People's Republic of China Animal Ethics Procedures and Guidelines. All mouse studies have been approved by the Animal Ethics Committee of Lanzhou Institute of Veterinary Medicine, Chinese Academy of Agricultural Sciences (Permit No. LVRIAEC-2023-018).

## Transfection

Transfection of HEK293T and THP1 cells was performed with Lipofectamine 3000 (Invitrogen) according to the manufacturer's instructions, Firstly, Plate cells so they will be 70–90% confluent at the time of transfection, then. Prepare plasmid DNA-lipid complexes. Finally, Add DNA-lipid complexes to cells.

## Lentivirus-mediated gene transfer and gene knockdown

HEK293T cells were transfected with pLKO.1-puro-shRNA or Scramble along with the packaging vectors psPAX2 and pMD2G using Lipofectamine 3000. The supernatants containing lentivirus were collected at 48 and 72 h after transfection and passed through a 0.45 μm filter. HEK293T or HeLa cells were incubated with lentivirus for 24 h. After incubation, the media were changed. The cells were selected with puromycin (2 μg/ml) for 7 days. The final positive cells are knock-down cell lines. The shRNA sequences or siRNA sequence are listed in Appendix Table S1.

## Protein purification and GST-pull down experiment

The GST and GST-RNF144B plasmids were transformed into BL21 competent cells and induced with IPTG (1 mM) at 37 °C for 8 h. The competent cells were lysed with lysis buffer (0.15 M NaCl, 0.0027 M KCl, 0.01 M $Na_2HPO_4$, 0.0018 M $KH_2PO_4$). Then, the proteins were purified by affinity chromatography using glutathione-Sepharose matrix, followed by glutathione (0.05 M Tris-HCl, 0.01 M GSH) elution and dialysis. Purified GST or GST-RNF144B and MDA5-Flag proteins were incubated with glutathione agarose overnight at 4 °C. The glutathione agarose was washed three times with PBS and subjected to immunoblotting analysis.

## qRT-PCR

Total RNA was extracted from the cultured cells or tissues with TRIZOL reagent (Invitrogen) according to the manufacturer's instructions, and the first-strand cDNA was reverse-transcribed with 5× PrimeScript RT Master Mix (TakaRa). Quantitative PCR

was performed using the TB Green® Premix Ex Taq™ kit (Takara), and the primer sequences are listed in Appendix Table S2.

## ELISA

The IFN-β in the sera or cell supernatants was determined by the indicated ELISA kits (CUSABIO). The detailed methods include the following: (1) Prepare all reagents, working standards, and samples as instructed. (2) Add 100 μL of standard and sample to each well. Incubate for 2 h at 37 °C. (3) Remove the liquid of each well, don't wash. (4) Add 100 μL of Biotin-antibody (1x) to each well. Incubate for 1 h at 37 °C. (5) Aspirate and wash 3 times. (6) Add 100 μL of HRP-avidin (1x) to each well. Incubate for 1 h at 37 °C. (7) Repeat the aspiration/wash process for five times. (8) Add 90 μL of TMB Substrate to each well. Incubate for 15–30 min at 37 °C (Protect from light). (9) Add 50 μL of Stop Solution to each well, Read at 450 nm within 5 min.

## Confocal microscopy

After virus infection or transfection, cells were fixed with 4% paraformaldehyde, permeabilized with 0.5% Triton X-100, and then blocked with 1% BSA in PBS. The cells were incubated with 1% BSA-containing primary antibodies overnight, followed by three PBS washes. Then, cells were stained with Alexa Fluor® 488- or 594-conjugated Goat Anti-rabbit/mouse IgG secondary antibodies and counterstained with DAPI (blue).

## Co-immunoprecipition and immunoblot assays

The RNF144B, single-point mutations in RNF144B (C-A), autophagic receptors (p62, BNIP3L, Tollip, NBR1) genes were subcloned into the pCMV-Myc or pCMV-HA expression vector. The PCR amplified sequences are listed in Appendix Table S3. Then, the plasmids were transformed into *Escherichia coli* JM109 cells and extracted using plasmid kit according to the manufacturer's instructions. The plasmids were transfected into 293T cells. After cell transfection, cells were lysed using IP buffer containing protease and phosphatase inhibitors (20 mM Tris, pH 7.5, 150 mM NaCl, 1% Triton X-100) and centrifuged at 12,000 rpm/min for 15 min. The supernatant was collected and incubated with specific antibodies or control IgG at 4 °C overnight. Protein G magnetic beads, pre-washed three times with IP buffer, were added to the supernatants and incubated for 2–4 h. The immunoprecipitates were washed three times with IP buffer, eluted with 1% SDS sample buffer, and analyzed by Western blot.

For immunoblot assays, cells were lysed using RIPA buffer (50 mM Tris, pH 7.4, 150 mM NaCl, 1% Triton X-100, 1% sodium deoxycholate, 0.1% SDS). Proteins were extracted, and protein concentrations were determined. The extracts were separated by SDS-PAGE and transferred to PVDF membranes for immunoblot analysis.

The following antibodies were used for immunoblot analysis or immunoprecipitation: anti-HA tag (Cat#3724, Cell Signaling Technology), anti-Myc (Cat#2276, Cell Signaling Technology), anti-p-TBK1 (Cat#5483, Cell Signaling Technology), anti-TBK1 (Cat#3504, Cell Signaling Technology), anti-p-IRF3 (Cat#4947, Cell Signaling Technology), anti-IRF3 (Cat#11904, Cell Signaling

Technology), anti-IκBα (Cat#4814, Cell Signaling Technology), anti-p-IκBα (Cat#2859, Cell Signaling Technology), anti-mouse IgG, HRP-linked Antibody (Cat#7076, Cell Signaling Technology), anti-rabbit IgG, HRP-linked Antibody (Cat#7074, Cell Signaling Technology), anti-RNF144B (Cat#MG470696M, ABSMART), anti-GST (Cat#324184, ABSMART), anti-β-Actin (Cat#A5441, Sigma-Aldrich), anti-Flag tag (Cat#F1804, Sigma-Aldrich), anti-MDA5 (Cat#21775-1-AP, Proteintech), anti-RNF144B (Cat#26306-1-AP, Proteintech), anti-p62 (Cat#18420-1-AP, Proteintech), anti-Tollip (Cat#11315-1-AP, Proteintech), anti-CCDC50 (Cat#68369-1-Ig, Proteintech).

## Ubiquitination assays

For denaturing IP, cells were lysed in IP buffer containing 1% SDS and heated at 95 °C for 5 min. A total of 1/10 volume of the cell lysates was saved as input for immunoblot analysis to detect the expression of target proteins. The rest of the cell lysates were diluted with 1–2 mL lysis buffer and immunoprecipitated (Denature-IP) with specific antibodies. The following steps were proceeded as regular IP.

## Plaque assay

HEK293T cells were plated on six-well plates. Serial 10-fold dilutions of virus stocks (200 μL/well) were inoculated onto the surface of cells for 1 h of incubation. Then, 2 mL of overlay medium containing 0.6% gum tragacanth (MP Biomedicals) and 1% FBS were added, and the cells were routinely incubated at 37 °C in 5% $CO_2$. After infection at 48 or 72 h, the cultured cells were fixed with cold acetone/methanol (1:1, V/V) and stained with 0.2% crystal violet (Sigma). The titer of each virus was evaluated as PFU/mL (plaque forming units, PFU) in at least duplicate per experiment, by counting the number of plaques formed on cell culture monolayers.

## $TCID_{50}$

For $TCID_{50}$ experiments, HEK293T cells were inoculated in 96-well plates. After 24 h, 0.1 mL of the culture supernatant was serially diluted and added dropwise to HEK293T cells, and $TCID_{50}$ was measured 72 h later.

## Viral infection of mice

For in vivo studies, age-matched and sex-matched $Rnf144b^{+/+}$ and $Rnf144b^{-/-}$ mice (female, 8 weeks old) were injected intraperitoneally with EMCV ($6 \times 10^7$ PFU per mouse), and the survival of animals was monitored every day. In non-lethal dose experiments, the serum was collected for ELISA measurement of IFN-β levels after EMCV infection for 12 h. The hearts and brains were collected for qRT-PCR analysis at 2 days after infection.

## Statistics

The data are presented as the means ± SD from at least three independent experiments. The statistical significance between different groups was determined using the unpaired Student t-test

or one-way ANOVA, with $p < 0.05$ considered statistically significant. The survival curve of mice was analyzed with the log-rank (Mantel-Cox) test, with a $p$-value $< 0.05$ considered significant.

## Data availability

Accession codes for RNA-Sequencing: BioProject PRJNA716767 (https://www.ncbi.nlm.nih.gov/bioproject/?term=716767; SRA: SRR14089915, SRR14089916, SRR14089917, SRR14089918, SRR14089919, SRR14089920, SRR14089921, SRR14089922, SRR14089923), PRJNA1109448 (https://www.ncbi.nlm.nih.gov/bioproject/?term=1109448; SRA:SRR28975519, SRR28975520, SRR28975521, SRR28975522, SRR28975523, SRR28975524, SRR28975525, SRR28975526, SRR28975527, SRR28975528, SRR28975529, SRR28975530).

The source data of this paper are collected in the following database record: biostudies:S-SCDT-10_1038-S44319-024-00256-w.

## Peer review information

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

## Acknowledgements

We thank Prof. Zhiyong Li (Wenzhou Medical University, School of Basic Medical Sciences) for providing the EMCV. We thank Prof. Qiyun Zhu (Chinese Academy of Agricultural Science, Lanzhou Veterinary Research Institute) for gifting the ubiquitin mutant plasmids. We thank the staff at the Instrument Center, Lanzhou Veterinary Research Institute, and Chinese Academy of Agricultural Science for advice and assistance in Confocal laser scanning microscopy sample observation and data collection. This research was financially supported by National Natural Science Foundation of China (32400586, 32372988, 31972687, 32270763), the Agricultural Science and Technology Innovation Program of CAAS (CAAS-ASTIP-2021-LVRI), National Key R&D Program of China (2021YFD1800300) and the Science and Technology Major Project of Gansu Province (22ZD6NA001).

## Author contributions

**Guoxiu Li**: Data curation; Formal analysis; Investigation; Methodology; Writing—original draft. **Jing Zhang**: Conceptualization; Supervision; Investigation; Methodology; Writing—original draft; Writing—review and editing. **ZhiXun Zhao**: Funding acquisition. **Jian Wang**: Software; Methodology. **Jiaoyang Li**: Software; Methodology. **Weihong Xu**: Investigation; Methodology. **Zhanding Cui**: Software; Methodology. **Pu Sun**: Resources. **Hong Yuan**: Resources; Software; Formal analysis. **Tao Wang**: Data curation; Software; Formal analysis. **Kun Li**: Software; Methodology. **Xingwen Bai**: Resources; Validation. **Xueqing Ma**: Methodology. **Pinghua Li**: Resources. **Yuanfang Fu**: Visualization. **Yimei Cao**: Data curation; Software; Methodology. **Huifang Bao**: Resources. **Dong Li**: Software. **Zaixin Liu**: Resources; Funding acquisition. **Ning Zhu**: Resources; Writing—review and editing. **Lijie Tang**: Writing—review and editing. **Zengjun Lu**: Funding acquisition; Project administration; Writing—review and editing.

Source data underlying figure panels in this paper may have individual authorship assigned. Where available, figure panel/source data authorship is listed in the following database record: biostudies:S-SCDT-10_1038-S44319-024-00256-w.

## Disclosure and competing interests statement

The authors declare no competing interests.1

# Expanded View Figures

**Figure EV1. RNF144B suppresses RNA-induced innate immune responses.**

(A) Venn diagram of common DEGs from RNA-Seq data of host cells infected with porcine reproductive and respiratory syndrome virus, foot-and-mouth disease virus, and Seneca virus A. (B–D) Validation of knockdown efficiency of siRNAs targeting FJX1, ZC3H12C, and RNF144B in HEK293T cells. $n = 3$ biological replicates. Statistical significance was determined by two-tailed unpaired Student's t-test. "ns" indicates no significant difference. **$P = 0.0017$ (siFJX1-1088), ***$P = 0.0002$ (siFJX1-1229), ****$P = 0.00001$ (siZC3H12C-186), ***$P = 0.0004$ (siZC3H12C-1564), **$P = 0.009$ (siZC3H12C-2254), ***$P = 0.0002$ (siRNF144B-532), ***$P = 0.0005$ (siRNF144B-976), ***$P = 0.0008$ (siRNF144B-1112). (E, F) HEK293T cells were transfected with *NC* or *siRNA* targeting FJX1, ZC3H12C, and RNF144B for 36 h. Cells were infected with VSV (MOI = 0.1) for indicated time points and subjected to qPCR analysis for VSV mRNA level. $n = 3$ biological replicates. Statistical significance was determined by two-tailed unpaired Student's t-test. "ns" indicates no significant difference. ***$P = 0.00010$ (F). (G) HEK293T cells were transfected with NC or *siRNA* targeting RNF144B for 36 h. Cells were infected with EMCV (MOI = 1) for 12 h and subjected to qPCR analysis for EMCV mRNA level. $n = 3$ biological replicates. Statistical significance was determined by two-tailed unpaired Student's t-test. *$P = 0.02080$. (H) HEK293T cells were transfected with empty vector or RNF144B-Myc for 18 h. Cells were infected with EMCV (MOI = 1) for 12 h and subjected to qPCR analysis for EMCV mRNA level. $n = 3$ biological replicates. Statistical significance was determined by two-tailed unpaired Student's t-test. **$P = 0.00110$. (I) Immunoblot analysis of RNF144B expression in HEK293T cells stably expressing *shRNA* against RNF144B. (J–M) Control or RNF144B KD HEK293T cells were transfected with poly (I:C) for 12 h and subjected to qPCR analysis for IFNB1 and ISG15 mRNA level. $n = 3$ biological replicates. Statistical significance was determined by two-tailed unpaired Student's t-test. ***$P = 0.0005$ (J), **$P = 0.0065$ (K), **$P = 0.0057$ (L), **$P = 0.0075$ (M). (N) HEK293T cells were transfected with poly(I:C) for 12 h and subjected to qPCR analysis of RNF144B mRNA level. $n = 3$ biological replicates. Statistical significance was determined by two-tailed unpaired Student's t-test. **$P = 0.0049$. Data information: Data shown are representative of at least three biological replicates, with each data point representing a biological experiment. Error bars are presented as mean ± SD. Statistical significance was determined by Student's t-test. Source data are available online for this figure.

▶

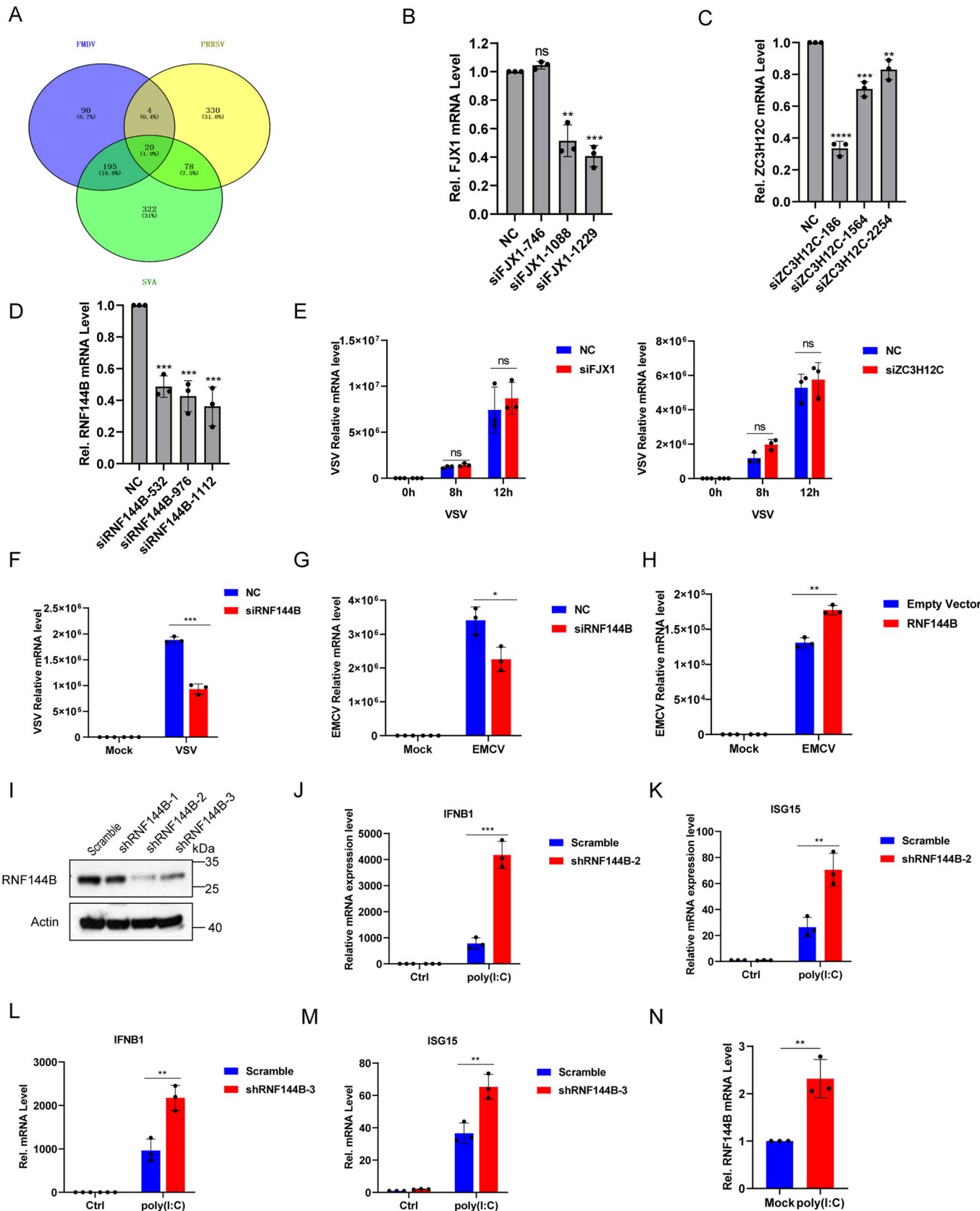

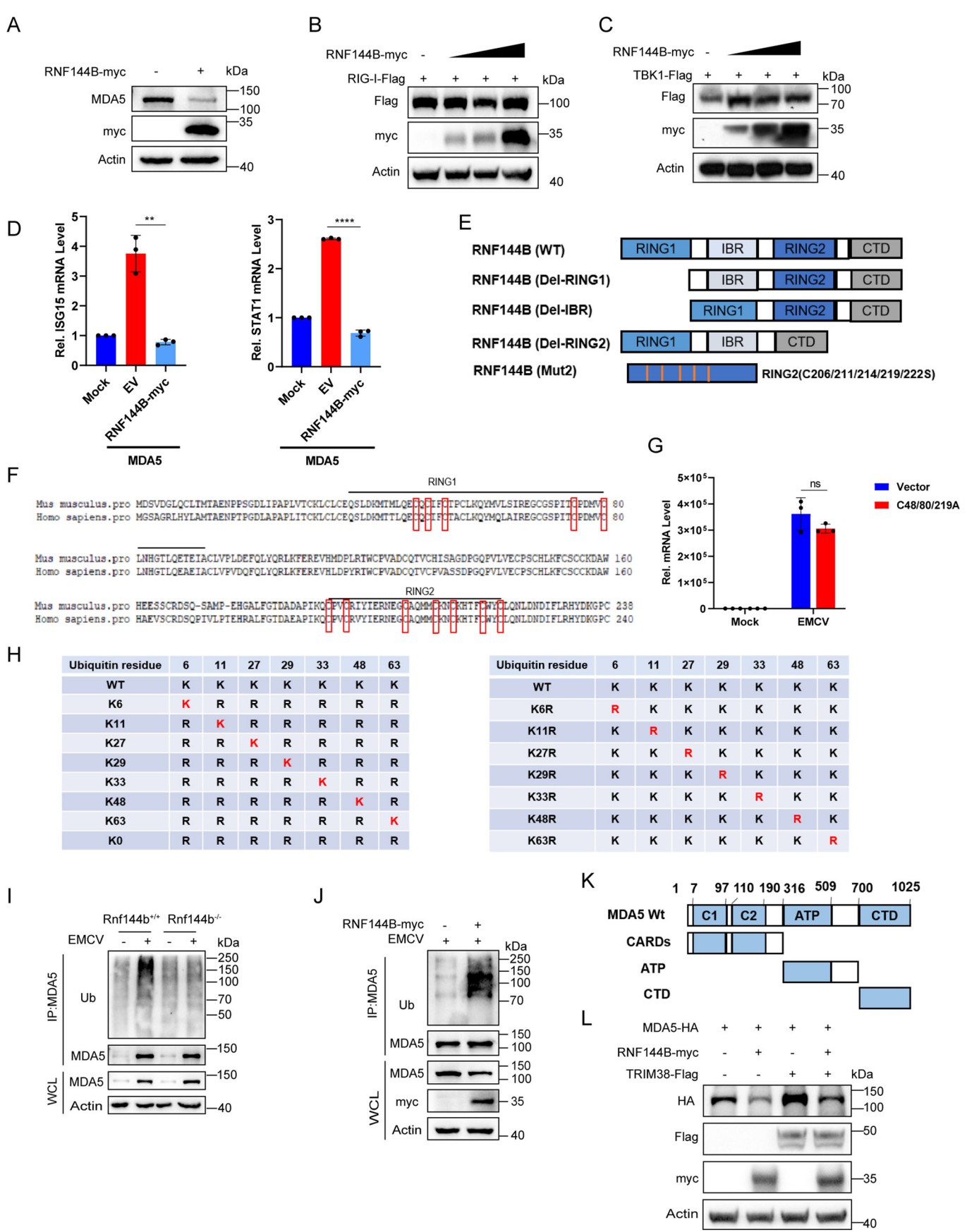

**Figure EV2. RNF144B targets MDA5.**

(A) A549 cells were transfected with plasmid encoding RNF144B. 18 h post-transfection, protein extracts were used for immunoblot analysis of the endogenous MDA5 protein level. (B) HEK293T cells were co-transfected with empty vector or RNF144B-Myc (2 µg) and RIG-I-Flag. 18 h post-transfection, cell lysates were analyzed by immunoblot using the indicated antibodies. (C) HEK293T cells were co-transfected with empty vector or RNF144B-Myc (2 µg) and TBK1-Flag. 18 h post-transfection, cell lysates were analyzed by immunoblot using the indicated antibodies. (D) HEK293T cells were transfected with plasmids encoding RNF144B-myc and MDA5-Flag. 18 h post-transfection, cells were subjected to qPCR analysis for ISG15 and STAT1 mRNA level. $n = 3$ biological replicates. Statistical significance was determined by two-tailed unpaired Student's t-test. **$P = 0.0011$, ****$P = 0.000001$. (E) The structure of RNF144B and its mutants. (F) The amino acid sequence of RING1 and RING2 domains of RNF144B. (G) HEK293T cells were transfected with empty vector or C48/80/219A-Myc plasmid. After 18 h, cells were infected with EMCV (MOI = 1) for 12 h. and subjected to qPCR analysis for EMCV mRNA level. $n = 3$ biological replicates. Statistical significance was determined by two-tailed unpaired Student's t-test. "ns" indicates no significant difference. (H) The model of mutant plasmids of Ubiquitin. (I) $Rnf144b^{+/+}$ and $Rnf144b^{-/-}$ MEFs infected with EMCV (MOI = 1) for 12 h. Cell lysates were harvested and immunoprecipitated using anti-MDA5 antibody, followed by immunoblots using the indicated antibodies. (J) A549 cells were transfected with RNF144B-myc. After 18 h, cells were infected with EMCV (MOI = 1) for indicated time points. Cell lysates were subjected to immunoprecipitation with anti-MDA5 antibody followed by immunoblot analysis with indicated antibodies. (K) The structure of MDA5 and its mutants. (L) HEK293T cells were co-transfected with empty vector or RNF144B-Myc, TRIM38-Flag, and MDA5-HA. 18 h post-transfection, cell lysates were analyzed by immunoblot using the indicated antibodies. Data information: Data shown are representative of at least three biological replicates, with each data point representing a biological experiment. Error bars are presented as mean ± SD. Statistical significance was determined by Student's t-test. Source data are available online for this figure.

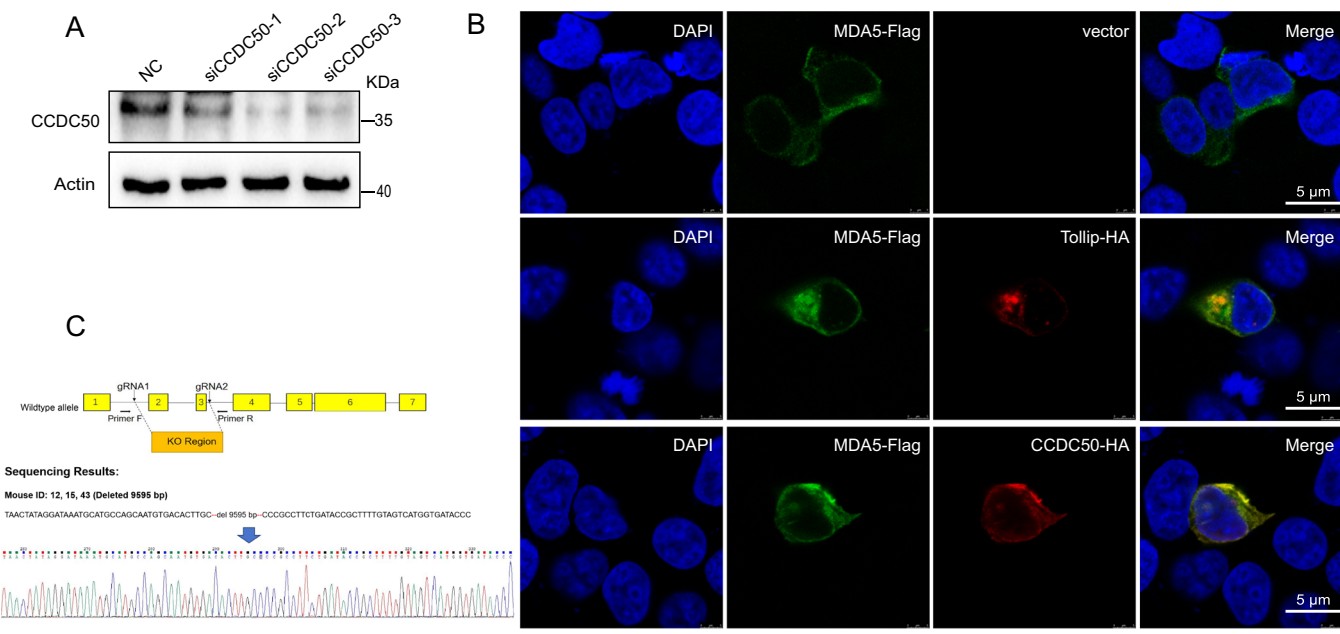

Figure EV3.  Rnf144b knockout does not affect immune cell development.

(A) Immunoblot analysis of CCDC50 expression in HEK293T cells expressing *siRNA* against CCDC50. (B) Confocal microscopy analysis of A549 cells co-transfected with MDA5-Flag (green) and Tollip-HA or CCDC50-HA (red). Scale bar, 5 μm. (C) Construct strategy diagram of *Rnf144b* gene knockout mouse. (D) Verification of knockout efficiency of *Rnf144b* through PCR analysis of tail genome DNA and by immunoblot analysis conducted on heart of WT and KO mice. (E) Flow cytometry analysis of percentage of the T cells, B cells, DCs and macrophages in splenocytes of WT and KO mice (6–8 weeks, female). (F) The percentage of dendritic cells (DCs), macrophages, CD4, CD8, CD19 positive cells in splenocytes from $Rnf144b^{+/+}$ and $Rnf144b^{-/-}$ mice were analyzed by flow cytometry. $n = 3$ biological replicates. Statistical significance was determined by two-tailed unpaired Student's t-test. "ns" indicates no significant difference. Data information: Data shown are representative of at least three biological replicates, with each data point representing a biological experiment. Error bars are presented as mean ± SD. Statistical significance was determined by Student's t-test. Source data are available online for this figure.

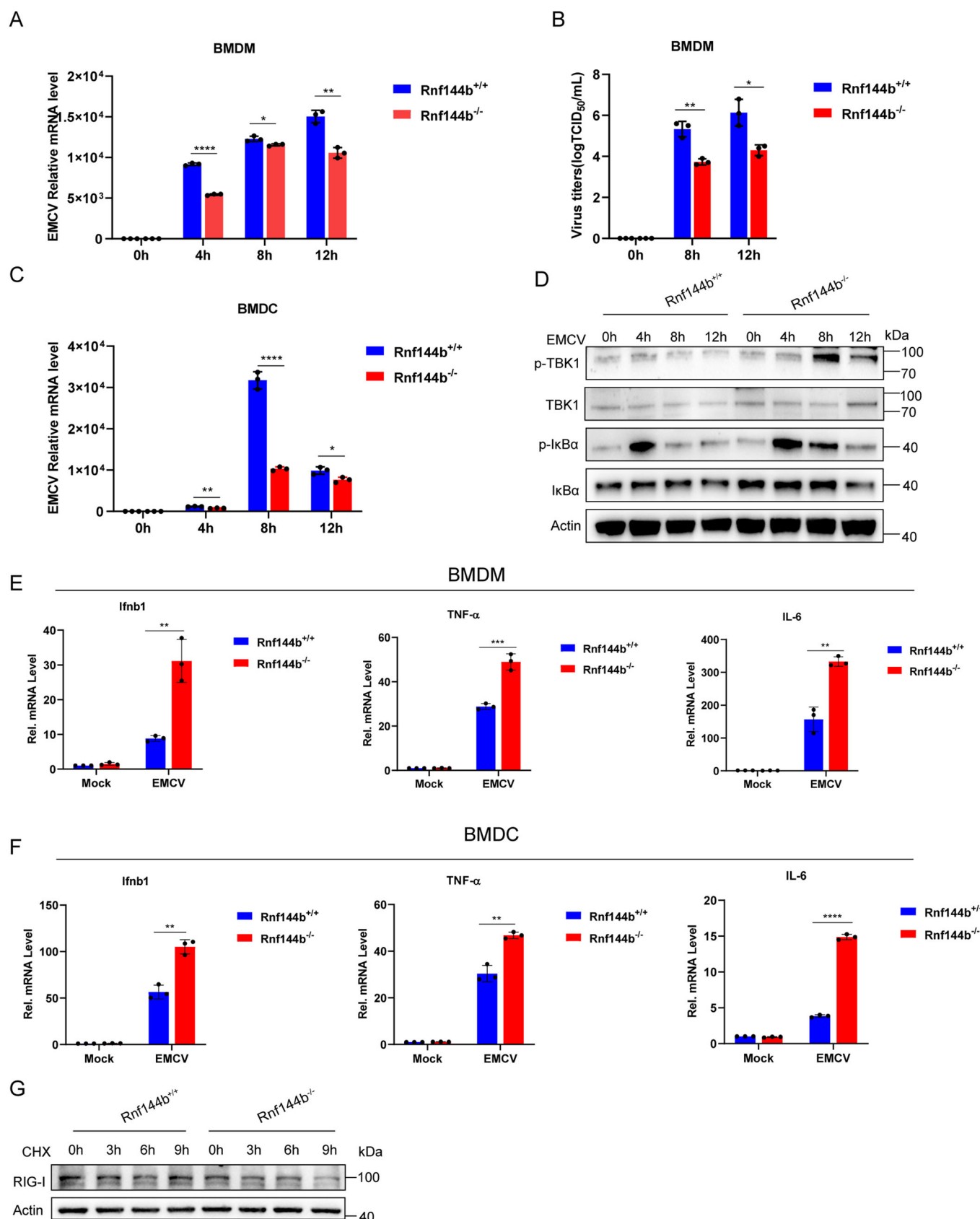

◀ **Figure EV4. Rnf144b deficiency enhances anti-RNA viral responses.**

(A) *Rnf144b⁺/⁺* and *Rnf144b⁻/⁻* BMDMs were infected with EMCV (MOI = 1), and then cell lysates were analyzed for EMCV replication level by qPCR. $n = 3$ biological replicates. Statistical significance was determined by two-tailed unpaired Student's t-test. ****$P = 0.000004$, *$P = 0.02040$, **$P = 0.00150$. (B) *Rnf144b⁺/⁺* and *Rnf144b⁻/⁻* BMDMs were infected with EMCV (MOI = 1) for indicated time points. The cell supernatant was harvested and analyzed by $TCID_{50}$ assay. n = 3 biological replicates. Statistical significance was determined by two-tailed unpaired Student's t-test. **$P = 0.00250$, *$P = 0.01070$. (C) *Rnf144b⁺/⁺* and *Rnf144b⁻/⁻* BMDCs were infected with EMCV (MOI = 1), and then cell lysates were analyzed for EMCV replication level by qPCR. $n = 3$ biological replicates. Statistical significance was determined by two-tailed unpaired Student's t-test. **$P = 0.00270$, ****$P = 0.000064$, *$P = 0.02330$. (D) Western blot analysis of the indicated signaling proteins in BMDMs from *Rnf144b⁺/⁺* or *Rnf144b⁻/⁻* mice infected with EMCV (MOI = 1) for the indicated time periods. (E) *Rnf144b⁺/⁺* and *Rnf144b⁻/⁻* BMDMs were infected with EMCV (MOI = 1). After 12 h, cell lysates were analyzed for Ifnb1, TNF-α, IL-6 mRNA level by qPCR. $n = 3$ biological replicates. Statistical significance was determined by two-tailed unpaired Student's t-test. **$P = 0.00340$ (Ifnb1), ***$P = 0.00080$ (TNF-α), **$P = 0.00160$ (IL-6). (F) *Rnf144b⁺/⁺* and *Rnf144b⁻/⁻* BMDCs were infected with EMCV (MOI = 1). After 12 h, cell lysates were analyzed for Ifnb1, TNF-α, IL-6 mRNA level by qPCR. $n = 3$ biological replicates. Statistical significance was determined by two-tailed unpaired Student's t-test. **$P = 0.00140$ (Ifnb1), **$P = 0.00170$ (TNF-α), ****$P = 0.000001$ (IL-6). (G) *Rnf144b⁺/⁺* MEFs and *Rnf144b⁻/⁻* MEFs were infected with EMCV (MOI = 1). 12 h post-infection, cells were treated by CHX (30 μM) for indicated time points. Protein extracts were used for immunoblot analysis of the endogenous RIG-I protein level. Data information: Data shown are representative of at least three biological replicates, with each data point representing a biological experiment. Error bars are presented as mean ± SD. Statistical significance was determined by Student's t-test. Source data are available online for this figure.

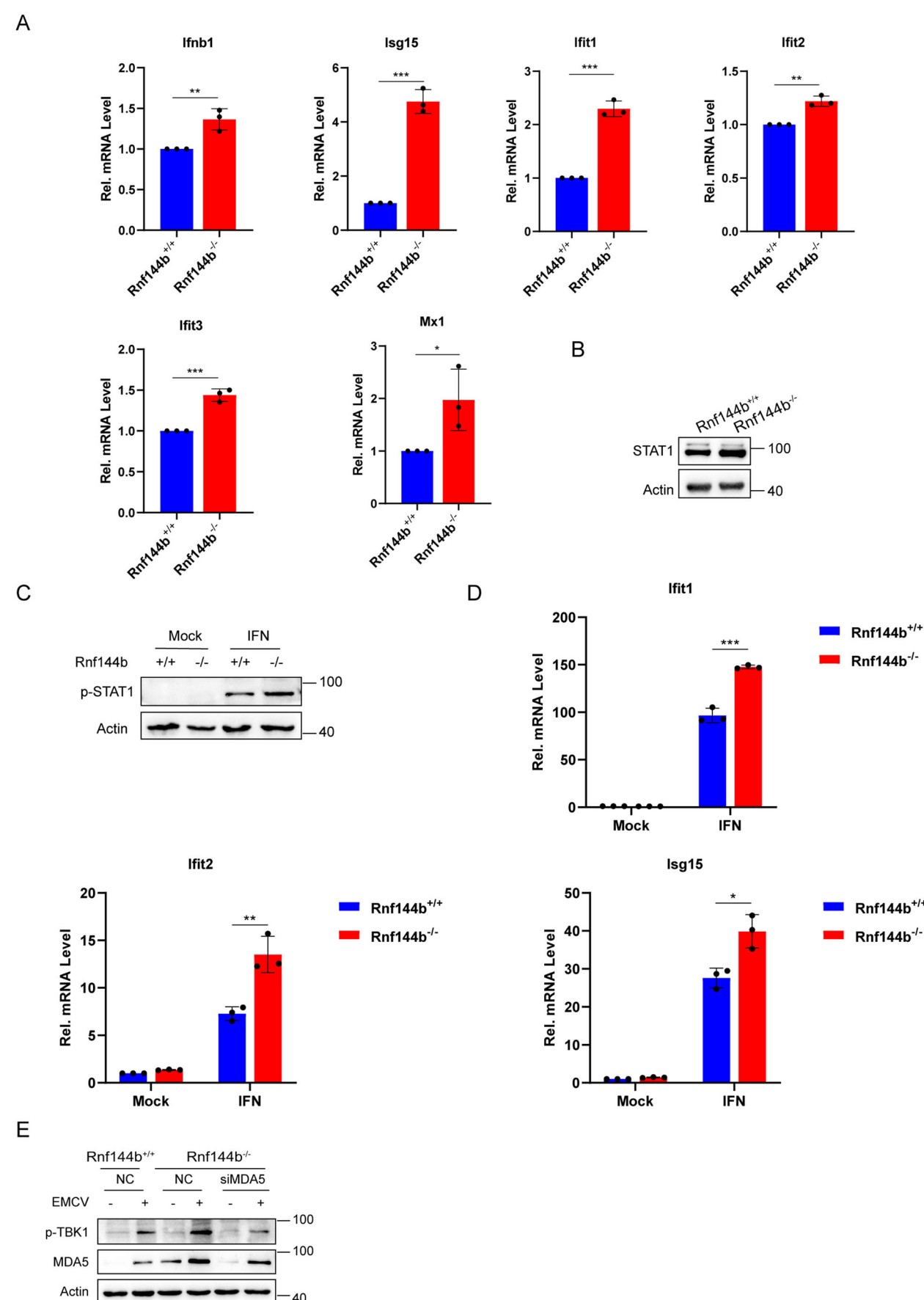

◀ **Figure EV5.  RNF144B moderately regulates tonic IFN signaling.**

(A) *Rnf144b*$^{+/+}$ and *Rnf144b*$^{-/-}$ MEFs cell lysates were analyzed for ISGs mRNA level by qPCR. $n = 3$ biological replicates. Statistical significance was determined by two-tailed unpaired Student's t-test. **$P = 0.0088$ (Ifnb1), ***$P = 0.0001$ (Isg15), ***$P = 0.00010$ (Ifit1), **$P = 0.0015$ (Ifit2), ***$P = 0.0006$ (Ifit3), *$P = 0.0445$ (Mx1). (B) *Rnf144b*$^{+/+}$ and *Rnf144b*$^{-/-}$ MEFs cell lysates were analyzed for STAT1 protein level by Western blot. (C) Western blot analysis of the indicated signaling proteins in MEFs from *Rnf144b*$^{+/+}$ or *Rnf144b*$^{-/-}$ mice were treated with IFN for 2 h. (D) qPCR analysis of ISGs mRNA level in MEFs from *Rnf144b*$^{+/+}$ or *Rnf144b*$^{-/-}$ mice were treated with IFN for the indicated time periods. $n = 3$ biological replicates. Statistical significance was determined by two-tailed unpaired Student's t-test. ***$P = 0.00040$ (Ifit1), **$P = 0.00610$ (Ifit2), *$P = 0.01410$ (Isg15). (E) *siRNAs* targeting MDA5 or NC were transfected into *Rnf144b*$^{+/+}$ or *Rnf144b*$^{-/-}$ MEFs cells and infected with EMCV (MOI = 1) for 12 h. Cell lysates were used for immunoblot analysis with the indicated antibodies. Data information: Data shown are representative of at least three biological replicates, with each data point representing a biological experiment. Error bars are presented as mean ± SD. Statistical significance was determined by Student's t-test. Source data are available online for this figure.

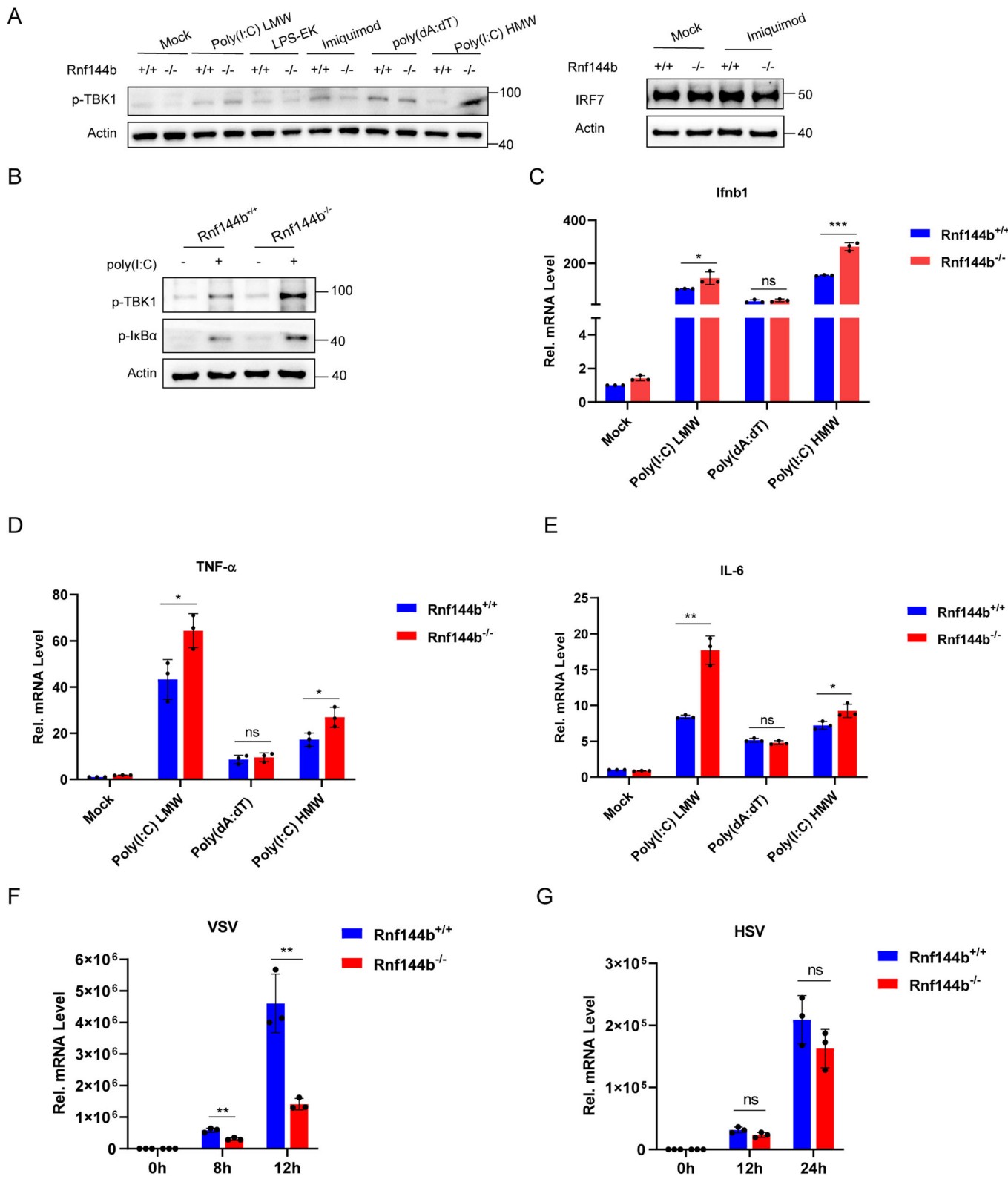

◀ **Figure EV6.  RNF144B negatively regulates EMCV induced IFN production and antiviral innate immunity.**

(A) Western blot analysis of the indicated signaling proteins in MEFs from *Rnf144b*$^{+/+}$ or *Rnf144b*$^{-/-}$ mice were transfected with different ligands for 12 h. (B) Western blot analysis of the indicated signaling proteins in MEFs from *Rnf144b*$^{+/+}$ or *Rnf144b*$^{-/-}$ mice were transfected with poly (I:C) for 12 h. (C) qPCR analysis of Ifnb1 mRNA level in MEFs from *Rnf144b*$^{+/+}$ or *Rnf144b*$^{-/-}$ mice were transfected with different ligands for 12 h. $n = 3$ biological replicates. Statistical significance was determined by two-tailed unpaired Student's t-test. *$P = 0.04020$, ***$P = 0.00020$. (D) qPCR analysis of TNF-α mRNA level in MEFs from *Rnf144b*$^{+/+}$ or *Rnf144b*$^{-/-}$ mice were transfected with different ligands for 12 h. $n = 3$ biological replicates. Statistical significance was determined by two-tailed unpaired Student's t-test. *$P = 0.03210$, *$P = 0.03140$. (E) qPCR analysis of IL-6 mRNA level in MEFs from *Rnf144b*$^{+/+}$ or *Rnf144b*$^{-/-}$ mice were transfected with different ligands for 12 h. $n = 3$ biological replicates. Statistical significance was determined by two-tailed unpaired Student's t-test. **$P = 0.0013$, *$P = 0.0311$. (F) *Rnf144b*$^{+/+}$ and *Rnf144b*$^{-/-}$ MEFs were infected with VSV, and then cell lysates were analyzed for VSV replication level by qPCR. $n = 3$ biological replicates. Statistical significance was determined by two-tailed unpaired Student's t-test. **$P = 0.00390$ (8 h), **$P = 0.00430$ (12 h). (G) *Rnf144b*$^{+/+}$ and *Rnf144b*$^{-/-}$ MEFs were infected with HSV, and then cell lysates were analyzed for HSV replication level by qPCR. $n = 3$ biological replicates. Statistical significance was determined by two-tailed unpaired Student's t-test. "ns" indicates no significant difference. Data information: Data shown are representative of at least three biological replicates, with each data point representing a biological experiment. Error bars are presented as mean ± SD. Statistical significance was determined by Student's t-test. Source data are available online for this figure.

