## [Peer Review File · EMBO Reports]

RNF144B negatively regulates antiviral immunity by targeting MDA5 for autophagic degradation

Guoxiu Li, Jing Zhang, Zhixun Zhao, Jian Wang, Jiaoyang Li, Weihong Xu, Zhanding Cui, Pu Sun, Hong Yuan, Tao Wang, Kun Li, Xingwen Bai, Xueqing Ma, Pinghua Li, Yuanfang Fu, Yimei Cao, Huifang Bao, Dong Li, Zaixin Liu, Ning Zhu, Lijie Tang, and Zengjun Lu

Corresponding author(s): Jing Zhang (zhangjing@caas.cn) , Zengjun Lu (luzengjun@caas.cn), Lijie Tang (tanglijie@neau.edu.cn)

Review Timeline:

Submission Date:	15th Jan 24
Editorial Decision:	14th Feb 24
Revision Received:	11th Jun 24
Editorial Decision:	10th Jul 24
Revision Received:	14th Aug 24
Accepted:	29th Aug 24

Editor: Achim Breiling

Transaction Report:

Dear Dr. Zhang,

Thank you for the submission of your research manuscript to EMBO reports. I have now received the reports from the three referees that were asked to evaluate your study, which can be found at the end of this email.

As you will see, the referees think that the findings are of interest. However, they have several comments, concerns, and suggestions, indicating that a major revision of the manuscript is necessary to allow publication of the study in EMBO reports. As the reports are below, and all the referee concerns need to be addressed, I will not detail them here.

Given the constructive referee comments, I would like to invite you to revise your manuscript with the understanding that all referee concerns must be addressed in the revised manuscript or in a detailed point-by-point response. Acceptance of your manuscript will depend on a positive outcome of a second round of review. It is EMBO reports policy to allow a single round of revision only and acceptance of the manuscript will therefore depend on the completeness of your responses included in the next, final version of the manuscript.

1) a .docx formatted version of the final manuscript text (including legends for main figures, EV figures and tables), but without the figures included. Figure legends should be compiled at the end of the manuscript text.

2) individual production quality figure files as .eps, .tif, .jpg (one file per figure), of main figures and EV figures. Please upload these as separate, individual files upon re-submission.

4) a complete author checklist, which you can download from our author guidelines

(<https://www.embopress.org/page/journal/14693178/authorguide>). Please insert page numbers in the checklist to indicate where the requested information can be found in the manuscript. The completed author checklist will also be part of the RPF.

5) that primary datasets produced in this study (e.g. RNA-seq, ChIP-seq, structural and array data) are deposited in an

appropriate public database. If no primary datasets have been deposited, please also state this in a dedicated section (e.g. 'No primary datasets have been generated and deposited'), see below.

The accession numbers and database should be listed in a formal "Data Availability" section (placed after Materials & Methods) that follows the model below. This is now mandatory (like the COI statement). Please note that the Data Availability Section is restricted to new primary data that are part of this study. This section is mandatory. As indicated above, if no primary datasets have been deposited, please state this in this section

Data availability

8) Regarding data quantification and statistics, please make sure that the number "n" for how many independent experiments were performed, their nature (biological versus technical replicates), the bars and error bars (e.g. SEM, SD) and the test used to calculate p-values is indicated in the respective figure legends (also for EV figures and all those in an Appendix). Please also check that all the p-values are explained in the legend, and that these fit to those shown in the figure. Please provide statistical testing where applicable. Please avoid the phrase 'independent experiment', but clearly state if these were biological or technical replicates. Please also indicate (e.g. with n.s.) if testing was performed, but the differences are not significant. In case n=2, please show the data as separate datapoints without error bars and statistics. See also: <http://www.embopress.org/page/journal/14693178/authorguide#statisticalanalysis>

9) Please add scale bars of similar style and thickness to microscopic images, using clearly visible black or white bars (depending on the background). Please place these in the lower right corner of the images themselves. Please do not write on or near the bars in the image but define the size in the respective figure legend.

10) Please also note our reference format:

12) We now use CRedit to specify the contributions of each author in the journal submission system. CRedit replaces the author contribution section. Please use the free text box to provide more detailed descriptions and do not provide your final manuscript text file with an author contributions section. See also our guide to authors: <https://www.embopress.org/page/journal/14693178/authorguide#authorshippinguidelines>

13) We would encourage you to use 'Structured Methods', our new Materials and Methods format. According to this format, the

Materials and Methods section should include a Reagents and Tools Table (listing key reagents, experimental models, software and relevant equipment and including their sources and relevant identifiers), uploaded as separate file, followed by a Methods and Protocols section in which we encourage the authors to describe their methods using a step-by-step protocol format with bullet points, to facilitate the adoption of the methodologies across labs. More information on how to adhere to this format as well as downloadable templates (.doc or .xls) for the Reagents and Tools Table can be found in our author guidelines (section 'Structured Methods'):

14) Please add up to 5 keywords to the manuscript and order the sections like this, using these names:

Title page - Abstract - Keywords - Introduction - Results - Discussion - Materials and Methods - Data availability section - Acknowledgements - Disclosure and Competing Interests Statement - References - Figure legends - Expanded View Figure legends

Finally, please note that all corresponding authors are required to supply an institutional e-mail address and an ORCID ID for their name upon submission of a revised manuscript. Please find instructions on how to link the ORCID ID to the account in our manuscript tracking system in our Author guidelines:

<http://www.embopress.org/page/journal/14693178/authorguide#authorshipguidelines>

I look forward to seeing a revised version of your manuscript when it is ready. Please let me know if you have questions or comments regarding the revision.

Yours sincerely,

Referee #1:

In this study, the authors conducted an RNA-seq analysis and identified RNF144B as a regulator of MDA5-mediated antiviral signaling. They showed that RNF144B mediates the ubiquitination of MDA5, leading to the degradation of the protein. They generated RNF144B knockout mice and demonstrated that cells from these mice were more resistant to EMCV infection. Furthermore, the knockout of RNF144B was shown to enhance the survival of mice following EMCV infection. However, it remains unclear whether the RNF144B-mediated degradation of MDA5 is directly responsible for the phenotype observed in RNF144B KO mice. I am concerned that there may be other targets of RNF144B that contribute this regulation. The authors should address this possibility. Specific comments are described below.

Comment 1:

RNF144B knockout markedly increased the viral replication. However, it is still unclear whether the RNF144B-mediated degradation of MDA5 can fully explain the marked phenotype, due to insufficient analysis of RNF144B defect cells. The authors should examine the cytokine expression profile in both WT and RNF144B-deficient cells before and after viral infection. Specifically, measuring the levels of type I IFNs and other pro-inflammatory cytokines in response to viral infection in MEFs, macrophages, dendritic cells would be important.

Comment 2:

The specificity of RNF144B's defect in relation to the MDA5 signaling pathway is unclear. The authors should test the response of RNF144B KO cells to the ligands for other signaling pathways, such as short poly I:C (RIG-I ligand), CL097 (TLR7 ligand), and cytoplasmic DNA (cGAS ligand), to determine whether the defect observed in RNF144B KO cells is specific to MDA5.

Comment 3:

Previous studies have reported the pos-translational modification of MDA5 and their regulatory roles in the MDA5 activation. However, the interactions between RNF144B and other regulators remain unclear. Although the authors noted that TRIM40 and Parkin mediate K48-linked polyubiquitination of MDA5, leading to the degradation of the protein, it is unclear why RNF144B, in addition to those negative regulators, is required to attenuate MDA5-mediated cytokine expression. The authors should compare the effects of RNF144B KO on MDA5 signaling with those observed in TRIM40 KO cells and Parkin KO cells.

Comment 4:

In Figure 11, the authors claim that RNF144B knockdown increase the TBK1 phosphorylation. However, the presented data do not appear to show a significant difference.

Comment 5:

The explanation provided for Figure 5B is confusing. Line 196 states that NH₄Cl was added; however, Figure 5B indicated Baf-A1 was used instead. Moreover, the effects of those inhibitors appear to be marginal.

Referee #2:

This manuscript by Li. et al demonstrates a role for the ubiquitin E3 ligase RNF144B in antiviral immunity. The authors find that RNF144B binds and polyubiquitinates MDA5, which leads to degradation of MDA5 via p62-dependent autophagy.

The manuscript is clear and well-written. The overall topic and findings are relevant and fit well within the scope of EMBO Reports. A strength of the paper is that the impact of loss or overexpression of RNF144B on antiviral immunity is shown both in cell lines (using shRNA and overexpression) and in RNF144B knockout mice. The functional data that is presented is clear and convincing, but there are a number of missing experiments that should be included to make the study more rigorous (see 'major points' below). Mechanistically, the authors demonstrate that MDA5 is degraded in the presence of (high levels of) RNF144B. They find that MDA5 is modified with K27/K33-linked ubiquitin chains by RNF144B, and that this leads to autophagic clearance via the receptor p62 and to a lesser extent CCDC50.

The mechanistic aspects of this study are less convincing. My main concern is that most data relies on overexpression of RNF144B. Another major caveat is that the ubiquitin smears that are shown appear below the molecular weight of MDA5, hence these cannot represent covalently attached ubiquitin chains. Perhaps these are unanchored ubiquitin chains? This requires further investigation.

Finally, it is unclear whether the loss of RNF144B in vivo (or in MEFs derived from RNF144B KO mice) leads to a general increase in tonic IFN signaling and hence an increased expression of ISGs (including MDA5) and an increased resistance to viral infection. This would be very important (and interesting) to investigate. Below I have indicated a number of points that should be addressed before publication in EMBO Reports.

- Line 95: The authors find that '20 genes are upregulated following all three viral infections'. Are these Interferon stimulated genes (ISGs)? If so, it is surprising that only 20 genes are overlapping; one would expect many more ISGs to be upregulated upon infection with diverse viruses. Or are these 20 genes non-ISGs? Please clarify how this analysis was done.
- The authors suggest that RNF144B protein expression is increased upon EMCV infection, VSV infection, or polyI:C treatment. This suggests that RNF144B is an interferon-stimulated gene (ISG). This should be tested by treating cells with recombinant type I IFN. Are RNF144B transcript levels also increased upon infection or polyI:C treatment? Or are the observed increases at protein level due to post-translational regulation? Note also that the observed increase in RNF144B protein expression upon EMCV infection of HEK293T cells as shown in panel 1C is not evident in panel 1I.
- Regarding EMCV infection experiments described in Fig. 1: the authors demonstrate that viral titer and signaling intermediates are affected by the overexpression of RNF144B. This should impact on IFN/ISG induction, but this is not demonstrated and should be included (qPCR and/or WB).
- ShRNA-mediated depletion of RNF144B increases phosphorylation of IκBα (Fig. 1I) but whether this affects IFN/ISG induction is not demonstrated. Related to this, the authors describe in the text that phosphorylation of TBK1 is increased (Fig. 1I) but this is not visible in the figure. Furthermore, loss of RNF144B is should increase IFN/ISG induction in response to polyI:C but no data is shown at the level of signaling intermediates (P-IRF3, P-TBK1, P-IκBα). This manuscript would be much strengthened if the authors would test in a more systematic manner whether the absence or presence of RNF144B impacts on a) viral titer, b) signaling, and c) IFN/ISG induction. The data that is shown is convincing, but the gaps in the manuscript give the feeling that the data is not as rigorous as it should be.
- Fig. 1G: this experiment would be much strengthened if a catalytic mutant (as described in Fig. 3) would be included.
- Fig. 1J and 2K should include at least one additional shRNA against RNF144B (especially since the authors show in panel 1H that they have two functional shRNAs against RNF144B).
- Fig. 2: MDA5 overexpression is known to strongly autoactivate the type I IFN response. How does the absence or presence of RNF144B affect autoactivation? If RNF144B indeed diminished MDA5 expression, one would expect a diminished type I IFN response. This could simply be tested by immunoblotting for an ISG, e.g. ISG15 or STAT1.
- Fig. 2B: an 'empty vector' control should be included in this experiment (empty vector versus RNF144B-myc).
- Fig. 2C: this observation would be strengthened if it were shown in an additional cell line, preferably one that expresses high levels of MDA5 (e.g. PMA-treated THP1 cells). In addition, it would be important to test this in the absence and presence of recombinant IFN (to increase MDA5 expression). This would demonstrate whether RNF144B is affecting MDA5 levels only at steady state or also in the context of IFN signaling.
- The ubiquitin smear on MDA5 in Figure 3 and 4 run below the molecular weight of MDA5 (135 kDa). This is a very important point. The data suggests that these are not covalently attached Ub chains, because as this would lead to an increased Mw (above 135 kDa). Instead, this may indicate that MDA5 co-immunoprecipitates with unanchored Ub chains, despite the denaturing conditions of the IP. Unanchored ubiquitin chains bind to RIG-I and can mediate its activation. A similar mechanism for MDA5 should be considered and further explored.
- Figure 3 and 4 are completely based on overexpression of RNF144B-myc. It would be important to show that loss of RNF144B

(through CRISPR/Cas9, siRNAs, or shRNAs) diminishes MDA5 ubiquitination.

- In many panels in Figure 3 and 4, MDA5 expression levels are not visibly affected by the presence of RNF144B. Examples are: Fig. 3E, 3I, 4D, 4E.
- Panel 3C: why does the ubiquitin smear in the CtoA mutant completely disappear? I.e. it should look like the 'vector only' control. Can this be explained by endogenous RNF144B that ubiquitinated MDA5-FLAG with HA-Ub? If so, a knockdown experiment (of RNF144B) would be very interesting to include.
- Panel 3D: HA WCL blot is missing. This is important to show the difference in expression level of the individual Ubiquitin mutants.
- Panel 3D: how do the authors explain the increased ubiquitination of MDA5 that is visible in multiple conditions (e.g. K6R, K11R, K48R, K63R). Please comment on this in the text.
- Figure 4: The authors identify K43 of MDA5 as a target site of ubiquitination by RNF144B. Sumoylation of K43 was previously shown to prevent K48-mediated ubiquitination and degradation of MDA5 in uninfected and early-infected cells (Hu et al., J. Exp. Med., 2017). Can the authors relate their observations to this previous study, either experimentally or in the discussion? Does increased expression of the sumoylation machinery revert the impact of RNF144B on MDA5 degradation? Does sumoylation of K43 also prevent p62-dependent autophagy of MDA5?
- Panel 5I: the siRNA against CCDC50 does not appear to work (no diminished CCDC50 protein expression). However, there is a clear effect on MDA5-FLAG expression. How can this be explained?
- Fig. 5K: how is ubiquitin (and its mutants) visualized here? Does it have a tag? If so, which one?
- Fig. 5K: why does MDA5 strongly localize to punctate structures/granulates at steady state (rather than being cytosolic)? Are these autophagosomes? Why are these much fewer p62+ structures in the K27R/K33R overexpressing cells compared to the K27O/K33O expressing cells?
- Fig. 6D/E: MDA5 expression is increased at steady state in RNF144B KO MEFs: the authors should test whether RIG-I expression is also affected. This would serve as a good negative control, to demonstrate that RNF144B specifically affects MDA5 expression, and that the loss of RNF144B is not leading to a global increase in tonic IFN signaling (and thus increased expression of ISGs. Note that even if the loss of RNF144B does lead to increased basal IFN signaling, this would be interesting to report.
- Panel 6H: related to the above point: how is basal IFN/ISG expression affected by the loss of RNF144B? This could simply be shown by qPCR analysis of uninfected RNF155B WT versus KO MEFs.
- Fig. 6: Is IFN signaling in response to recombinant type I IFN affected by the loss of RNF144B?
- Fig. 6: if RNF144B specifically affects MDA5 expression (without an impact on RIG-I expression or basal IFN signaling) one would expect that RNF144B-sufficient and -deficient MEFs are equally resistant to infection with a RIG-I dependent virus or a DNA virus (cGAS/STING-dependent virus). This would be very important and interesting to test.

Minor points:

- Line 93: Please clarify which 'host cells' were analysed (i.e. cell type).
- Fig. 3E: why is endogenous MDA5 expression not increased upon EMCV infection? It is an ISG and its expression level is therefore expected to increase upon viral infection.
- Line 195: wrong word used. Reinstated should be 'reverted' or 'rescued' or 'blocked'.
- Line 209: as above: the word reinstated is not correct.
- Throughout the text, several statements are made that are missing a reference to the literature. Please review the text carefully and include references where necessary. Example: line 294-296.
- Please clarify how the qPCR data in panel 6G is normalized. Are PBS controls set to 1?

----- Referee #3:

In this manuscript the authors demonstrate that the E3 ubiquitin ligase RNF144b inhibits viral infection, IRF3 phosphorylation, and the downstream interferon response. By examining components responsible for the interferon response, they found that MDA5 was subject to ubiquitination and degradation by RNF144b. They further characterized the site of MDA5 ubiquitination, the type of ubiquitin linkage, and the motifs RNF144b necessary to perform this ubiquitination. They also determined that MDA5 is degraded by the autophagic machinery. Finally, they knocked out RNF144b in mice and determined that knockout was protective against ECMV presumably by increasing the interferon / cytokine response and inhibiting viral infection. This was a very thorough study with compelling data that would benefit from some additional experiments, quantification, and clarification of the text.

Major concerns:

1. Aside from a comment in the methods, there is no discussion about the number of time experiments were repeated and no quantification of any of the western blots was provided. Figure 2A, 3C, 3F, 4E, 4G, 5A or 5B, 5G, 5H, 5I all could use quantification
2. There is a disconnect from the functional data and the MDA5 degradation. Do any of the RNF144b mutants (C48/80/219A) or MDA5 mutants (K23/K43R) have any effect on interferon signaling or viral replication?
3. Figure 3 all panels: Why is there a ubiquitin signal in the MDA5 pulldown well below the molecular weight of MDA5? Does the overexpressed protein also have lower molecular weight bands? Is the protein truncated? You would expect to only see a smear

above 100 kDa. Is it possible that there is non-specific pulldown in your sample?

Minor concerns:

1. For several panels, the figures are labeled prior to their mention in the text (e.g. figure 4B and 4C are referred to before 4A).
2. In figure 2H, including the truncation map of MDA5 is confusing. That should probably be moved to the supplement.
3. In figure 2I could you please point out specific and non-specific bands. I assume the unchanged bands at 25 and 55 are the heavy and light chain?
4. What are the levels of expression of the ubiquitin constructs? Were they relatively even? Total HA-ubiquitin from the K-R and the K-O could be added to the supplement or at least a statement that all constructs expressed at the same level.
5. In figure 3C, the triple mutant has considerably lower expression than the Wt. If this is normalized and quantified, is the inhibition of ubiquitination still consistent?
6. Figure 4C is too noisy to interpret. You also mention that the other domains aren't ubiquitinated, but figure S2G is not related to this experiment and I couldn't locate the correct blot.
7. Figure 5B you mention NH₄Cl, but the figure shows Bafilomycin A1 instead. Which was used?
8. Figure 5K is unconvincing. If the argument is that K27R/K33R is different than any of the other conditions addition quantification or experiment analysis is needed. Visually the overlap looks identical.
9. Figure 6E. A shorter time course where MDA5 degradation is observable in the control condition would be helpful to establish kinetics.

Referee #1:

In this study, the authors conducted an RNA-seq analysis and identified RNF144B as a regulator of MDA5-mediated antiviral signaling. They showed that RNF144B mediates the ubiquitination of MDA5, leading to the degradation of the protein. They generated RNF144B knockout mice and demonstrated that cells from these mice were more resistant to EMCV infection. Furthermore, the knockout of RNF144B was shown to enhance the survival of mice following EMCV infection. However, it remains unclear whether the RNF144B-mediated degradation of MDA5 is directly responsible for the phenotype observed in RNF144B KO mice. I am concerned that there may be other targets of RNF144B that contribute this regulation. The authors should address this possibility. Specific comments are described below.

Comment 1:

RNF144B knockout markedly increased the viral replication. However, it is still unclear whether the RNF144B-mediated degradation of MDA5 can fully explain the marked phenotype, due to insufficient analysis of RNF144B defect cells. The authors should examine the cytokine expression profile in both WT and RNF144B-deficient cells before and after viral infection. Specifically, measuring the levels of type I IFNs and other pro-inflammatory cytokines in response to viral infection in MEFs, macrophages, dendritic cells would be important.

Response: This is a very good suggestion and has been well taken. We assessed the mRNA expression levels of IFN- β , TNF- α , and IL-6 in response to EMCV infection in both wild-type (WT) and RNF144B-deficient cells, including mouse embryonic fibroblasts (MEFs), bone marrow-derived macrophages (BMDMs), and bone marrow-derived dendritic cells (BMDCs). Our results showed that RNF144B deficiency led to increased expression of type I IFNs and other pro-inflammatory cytokines. Please refer to Fig. 6D, Fig. EV4E, and Fig. EV4F for details.

Fig. 6D (In MEFs)

(D) Rnf144b^{+/+} and Rnf144b^{-/-} MEFs were infected with EMCV (MOI= 1). After 12 hours, cell lysates were analyzed for Ifnb1, TNF- α , IL-6 mRNA level by qPCR.

Fig. EV4E and Fig. EV4F

E

F

(E) Rnf144b^{+/+} and Rnf144b^{-/-} BMDMs were infected with EMCV (MOI= 1). After 12 hours, cell lysates were analyzed for Ifnb1, TNF-α, IL-6 mRNA level by qPCR.

(F) Rnf144b^{+/+} and Rnf144b^{-/-} BMDCs were infected with EMCV (MOI= 1). After 12 hours, cell lysates were analyzed for Ifnb1, TNF-α, IL-6 mRNA level by qPCR.

Comment 2:

The specificity of RNF144B's defect in relation to the MDA5 signaling pathway is unclear. The authors should test the response of RNF144B KO cells to the ligands for other signaling pathways, such as short poly I:C (RIG-I ligand), CL097 (TLR7 ligand), and cytoplasmic DNA (cGAS ligand), to determine whether the defect observed in RNF144B KO cells is specific to MDA5.

Response: Thank you for your suggestion. We transfected short poly (I:C) (RIG-I ligand), long poly I:C (MDA5 ligand), LPS (TLR4 ligand), Imiquimod (TLR7 ligand), and poly(dA:dT) (cGAS ligand) into both wild-type (WT) and RNF144B knockout (KO) MEFs. We observed a significant increase in TBK1 phosphorylation upon transfection with long poly(I:C), a slight increase in TBK1 phosphorylation following short poly (I:C) transfection, and no observable difference in TBK1 phosphorylation between WT and KO cells after poly(dA:dT) transfection (Fig. EV5A). Additionally, we detected no obvious change of IRF7 protein level upon imiquimod treatment in Rnf144b KO cells. Moreover, we found that RNF144B KO diminished the replication of EMCV and VSV. While the replication of HSV was slightly decreased in RNF144B KO cells, this reduction was not statistically significant (Fig. EV5B-Fig. EV5C). These results indicated that RNF144B specifically regulated RNA virus induced IFN signaling. Moreover, our findings revealed that RNF144B did not promote the degradation of RIG-I (Fig. EV2B), suggesting that it might regulate the activity of RIG-I without affecting its protein level. However, we cannot rule out the possibility that there are other targets of RNF144B, and we are going to investigate this possibility in our future work.

Fig.EV5A

(A) Western blot analysis of the indicated signaling proteins in MEFs from Rnf144b^{+/+} or Rnf144b^{-/-} mice were transfected with toll-like receptors ligands for 12h.

Fig.EV5B-Fig.EV5C

(B) Rnf144b^{+/+} and Rnf144b^{-/-} MEFs were infected with VSV, and then cell lysates were analyzed for VSV replication level by qPCR.

(C) Rnf144b^{+/+} and Rnf144b^{-/-} MEFs were infected with HSV, and then cell lysates were analyzed for HSV replication level by qPCR.

Comment 3:

Previous studies have reported the pos-translational modification of MDA5 and their regulatory roles in the MDA5 activation. However, the interactions between RNF144B and other regulators remain unclear. Although the authors noted that TRIM40 and Parkin mediate K48-linked polyubiquitination of MDA5, leading to the degradation of the protein, it is unclear why RNF144B, in addition to those negative regulators, is required to attenuate MDA5-mediated cytokine expression. The authors should compare the effects of RNF144B KO on MDA5 signaling with those observed in TRIM40 KO cells and Parkin KO cells.

Response: Thank you for your suggestion. Our results showed that p-TBK1 levels induced by EMCV, poly(I:C), and IFN were all upregulated in RNF144B KO cells compared to wild-type cells (Fig. EV4D, Fig. EV5D, Fig. EV4K). Additionally, we transfected siRNAs targeting Parkin and TRIM40 into HEK293T cells. The results showed that p-TBK1 levels induced by EMCV were also upregulated after siRNA knockdown of TRIM40 and Parkin (Fig. EV5E-Fig. EV5F). Furthermore, MDA5 expression was upregulated in RNF144B KO (Fig. 6C). These findings indicate that all three E3 ubiquitin ligases promote MDA5 degradation. However, the mechanisms by which these E3s regulate MDA5 differ. TRIM40 catalyzes K27 and K48 polyubiquitination of MDA5, while Parkin catalyzes K48 polyubiquitination, with both promoting degradation through the proteasome pathway. In contrast, RNF144B catalyzes K27 and K33 polyubiquitination of MDA5, promoting degradation via autophagy.

MDA5 is a critical molecule in the innate immune system, where its overactivation can lead to autoimmune diseases, and under activation can impair the host antiviral response. Therefore, the regulation of MDA5 is tightly controlled. It is not surprising that multiple enzymes are involved in its degradation, providing a redundancy in regulatory mechanisms. Ubiquitination is a key regulatory mechanism for both its activation and degradation, with different ubiquitination enzymes evolved to manage its regulation. Similarly, RIG-I, another member of the same family, can also be degraded by multiple E3 ubiquitin ligases through targeting RIG-I CARDs to mediate RIG-I degradation (Arimoto *et al*, 2007; Chen *et al*, 2013; Wang *et al*, 2016)

Fig. EV5E-Fig. EV5F

Figure for referee with unpublished data and its description has been removed upon request by the authors.

Comment 4:

In Figure 11, the authors claim that RNF144B knockdown increase the TBK1 phosphorylation. However, the presented data do not appear to show a significant difference.

Response: Thanks for pointing out that. Because p-TBK1 upregulation is significant in RNF144B KO cells (Fig. 6C and Fig. EV4D), we suspected that the lack of noticeable differences in knockdown cells might be due to insufficient knockdown efficiency. To address this issue, we re-screened the cells with puromycin, isolated monoclonal lines, and repeated the experiment. In two repeated experiments, we observed that p-TBK1 was upregulated in RNF144B knockdown cells after EMCV infection (Fig. 11). Additionally, we noted different kinetics of p-TBK1 upon EMCV infection between KO and KD cells. In KO cells, p-TBK1 levels remained high 12 hours post-infection, whereas in KD cells, p-TBK1 levels significantly decreased at the same time point.

Fig. 11

(I) Control or RNF144B KD HEK293T cells were infected with EMCV (MOI= 1) for indicated time points. Cell lysates were used for immunoblot analysis with the indicated antibodies.

Comment 5:

The explanation provided for Figure 5B is confusing. Line 196 states that NH₄Cl was added; however, Figure 5B indicated Baf-A1 was used instead. Moreover, the effects of those inhibitors appear to be marginal.

Response: Thanks for pointing out this mistake. Baf-A1 was used as indicated in Figure 5B. The text has been revised. We conducted quantification of the Western blots with image J and found that CQ and Baf-A1 completely restored MDA5 levels by inhibiting RNF144B-mediated degradation, while 3-MA partially restored MDA5 levels. Please refer to Fig. 5B for details.

Referee #2:

This manuscript by Li. et al demonstrates a role for the ubiquitin E3 ligase RNF144B in antiviral immunity. The authors find that RNF144B binds and polyubiquitinates MDA5, which leads to degradation of MDA5 via p62-dependent autophagy.

The manuscript is clear and well-written. The overall topic and findings are relevant and fit well within the scope of EMBO Reports. A strength of the paper is that the impact of loss or overexpression of RNF144B on antiviral immunity is shown both in cell lines (using shRNA and overexpression) and in RNF144B knockout mice. The functional data that is presented is clear and convincing, but there are a number of missing experiments that should be included to make the study more rigorous (see 'major points' below). Mechanistically, the authors demonstrate that MDA5 is degraded in

the presence of (high levels of) RNF144B. They find that MDA5 is modified with K27/K33-linked ubiquitin chains by RNF144B, and that this leads to autophagic clearance via the receptor p62 and to a lesser extent CCDC50.

The mechanistic aspects of this study are less convincing. My main concern is that most data rely on overexpression of RNF144B. Another major caveat is that the ubiquitin smears that are shown appear below the molecular weight of MDA5, hence these cannot represent covalently attached ubiquitin chains. Perhaps these are unanchored ubiquitin chains? This requires further investigation.

Finally, it is unclear whether the loss of RNF144B in vivo (or in MEFs derived from RNF144B KO mice) leads to a general increase in tonic IFN signaling and hence an increased expression of ISGs (including MDA5) and an increased resistance to viral infection. This would be very important (and interesting) to investigate. Below I have indicated a number of points that should be addressed before publication in EMBO Reports.

- Line 95: The authors find that '20 genes are upregulated following all three viral infections'. Are these Interferon stimulated genes (ISGs)? If so, it is surprising that only 20 genes are overlapping; one would expect many more ISGs to be upregulated upon infection with diverse viruses. Or are these 20 genes non-ISGs? Please clarify how this analysis was done.

Response: Thank you for your question. Among the 20 identified genes that showed upregulation upon infection with PRRSV, FMDV, and SVA, nine of them are classified as ISGs. These include PMAIP1, STX11, MX2, SOCS1, CD274, IDO1, TNFAIP3, PARP14, and BATF2. Please refer to the supplementary material for the complete list of differentially expressed genes. Several factors may contribute to the relatively small number of overlapping genes. Firstly, these viruses infect different cell types: PRRSV targets porcine alveolar macrophages, while FMDV and SVA infect PK-15 cells (isolated from porcine kidney). Additionally, PRRSV is known for its immunosuppressive nature, which may lead to a less pronounced type I IFN response compared to FMDV and SVA infections.

- The authors suggest that RNF144B protein expression is increased upon EMCV infection, VSV infection, or poly(I:C) treatment. This suggests that RNF144B is an interferon-stimulated gene (ISG). This should be tested by treating cells with recombinant type I IFN. Are RNF144B transcript levels also increased upon infection or poly(I:C) treatment? Or are the observed increases at protein level due to post-translational regulation? Note also that the observed increase in RNF144B protein expression upon EMCV infection of HEK293T cells as shown in panel 1C is not evident in panel 1I.

Response: Thank you for your question. Our result suggests that RNF144B protein expression is increased upon EMCV infection, VSV infection, or poly(I:C) transfection. Additionally, we treated 293T cells with IFN- α and assessed the mRNA and protein levels of RNF144B. The findings revealed that the protein level of RNF144B increased in a time-dependent manner upon IFN α treatment, whereas the mRNA expression level remained unaffected (Fig. 1L and 1M). Moreover, RNF144B transcript levels also increased upon infection or poly(I:C) treatment (Fig 1A and Fig EV11). These results suggested that RNF144B is not a typical ISG, the protein level increases may be attributed to post-translational regulation. Finally, we considered that in panel 1I, the lack of an increase in RNF144B protein expression level after EMCV infection might be due to the method used to establish the cell lines. Specifically, 293T cells were infected with shscramble lentivirus and selected with puromycin to establish a cell pool within a week, rather than isolating a single cell clone. It is possible that the lentivirus infection increased the basal level of RNF144B expression. To address this, we re-screened the shscramble and shRNF144B lentivirus-infected cells with puromycin, isolated monoclonal lines, and repeated the experiment. We observed that RNF144B was indeed upregulated after EMCV infection (Fig. 1I). Please refer to updated Fig. 1I for details.

Fig. 1L and 1M and Fig. EV1I

(L) HEK293T cells were treated with IFN- α 2b for indicated time points and subjected to immunoblot analysis of RNF144B expression.

(M) HEK293T cells were treated with IFN- α 2b for indicated time points and subjected to qPCR analysis of RNF144B mRNA level.

(I) HEK293T cells were transfected with poly(I:C) for indicated time points and subjected to qPCR analysis of RNF144B mRNA level.

Fig. 1I

(I) Control or RNF144B KD HEK293T cells were infected with EMCV (MOI= 1) for indicated time points. Cell lysates were used for immunoblot analysis with the indicated antibodies.

- Regarding EMCV infection experiments described in Fig. 1: the authors demonstrate that viral titer and signaling intermediates are affected by the overexpression of RNF144B. This should impact on IFN/ISG induction, but this is not demonstrated and should be included (qPCR and/or WB).

Response: Thanks for your suggestion. We detected IFN/ISG mRNA level by qPCR in HEK293T cells overexpressing RNF144B after EMCV infection. We observed that overexpression of RNF144B led to reduced mRNA levels of IFNB1 and ISG15 upon EMCV infection. Please find the results in Fig. 1H.

Fig. 1H

(H) HEK293T cells were transfected with empty vector or RNF144B-Myc plasmid. After 24 h, cells were infected with EMCV for the indicated time points and subjected to qPCR analysis for IFNB1 and ISG15 mRNA level.

- ShRNA-mediated depletion of RNF144B increases phosphorylation of IκBα (Fig. 1I) but whether this affects IFN/ISG induction is not demonstrated. Related to this, the authors describe in the text that phosphorylation of TBK1 is increased (Fig. 1I) but this is not visible in the figure. Furthermore, loss of RNF144B is should increase IFN/ISG induction in response to poly I:C but no data is shown at the level of signaling intermediates (P-IRF3, P-TBK1, P-IκBα). This manuscript would be much strengthened if the authors would test in a more systematic manner whether the absence or presence of RNF144B impacts on a) viral titer, b) signaling, and c) IFN/ISG induction. The data that is shown is convincing, but the gaps in the manuscript give the feeling that the data is not as rigorous as it should be.

Response: We appreciated the question. We added additional data on viral titers and IFN/ISG induction in RNF144B knockdown cells (Fig.1J and Fig.1K). The results showed that RNF144B knockdown enhanced IFN production and impaired virus replication. Please find the effects of overexpression of RNF144B on a) viral titer, b) signaling, and c) IFN/ISG induction in Fig.1F-1H, and the knockdown of RNF144B in Fig.1I-1K. In addition, poly(I:C) induced IFN signaling (P-TBK1, P-IκBα) was increased in RNF144B KO cells (Fig. EV5D). As shown in Fig.6C, upregulation of p-TBK1 is significant in RNF144B KO cells, we suspected that the lack of noticeable differences in knockdown cells might be due to insufficient knockdown efficiency (Fig.1I). To address this issue, we re-screened the cells with puromycin, isolated monoclonal lines, and repeated the experiment. In two repeated experiments, we observed that p-TBK1 was upregulated in RNF144B knockdown cells after EMCV infection. Please find the updated results in Fig.1I.

Fig.1J and Fig.1K

K

(J) Control or RNF144B KD HEK293T cells were infected with EMCV (MOI= 1) for 12 h and subjected to qPCR analysis for IFNB1 and ISG15 mRNA level.

(K) Control or RNF144B KD HEK293T cells were infected with EMCV (MOI= 1) for indicated time points, The cell supernatant was harvested and analyzed by TCID₅₀.

Fig.EV5D

(D) Western blot analysis of the indicated signaling proteins in MEFs from Rnf144b^{+/+} or Rnf144b^{-/-} mice were transfected with poly(I:C) for the indicated time periods.

- Fig. 1G: this experiment would be much strengthened if a catalytic mutant (as described in Fig. 3) would be included.

Response: Thank you for the suggestion. Following your advice, we evaluated the effect of the catalytic mutant on IFN signaling and EMCV replication levels. The results showed no alteration in EMCV replication or in the phosphorylation levels of TBK1, IRF3, and IkB α . Please find in Fig.3D and Fig.3E.

Fig.3D and Fig.3E

(D) HEK293T cells were transfected with empty vector or C48/80/219A-Myc plasmid. After 24 h, cells were infected with EMCV (MOI= 1) for indicated time points and subjected to qPCR analysis for EMCV mRNA level.

(E) HEK293T cells were transfected with empty vector or C48/80/219A-Myc plasmid. After 24 h, cells were infected

with EMCV (MOI= 1) for the indicated time points. Cells were harvested and protein extracts were analyzed by immunoblot using the indicated antibodies.

- Fig. 1J and 2K should include at least one additional shRNA against RNF144B (especially since the authors show in panel 1H that they have two functional shRNAs against RNF144B).

Response: Thank you for your suggestion. As suggested, we employed a different shRNA-3 to silence RNF144B and examined the expression of IFNB1 and ISG15 following poly(I:C) transfection. The findings demonstrated increased expression of both IFNB1 and ISG15 upon RNF144B knockdown, consistent with the results obtained with shRNA2. Please refer to Fig. EV1M-EV1N for details.

Fig. EV1M and EV1N

(M-N) Control or RNF144B KD HEK293T cells were transfected with poly(I:C) for indicated time points and subjected to qPCR analysis for IFNB1 and ISG15 mRNA level.

- Fig. 2: MDA5 overexpression is known to strongly autoactivate the type I IFN response. How does the absence or presence of RNF144B affect autoactivation? If RNF144B indeed diminished MDA5 expression, one would expect a diminished type I IFN response. This could simply be tested by immunoblotting for an ISG, e.g. ISG15 or STAT1.

Response: Thank you for your question. We performed the experiments by immunoblotting and qPCR. Our results demonstrated that the overexpression of RNF144B led to a decrease in STAT1 phosphorylation and protein levels, along with reduced mRNA expression of STAT1 and ISG15. Please find the results in Fig.2E and Fig. EV2D. And knockout of RNF144B increased basal level of STAT1 protein level (Fig. EV4H).

Fig.2E and Fig. EV2D

(E) HEK293T cells were transfected with plasmids encoding RNF144B and MDA5-Flag. 24 h post-transfection, cell

lysates were used for immunoblot analysis with the indicated antibodies.

(D) HEK293T cells were transfected with plasmids encoding RNF144B-myc and MDA5-Flag. 24 h post-transfection, cells were subjected to qPCR analysis for ISG15 and STAT1 mRNA level.

Fig.EV4H

(H) Rnf144b^{+/+} and Rnf144b^{-/-} MEFs cell lysates were analyzed for STAT1 protein level by western blot.

- Fig. 2B: an 'empty vector' control should be included in this experiment (empty vector versus RNF144B-myc).

Response: An empty vector has been included in the experiment. Please find the updated results in Fig.2B.

Fig.2B

(B) HEK293T cells were transfected with plasmids encoding RNF144B (0, 0.5, 1, 2 μg) and MDA5-Flag (2 μg). 24 h post-transfection, cell lysates were used for immunoblot analysis with the indicated antibodies.

- Fig. 2C: this observation would be strengthened if it were shown in an additional cell line, preferably one that expresses high levels of MDA5 (e.g. PMA-treated THP1 cells). In addition, it would be important to test this in the absence and presence of recombinant IFN (to increase MDA5 expression). This would demonstrate whether RNF144B is affecting MDA5 levels only at steady state or also in the context of IFN signaling.

Response: We examined MDA5 degradation by RNF144B in A549 cells. The findings revealed that RNF144B facilitated the degradation of endogenous MDA5 in A549 cells. Please refer to Fig. EV2A for details. Furthermore, our observations showed that RNF144B overexpression reduced MDA5 protein levels both in the absence and presence of recombinant IFN (Fig. 2D). Taken together, these results indicated that RNF144B promoted the degradation of MDA5 both at steady state and in the context of IFN signaling.

Fig. EV2A

A

(A) A549 cells were transfected with plasmid encoding RNF144B. 24 h post-transfection, protein extracts were used for immunoblot analysis of the endogenous MDA5 protein level.

Fig. 2D

D

(D) A549 cells were transfected with empty vector or RNF144B-Myc plasmid. 12 h post-transfection, cells were treated with interferon for 12 h, protein extracts were used for immunoblot analysis of the endogenous MDA5 protein level.

- The ubiquitin smear on MDA5 in Figure 3 and 4 run below the molecular weight of MDA5 (135 kDa). This is a very important point. The data suggests that these are not covalently attached Ub chains, because as this would lead to an increased Mw (above 135 kDa). Instead, this may indicate that MDA5 co-immunoprecipitates with unanchored Ub chains, despite the denaturing conditions of the IP. Unanchored ubiquitin chains bind to RIG-I and can mediate its activation. A similar mechanism for MDA5 should be considered and further explored.

Response: Thank you for raising this important point. We transfected USP5, a DUB that specifically removes unanchored ubiquitin chains, resulted in the disappearance of the smear (Please refer to Fig. EV2J for details), indicating that RNF144B also facilitates the synthesis of unanchored ubiquitin chains. It is a very interesting discovery. It has been reported that unanchored K63 polyubiquitin chains activate MDA5 (Song *et al*, 2021). We believe that the role of unanchored ubiquitin chains binding to MDA5 requires further exploration. We find it particularly interesting to study the mechanism of unanchored ubiquitin chains in regulating MDA5 signaling. Specifically, further studies are needed to determine the length of unanchored ubiquitin chains, identify the binding sites for these chains on MDA5, understand their physiological function, and assess the differences in the induction of downstream genes. These studies will enhance our understanding of the mechanisms that fine-tune MDA5-mediated type I IFN signaling. However, due to limited time and budget, we cannot complete these experiments in the short term. We are seeking experts in biochemistry and structural biology to collaborate on this project.

Fig. EV2J

Figure for referee with unpublished data and its description has been removed upon request by the authors.

- Figure 3 and 4 are completely based on overexpression of RNF144B-myc. It would be important to show that loss of RNF144B (through CRISPR/Cas9, siRNAs, or shRNAs) diminishes MDA5 ubiquitination.

Response: This is an important suggestion and has been well taken. We observed reduced ubiquitination of MDA5 induced by EMCV infection in RNF144B KO MEFs. Please refer to Fig. EV2I for details.

Fig. EV2I

(I) Rnf144b^{+/+} and Rnf144b^{-/-} MEFs infected with EMCV (MOI= 1) for the indicated time points. cell lysates were harvested and immunoprecipitated using anti-MDA5 antibody, followed by immunoblots using the indicated antibodies.

- In many panels in Figure 3 and 4, MDA5 expression levels are not visibly affected by the presence of RNF144B. Examples are: Fig. 3E, 3I, 4D, 4E.

Response: Thank you for pointing out that. We apologize for not mentioning this in the figure legend initially. In these experiments, CQ was added to prevent the degradation of MDA5. We have now added a description in the figure legend and indicated it in the figures.

- Panel 3C: why does the ubiquitin smear in the CtoA mutant completely disappear? I.e. it should look like the 'vector only' control. Can this be explained by endogenous RNF144B that ubiquitinated MDA5-FLAG with HA-Ub? If so, a knockdown experiment (of RNF144B) would be very interesting to include.

Response: Thank you for your question. We quantified the WB results and found that the loading amount of the catalytic mutant was too small. We repeated the experiment and adjusted the loading amount. The results showed that the ubiquitination level in the mutant transfection group was similar to that of the vector transfection group. Please find the updated results in Fig.3C. Furthermore, we observed reduced ubiquitination of MDA5 induced by EMCV infection in RNF144B KO MEFs. Please refer to Fig. EV2I for details.

Fig.3C

(C) HEK293T cells co-transfected with HA-Ub, MDA5-Flag, along with empty vector, wild-type RNF144B-myc or RNF144B-myc mutants. Cell lysates were subjected to denature-immunoprecipitation with anti-Flag antibody followed by immunoblot analysis with indicated antibodies.

Fig. EV2I

(I) Rnf144b^{+/+} and Rnf144b^{-/-} MEFs infected with EMCV (MOI= 1) for the indicated time points. cell lysates were harvested and immunoprecipitated using anti-MDA5 antibody, followed by immunoblots using the indicated antibodies.

- Panel 3D: HA WCL blot is missing. This is important to show the difference in expression level of the individual Ubiquitin mutants.

Response: Thank you for pointing out that. The figure has been updated to include the HA WCL blot, demonstrating similar expression levels for the ubiquitin mutants. Please find the revised results in Fig.3F and Fig.3G.

- Panel 3D: how do the authors explain the increased ubiquitination of MDA5 that is visible in multiple conditions (e.g. K6R, K11R, K48R, K63R). Please comment on this in the text.

Response: Thank you for this question. We conducted this experiment many times to determine which ubiquitin chains were attached to MDA5. During these experiments, we found that the levels of WT ubiquitin were not always stable. Despite our efforts to normalize the expression levels of ubiquitin mutants, the results were not ideal. We believe that the increased ubiquitination of MDA5 in the K6R, K11R, K48R, and K63R transfected groups was due to the relatively low levels of WT ubiquitin. Therefore, we need to interpret the results in combination with mutants containing only one ubiquitination site (Fig. 3G).

- Figure 4: The authors identify K43 of MDA5 as a target site of ubiquitination by RNF144B. Sumoylation of K43 was previously shown to prevent K48-mediated ubiquitination and degradation of MDA5 in uninfected and early-infected cells (Hu et al., J. Exp. Med., 2017). Can the authors relate their observations to this previous study, either experimentally or in the discussion? Does increased expression of the sumoylation machinery revert the impact of RNF144B on MDA5 degradation? Does sumoylation of K43 also prevent p62-dependent autophagy of MDA5?

Response: We co-transfected TRIM38 and RNF144B and found that overexpression of TRIM38 attenuated the degradation of MDA5 induced by RNF144B (Fig. EV2L). We thought that it is very likely that sumoylation of K43 prevent p62-dependent autophagy of MDA5. Our results together with previous study suggest that RNF144B-mediated ubiquitination and TRIM38-mediated sumoylation at the K43 site of MDA5 may exclusively compete with each other to regulate the stability of MDA5. K23 and K43 can be ISGylated and promote the activation of MDA5 (Liu et al, 2021). These results indicated that K23 and K43 are very important sites for post-translational modification of MDA5. We have discussed in the discussion. Please find in line 311-319.

Fig. EV2L

(L) HEK293T cells were co-transfected with empty vector or RNF144B-Myc, TRIM38-Flag and MDA5-HA. 24 h post-transfection, cell lysates were analyzed by immunoblot using the indicated antibodies.

- Panel 5I: the siRNA against CCDC50 does not appear to work (no diminished CCDC50 protein expression). However, there is a clear effect on MDA5-FLAG expression. How can this be explained?

Response: Thank you for your question. We synthesized additional siRNAs and verified their interference efficiency. We utilized one siRNA with higher interference efficiency. The results showed that when the knockdown efficiency exceeded 50%, the knockdown of CCDC50 partially restored RNF144B's degradation of MDA5, consistent with our previous findings. Please find the updated Fig EV3A and Fig.5I.

Fig EV3A

(A) Immunoblot analysis of CCDC50 expression in HEK293T cells expressing siRNA against CCDC50

Fig.5I

(I) HEK293T cells were transfected siCCDC50, along with MDA5-Flag and empty vector or RNF144B. Cell lysates were used for immunoblot analysis with the indicated antibodies.

- Fig. 5K: how is ubiquitin (and its mutants) visualized here? Does it have a tag? If so, which one?

Response: Ubiquitin and its mutants are HA-tagged. We apologize for the lack of clarity and have now clarified this in the figure legend.

- Fig. 5K: why does MDA5 strongly localize to punctate structures/granulates at steady state (rather than being cytosolic)? Are these autophagosomes? Why are these much fewer p62+ structures in the K27R/K33R overexpressing cells compared to the K27O/K33O expressing cells?

Response: Thank you for your question. We thought that the relatively low expression of MDA5 in the resting state resulted in low fluorescence intensity in the cytoplasm. This phenomenon is consistent with observations in published articles, where overall MDA5 fluorescence intensity in the cytoplasm increased following viral infection. It is possible that these are autophagosomes, since it co-localized with LC3B according to previous study (Hou *et al*, 2021). We observed low fluorescence levels in the K27R, K33R, and K27R/K33R groups for all antibodies. Consequently, we adjusted the amounts of plasmids used in the transfection and repeated the experiments. The results were consistent with previous findings (Fig 5K): some colocalization of MDA5, P62, and ubiquitin was still observed in the K27R and K33R single mutant groups, while colocalization among the three was barely detectable in the K27R/K33R group.

Fig 5K

(K) Confocal microscopy analysis of A549 cells transfected with MDA5-Flag (green), p62-BFP (blue), K27O, K33O, K27R, K33R and K27R/K33R mutants (red).

- Fig. 6D/E: MDA5 expression is increased at steady state in RNF144B KO MEFs: the authors should test whether RIG-I expression is also affected. This would serve as a good negative control, to demonstrate that RNF144B specifically affects MDA5 expression, and that the loss of RNF144B is not leading to a global increase in tonic IFN signaling (and thus increased expression of ISGs. Note that even if the loss of RNF144B does lead to increased basal IFN signaling, this would be interesting to report.

Response: This is an important suggestion and has been well taken. We observed that RNF144B did not promote the degradation of RIG-I (Fig. EV4G). Although loss of RNF144B led to a slight increase in IFNB1 expression, it did not result in a widespread elevation of tonic IFN signaling (Fig. EV4I and EV4J).

Fig. EV4G

G

G Rnf144b^{+/+}MEFs and Rnf144b^{-/-}MEFs were infected with EMCV (MOI= 1) . 12 h post-infection, cells were treated by CHX (30 μ M) for indicated time points. Protein extracts were used for immunoblot analysis of the endogenous RIG-I protein level.

Fig.EV4I and EV4J

(I, J) Rnf144b^{+/+} and Rnf144b^{-/-} MEFs cell lysates were analyzed for Ifnb1 and Isg15 mRNA level by qPCR.

- Panel 6H: related to the above point: how is basal IFN/ISG expression affected by the loss of RNF144B? This could simply be shown by qPCR analysis of uninfected RNF144B WT versus KO MEFs.

Response: We examined the mRNA expression levels of basal IFN and ISG expression in both WT and RNF144B KO cells (Fig.EV4I and EV4J). The findings showed a slight increase in basal IFN expression in RNF144B KO cells, along with up-regulated basal Isg15 expression. These results suggest that RNF144B KO does not induce a global increase in IFN production at steady state.

Fig.EV4I and EV4J

(I, J) Rnf144b^{+/+} and Rnf144b^{-/-} MEFs cell lysates were analyzed for Irfn1 and Isg15 mRNA level by qPCR.

- Fig. 6: Is IFN signaling in response to recombinant type I IFN affected by the loss of RNF144B?

Response: Thank you for your question. We performed the experiments and observed that the phosphorylation level of TBK1 increased in response to recombinant type I IFN in RNF144B KO cells. Please refer to the results in Fig. EV4K for further details.

Fig. EV4K

Figure for referee with unpublished data and its description has been removed upon request by the authors.

- Fig. 6: if RNF144B specifically affects MDA5 expression (without an impact on RIG-I expression or basal IFN signaling) one would expect that RNF144B-sufficient and -deficient MEFs are equally resistant to infection with a RIG-I dependent virus or a DNA virus (cGAS/STING-dependent virus). This would be very important and interesting to test.

Response: Thank you for this important question. We evaluated the replication levels of VSV and HSV in both RNF144B WT and KO cells. We observed that RNF144B KO hindered VSV replication while having no significant impact on HSV replication (Fig.EV5B and Fig.EV5C). This result is consistent with our finding that poly(I:C) induced p-TBK1 was increased in RNF144B KO cells, but there is no difference of other ligands induced p-TBK1 (Fig.EV5A). We considered two possible explanations for this result: first, RNF144B might influence RIG-I activity without affecting its protein level, as we observed that it did not degrade RIG-I (Fig. EV2B and Fig. EV4G). Second, there may be other targets of RNF144B involved. We are going to investigate these possibilities in our future work.

Fig.EV5B and Fig.EV5C

(B) Rnf144b^{+/+} and Rnf144b^{-/-} MEFs were infected with VSV, and then cell lysates were analyzed for VSV replication level by qPCR.

(C) Rnf144b^{+/+} and Rnf144b^{-/-} MEFs were infected with HSV, and then cell lysates were analyzed for HSV replication level by qPCR.

Minor points:

- Line 93: Please clarify which 'host cells' were analysed (i.e. cell type).

Response: PRRSV infected primary porcine alveolar macrophages, while FMDV and SVA infected PK-15 cells. We have clarified the host cells in the text.

- Fig. 3E: why is endogenous MDA5 expression not increased upon EMCV infection? It is an ISG and its expression level is therefore expected to increase upon viral infection.

Response: Thank you for your question. We reviewed the literature to optimize the dose and timing for EMCV infection. We doubled the amount of EMCV inoculated. The results showed that the upregulation of MDA5 protein levels by EMCV infection was more pronounced at an MOI of 2. Consistent with previous results, RNF144B interacted with MDA5 both before and after EMCV infection. Please refer to the results in Fig. 2G for further details.

- Line 195: wrong word used. Reinstated should be 'reverted' or 'rescued' or 'blocked'.

Response: We are sorry about the mistakes. We have corrected the word in the text.

- Line 209: as above: the word reinstated is not correct.

Response: We are sorry about the mistakes. We have corrected the word in the text.

- Throughout the text, several statements are made that are missing a reference to the literature. Please review the text carefully and include references where necessary. Example: line 294-296.

Response: Thanks for the point out. We have reviewed the text and included references accordingly.

- Please clarify how the qPCR data in panel 6G is normalized. Are PBS controls set to 1?

Response: Thanks for the question. The qPCR showed relative expression of the cytokines. One of the PBS treated wild-type mouse was set to 1.

Referee #3:

In this manuscript the authors demonstrate that the E3 ubiquitin ligase RNF144b inhibits viral infection, IRF3 phosphorylation, and the downstream interferon response. By examining components responsible for the interferon response, they found that MDA5 was subject to ubiquitination and degradation by RNF144b. They further characterized the site of MDA5 ubiquitination, the type of ubiquitin linkage, and the motifs RNF144b necessary to perform this ubiquitination. They also determined that MDA5 is degraded by the autophagic machinery. Finally, they knocked out RNF144b in mice and determined that knockout was protective against ECMV presumably by increasing

the interferon / cytokine response and inhibiting viral infection. This was a very thorough study with compelling data that would benefit from some additional experiments, quantification, and clarification of the text.

Major concerns:

1. Aside from a comment in the methods, there is no discussion about the number of time experiments were repeated and no quantification of any of the western blots was provided. Figure 2A, 3C, 3F, 4E, 4G, 5A or 5B, 5G, 5H, 5I all could use quantification

Response: Thank you for your suggestion. Most of the experiments were repeated at least three times. We have specified in figure legends. We have quantified the western blots with image J, and have been marked in the figures.

2. There is a disconnect from the functional data and the MDA5 degradation. Do any of the RNF144b mutants (C48/80/219A) or MDA5 mutants (K23/K43R) have any effect on interferon signaling or viral replication?

Response: This question is of particular importance. We observed that transfection of RNF144B C48/80/219A mutants into HEK293T cells did not impair the replication of EMCV or interferon signaling. Please find the results in (Fig.3D and Fig.3E). We did not test the effect of MDA5 K23/K43R mutants on IFN signaling or viral replication because we failed to establish an MDA5 KO cell line in a short time. It has been reported that K43R inhibited EMCV-induced transcription of *Irfn1* (Hu et al, 2017). K43 can also undergo sumoylation and K48-linked ubiquitination for its activation or degradation. K23 and K43 can be ISGylated and promote the activation of MDA5. So the effects of their mutations on interferon pathways and viral replication will be complicated. We will try to figure out in our future study.

Fig.3D and Fig.3E

(D) HEK293T cells were transfected with empty vector or C48/80/219A-Myc plasmid. After 24 h, cells were infected with EMCV (MOI= 1) for indicated time points and subjected to qPCR analysis for EMCV mRNA level.

(E) HEK293T cells were transfected with empty vector or C48/80/219A-Myc plasmid. After 24 h, cells were infected with EMCV (MOI= 1) for the indicated time points. Cells were harvested and protein extracts were analyzed by immunoblot using the indicated antibodies.

3. Figure 3 all panels: Why is there a ubiquitin signal in the MDA5 pulldown well below the molecular weight of MDA5? Does the overexpressed protein also have lower molecular weight bands? Is the protein truncated? You would expect to only see a smear above 100 kDa. Is it possible that there is non-specific pulldown in your sample?

Response: Thank you for the important question. We thought that the smear below 135 kDa might be unanchored ubiquitin chains. To test this, we transfected USP5, a DUB that specifically removes unanchored ubiquitin chains, which resulted in the disappearance of the smear (see Fig. EV2J for details). Our results suggest that RNF144B also

facilitates the synthesis of unanchored ubiquitin chains. Their physiological functions need to be further explored.

Fig. EV2J

Figure for referee with unpublished data and its description has been removed upon request by the authors.

Minor concerns:

1. For several panels, the figures are labeled prior to their mention in text (e.g. figure 4B and 4C are referred to before 4A).

Response: Thanks for the point out. We have rearranged the figures. Figure 4A was moved to Figure 4E in the current version.

2. In figure 2H, including the truncation map of MDA5 is confusing. That should probably be moved to the supplement.

Response: Thanks for the suggestion. We have moved the truncation map of MDA5 to Fig.EV2K.

3. In figure 2I could you please point out specific and non-specific bands. I assume the unchanged bands at 25 and 55 are the heavy and light chain?

Response: Thanks for the point out. We have labeled specific and non-specific bands. The bands at 25 and 55 are heavy and light chains.

4. What are the levels of expression of the ubiquitin constructs? Were they relatively even? Total HA-ubiquitin from the K-R and the K-O could be added to the supplement or a least a statement that all constructs expressed at the same level.

Response: Thank you for your question. We have added the blot of HA-ubiquitin in Whole cell lysis. The expression

levels of ubiquitin constructs are relatively even. Please find the updated results in Fig.3F and Fig.3G.

5. In figure 3C, the triple mutant has considerably lower expression than the Wt. If this is normalized and quantified, is the inhibition of ubiquitination still consistent?

Response: Thank you for the question. We quantified the WB results and found that the loading amount of the catalytic mutant was too small. After repeating the experiment with adjusted loading amounts, we observed that the ubiquitination level in the mutant transfection group was similar to that of the vector transfection group. Please find the updated results in Fig. 3C.

Fig.3C

(C) HEK293T cells co-transfected with HA-Ub, MDA5-Flag, along with empty vector, wild-type RNF144B-myc, or RNF144B-myc mutants. Cell lysates were subjected to denature-immunoprecipitation with anti-Flag antibody followed by immunoblot analysis with indicated antibodies.

6. Figure 4C is too noisy to interpret. You also mention that the other domains aren't ubiquitinated, but figure S2G is not related to this experiment and I couldn't locate the correct blot.

Response: Sorry for the confusion. As you mentioned that the CARDS domain is too noisy, we performed experiments using plasmids with different tags. We transfected HEK293T cells with plasmids expressing His-tagged ubiquitin, Myc-tagged RNF144B, and HA-tagged full-length (WT) MDA5 or truncated MDA5 plasmids containing the CARDS, ATP, or CTD domains (Fig. EV2K). We conducted an *in vivo* ubiquitination assay on the CARD, ATP, and CTD domains of MDA5, respectively. The results demonstrated that the ubiquitination of WT MDA5 or the CARDS domain of MDA5 was strongly enhanced by overexpression of RNF144B (Fig. 4A and Fig. 4B). However, the ubiquitination of the ATP and CTD domains showed no significant effect from the overexpression of RNF144B (Fig. 4C and Fig. 4D). Please find the results in Fig.4A- and Fig.4D.

Fig.4A and Fig.4D

A-D HEK293T cells co-transfected with His-Ub, MDA5-HA (wild-type and mutants), along with empty vector or RNF144B-myc. Cell lysates were subjected to denature-immunoprecipitation with anti-Flag antibody followed by immunoblot analysis with indicated antibodies.

7. Figure 5B you mention NH4Cl, but the figure shows Bafilomycin A1 instead. Which was used?

Response: We are sorry for the mistakes. It was Bafilomycin A1 used as indicated in the figure, and we have corrected the text.

8. Figure 5K is unconvincing. If the argument is that K27R/K33R is different than any of the other conditions addition quantification or experiment analysis is needed. Visually the overlap looks identical.

Response: Thank you for your question. We noticed that the expression levels of p62, ubiquitin, and MDA5 were relatively lower in the K27R, K33R, and K27R/K33R groups compared to other groups. Therefore, we repeated the experiments and adjusted the expression levels of these proteins. The results were consistent with previous findings: some colocalization of MDA5, p62, and ubiquitin was still observed in the K27R and K33R single mutant groups, while colocalization among the three was barely detectable in the K27R/K33R group.

Fig 5K

(K) Confocal microscopy analysis of A549 cells transfected with MDA5-Flag (green), p62-BFP (blue), K27O, K33O, K27R, K33R and K27R/K33R mutants (red).

9. Figure 6E. A shorter time course where MDA5 degradation is observable in the control condition would be helpful to establish kinetics.

Response: Thanks for the point out. We have performed a shorter time course experiment. Please refer to the updated Western blot results in the Fig.6E.

Fig.6E

(E) Rnf144b^{+/+}MEFs and Rnf144b^{-/-}MEFs were infected with EMCV (MOI= 1) .12 h post-infection, cells were treated by CHX (30 μM) for indicated time points. Protein extracts were used for immunoblot analysis of the endogenous MDA5 protein level.

Reference

- Arimoto K, Takahashi H, Hishiki T, Konishi H, Fujita T, Shimotohno K (2007) Negative regulation of the RIG-I signaling by the ubiquitin ligase RNF125. *Proceedings of the National Academy of Sciences of the United States of America* 104: 7500-7505
- Chen W, Han C, Xie B, Hu X, Yu Q, Shi L, Wang Q, Li D, Wang J, Zheng P *et al* (2013) Induction of Siglec-G by RNA viruses inhibits the innate immune response by promoting RIG-I degradation. *Cell* 152: 467-478
- Hou P, Yang K, Jia P, Liu L, Lin Y, Li Z, Li J, Chen S, Guo S, Pan J *et al* (2021) A novel selective autophagy receptor, CCDC50, delivers K63 polyubiquitination-activated RIG-I/MDA5 for degradation during viral infection. *Cell research* 31: 62-79
- Hu MM, Liao CY, Yang Q, Xie XQ, Shu HB (2017) Innate immunity to RNA virus is regulated by temporal and reversible sumoylation of RIG-I and MDA5. *The Journal of experimental medicine* 214: 973-989
- Liu G, Lee JH, Parker ZM, Acharya D, Chiang JJ, van Gent M, Riedl W, Davis-Gardner ME, Wies E, Chiang C *et al* (2021) ISG15-dependent activation of the sensor MDA5 is antagonized by the SARS-CoV-2 papain-like protease to evade host innate immunity. *Nature microbiology* 6: 467-478
- Song B, Chen Y, Liu X, Yuan F, Tan EYJ, Lei Y, Song N, Han Y, Pascal BD, Griffin PR *et al* (2021) Ordered assembly of the cytosolic RNA-sensing MDA5-MAVS signaling complex via binding to unanchored K63-linked poly-ubiquitin chains. *Immunity* 54: 2218-2230.e2215
- Wang W, Jiang M, Liu S, Zhang S, Liu W, Ma Y, Zhang L, Zhang J, Cao X (2016) RNF122 suppresses antiviral type I interferon production by targeting RIG-I CARDS to mediate RIG-I degradation. *Proceedings of the National Academy of Sciences of the United States of America* 113: 9581-9586

Dear Dr. Zhang,

Thank you for the submission of your revised manuscript to our editorial offices. I have already forwarded the reports I have received from the referees that I asked to re-evaluate the study to you, you will also find again below. As you know, referee #3 is satisfied with the revision. However, both referees #1 and #2 have remaining concerns and suggestions to improve the study, or ask for changes and clarifications. Going through your provisional p-b-p-response (further revision plan), I think these points will be addressed accordingly. I thus invite you to submit a final revised version of the manuscript, addressing all remaining points as indicated by the referees and in your revision plan. Please also provide a final p-b-p-response regarding the remaining points of the referees.

- Please provide the abstract written in present tense throughout.
 - The first and last name for authors Zhu Ning are displayed in different order in manuscript file and in the submission system. Please provide the correct name both in the manuscript and in the submission system. Moreover, please provide an institutional e-mail address for co-corresponding author Lijie Tang (in the manuscript file and in the submission system).
 - Please note that all corresponding and co-corresponding authors are required to supply an ORCID ID for their name upon submission of a revised manuscript. We still require this for author Lijie Tang. Please find instructions on how to link the ORCID ID to the account in our manuscript tracking system in our Author guidelines:
<http://www.embopress.org/page/journal/14693178/authorguide#authorshipguidelines>
 - We now use CRediT to specify the contributions of each author in the journal submission system. CRediT replaces the author contribution section. Please use the free text box to provide more detailed descriptions and do NOT provide your final manuscript text file with an author contributions section. See also our guide to authors:
<https://www.embopress.org/page/journal/14693178/authorguide#authorshipguidelines>
 - Please order the manuscript sections like this, using these names:
Abstract - Keywords - Introduction - Results - Discussion - Methods - Data availability section - Acknowledgements - Disclosure and Competing Interests Statement - References - Figure legends - Expanded View Figure legends
 - Please make sure that the number "n" for how many independent experiments were performed, their nature (biological versus technical replicates), the bars and error bars (e.g. SEM, SD) and the test used to calculate p-values is indicated in the respective figure legends (also for potential EV figures and all those in the final Appendix). Please also check that all the p-values are explained in the legend, and that these fit to those shown in the figure. Please provide statistical testing where applicable. Please avoid the phrase 'independent experiment', but clearly state if these were biological or technical replicates. Please also indicate (e.g. with n.s.) if testing was performed, but the differences are not significant. In case n=2, please show the data as separate datapoints without error bars and statistics. See also:
<http://www.embopress.org/page/journal/14693178/authorguide#statisticalanalysis>
- If n<5, please show single datapoints for diagrams. Presently, some diagrams seem to miss the 'n.s.'. Please check. Moreover:
- Please note that the exact p values are not provided in the legends of figures 1f, h, j, k, m; 6a-b, d, f-h; EV 1b-d, g-i, k-n; EV 2d; EV 4a-c, e-f, i-j; EV 5b.
 - Please indicate the statistical test used for data analysis in the legend of figure 6i.
 - Please note that in figures 1f, h, j, k; 3d; 6a-b, d, f-h; EV 1b-d, f-i, k-n; EV 2d; EV 3f; EV 4b, e-f, i-j; EV 5b-c; there is a mismatch between the annotated p values in the figure legend and the annotated p values in the figure file that should be corrected.
 - Please note that for the figure 1a, p-values and statistical tests are indicated in the legends. However, these p-values ("****/**/**/*") are not shown in the figure. Please rectify this in the figure or legend as applicable.
 - Please add to each legend (main, and EV figures, where applicable) a 'Data Information' section explaining the statistics used or providing information regarding replicates and scales. See:

- Please make sure that all the funding information is also entered into the online submission system and that it is complete and similar to the one in the acknowledgement section of the manuscript text file. Presently, funds National Natural Science Foundation of China (32270763), National Key R&D Program of China (2021YFD1800300) and the Science and Technology Major Project of Gansu Province (22ZD6NA001) are missing from the submission system. Please check.
- Please make sure that all figure panels and tables are called out separately and sequentially. Presently, there seems to be no callout for Appendix Table S3. Please check.

- Appendix tables S4 is a dataset. Please remove this from the Appendix and upload the original excel file as dataset (with a legend on the first TAB). Please name this file 'Dataset EV1' and change all callouts respectively.

- Please remove the default text from the Reagents table.

- Thanks you for providing the source data (SD) for this study. But please provide also a filled in source data checklist with your final submission (attached again) and make sure all SD requested is provided (it seems SD for Fig. 4E is missing). Moreover, please upload the SD files as one ZIPed folder per figure for each main figure. All SD for EV figures can be ZIP together and uploaded in one folder.

In addition, I would need from you:

Yours sincerely,

Referee #1:

The authors have addressed my previous concerns. However, I still have several issues regarding the data presented in this revision. Specifically, it is still unclear whether the role of RNF144B in the innate immune response is specific to MDA5. Fig. EV5A suggests that RNF144B knockdown slightly increases TBK1 phosphorylation in response to LMW poly I:C (RIG-I ligand) and reduces the phosphorylation in response to imiquimod (TLR7 ligand). It is unclear whether the differences are significant and if they affect the cytokine expression. The authors should measure the cytokine expression (such as type I IFNs and other pro-inflammatory cytokines) in RNF144B KO cells in response to those ligands and clarify whether RNF144B is specific to MDA5 or it moderately regulates other factors.

Additionally, Fig EV5B showed that RNF144B KO attenuated VSV replication. Given that VSV is recognized by RIG-I, this data suggests that RNF144B regulates not only MDA5 but also RIG-I. This appears to be inconsistent with the data showing that RNF144B does not reduce the RIG-I protein levels (Figs 2A and EV2B). Based on these observations, it is possible that RNF144B attenuates both MDA5 and RIG-I signaling, potentially targeting other factors downstream of RIG-I. The authors should address this inconsistency in their discussion. Regarding this point, a previous study has shown that RNF144B inhibits TLR4 signaling by targeting TBK1. Considering that TBK1 is involved in both RIG-I and MDA5 signaling, the inactivation of TBK1 by RNF144B could explain why VSV replication was attenuated by RNF144B KO. However, TBK1 is also crucial for HSV-1-induced innate immune response, indicating that RNF144B might target not only TBK1 but also other factors. The authors should appropriately discuss this issue.

Referee #2:

Overall, the manuscript by Li et al. has improved by the inclusion of additional data. The authors have clearly made an effort to address all reviewers' comments. However, not all points that were previously brought up were convincingly addressed. A few important issues therefore remain to be addressed to make this paper ready for publication.

Major issues:

- Fig. 3D: a comparison should be made between WT and the triple CtoA mutant, not between EV and the CtoA mutant. And why was viral titer not determined in this experiment, like in Fig. 1?
- The authors indicate that the ubiquitin smears below 135 kDa (i.e. below the molecular weight of MDA5) may be due to unanchored ubiquitin chains. Covalent and non-covalently bound ubiquitin chains can be distinguished by using a particular lysis

buffer, i.e. 1% SDS eliminates unanchored chains and leaves only covalently bound ubiquitin chains. From the materials and methods, I understand that the authors have used a 1% SDS-based lysis buffer for their ubiquitination experiments, hence the smearing below 135 kDa cannot be due to unanchored chains. It therefore remains very puzzling why there is so much smearing below 135 kDa. It may be useful to do the reverse IP (IP of HA-ubiquitin, immunoblot for FLAG-MDA5). Or, even better, to do an IP of endogenous MDA5 (as done in S2I for mouse RNF144b KO cells). Perhaps the smearing below 135 kDa is an overexpression artefact?

- Line 286: "it did not induce a global increase in tonic IFN signaling". Based on the ISG15 plot, I would say that basal ISGs are modestly upregulated in the KO cells and that the conclusion of the authors is incorrect. On the other hand, the fact that the KO cells are equally susceptible to HSV infection suggests that differences in tonic signaling are minor. I suggest that the authors:
 - o 1) analyse a few additional ISGs by qPCR analysis (as done for ISG15) to obtain a more global picture of alterations in ISG expression in the KO cells. Irrespective of the outcome of this analysis, the results will be worthwhile to incorporate and clarify to the reader what happens with basal ISG expression in these cells.
 - o 2) rephrase the text to make a less strong statement on tonic IFN signaling.
- What do the authors demonstrate in Fig. S4K? There is no referral to this figure in the text. The authors mention this figure in their rebuttal letter, and indicate that this experiment was done to test whether recombinant type I IFN signaling is affected by the loss of RNF144B. However, this is not a proper readout to test this, as TBK1 is upstream of the IFNAR. Instead, the authors should have tested Phospho-STAT1 or ISG transcript levels after recombinant IFN stimulation. Now, going back to the experiment displayed in S4K, this seems a very important experiment with a clear result. Is the increased phosphorylation of TBK1 (in absence of any viral ligand!) upon IFN stimulation in RNF144b KO cells a result of MDA5 stabilization and therefore increased autoactivation in these cells? Or is this a direct effect of RNF144B on TBK1 as reported by Zhang et al. 2019? The first point could be addressed by testing whether the increase in P-TBK1 disappears upon knockdown/knockout of MDA5 in RNF144b KO cells.

Minor issues/textual issues:

- Fig. 2J: please explain what 'mut2' refers to.
- Fig. 5K: please label panels as in 5J (indicating whether proteins are tagged with FLAG or HA, to clarify that 5K shows overexpressed proteins, rather than endogenous proteins).
- Line 220: MG132 is a proteasome inhibitor (not a protease inhibitor).
- Line 210: 'this mutant' should be 'the K23/43R double mutant'.
- Line 216: please include a concluding sentence.
- Fig. S5: eliminate panel E and F (unclear what the relevance of these figures is). It is also unclear what the panel on the right in S5A demonstrates. There is no referral in the text to this panel.

Referee #3:

The authors have addressed all the major concerns. I have no further concerns and recommend the manuscript for publication.

Referee #1

The authors have addressed my previous concerns. However, I still have several issues regarding the data presented in this revision.

Specifically, it is still unclear whether the role of RNF144B in the innate immune response is specific to MDA5. Fig. EV5A suggests that RNF144B knockdown slightly increases TBK1 phosphorylation in response to LMW poly I:C (RIG-I ligand) and reduces the phosphorylation in response to imiquimod (TLR7 ligand). It is unclear whether the differences are significant and if they affect the cytokine expression. The authors should measure the cytokine expression (such as type I IFNs and other pro-inflammatory cytokines) in RNF144B KO cells in response to those ligands and clarify whether RNF144B is specific to MDA5 or it moderately regulates other factors.

Response: Thank you for the suggestion. We measured IFNs and pro-inflammatory cytokines expression in RNF144B KO cells in response to those ligands. Our results showed that RNF144B deficiency led to increased expression of type I IFN and other pro-inflammatory cytokines upon transfection with long poly(I:C) and slight increase of short poly(I:C), and no observable respond to poly (dA:dT). Please refer to Fig. EV6C-6E for details. We thought that the regulatory role of RNF144B on MDA5 played an essential role in regulating RNA viruses induced IFN signaling. These results indicate that, in addition to MDA5, RNF144B may also moderately regulate RIG-I or its downstream genes. We are currently investigating this possibility in our lab, and discussed in the Discussion section.

Fig. EV6C-6E

(C-E) qPCR analysis of *Ifnb1*, *TNF- α* , *IL-6* mRNA level in MEFs from *Rnf144b*^{+/+} or *Rnf144b*^{-/-} mice were transfected with different ligands for 12 h.

Additionally, Fig EV5B showed that RNF144B KO attenuated VSV replication. Given that VSV is recognized by RIG-I, this data suggests that RNF144B regulates not only MDA5 but also RIG-I. This appears to be inconsistent with the data showing that RNF144B does not reduce the RIG-I protein levels (Figs 2A and EV2B). Based on these observations, it is possible that RNF144B attenuates both MDA5 and RIG-I signaling, potentially targeting other factors downstream of RIG-I. The authors should address this inconsistency in their discussion.

Response: We acknowledge that our current data are insufficient to explain how RNF144B regulates the RIG-I pathway. Although its promotion of VSV replication is significant, repeated experiments have confirmed that RNF144B does not degrade RIG-I. One possibility to explain this inconsistency is that RNF144B may regulate the activity of RIG-I rather than its protein level. Additionally, we suspect the possibility that RNF144B regulates other molecules downstream of RIG-I. We will explore this in future work. We have discussed this inconsistency in the discussion section. Please find in line 378-384.

Regarding this point, a previous study has shown that RNF144B inhibits TLR4 signaling by targeting TBK1. Considering that TBK1 is involved in both RIG-I and MDA5 signaling, the inactivation of TBK1 by RNF144B could explain why VSV replication was attenuated by RNF144B KO. However, TBK1 is also crucial for HSV-1-induced innate immune response, indicating that RNF144B might target not only TBK1 but also other factors. The authors should appropriately discuss this issue.

Response: Thank you for the suggestion. In the previous study, it was shown that RNF144B's negative regulation of TBK1 is independent of its E3 ligase activity. However, our study found that RNF144B's promotion of EMCV replication is dependent on its E3 ligase activity and its regulation of MDA5. These results suggest that RNF144B's regulation of TBK1 may be context-dependent, varying according to different ligands. In addition, we found that poly (dA:dT), which

is a DNA ligand, induced IFN production is not increased in RNF144B KO cells. This result indicating that negative regulation of TBK1 by RNF144B played minor role in regulating ligands induced IFN activation. In summary, we believe that the negative regulatory role of RNF144B on TBK1 does not significantly contribute to the reduced VSV replication observed in RNF144B KO cells. We have discussed in Discussion (line 385-398).

Referee #2

Overall, the manuscript by Li et al. has improved by the inclusion of additional data. The authors have clearly made an effort to address all reviewers' comments. However, not all points that were previously brought up were convincingly addressed. A few important issues therefore remain to be addressed to make this paper ready for publication.

Major issues:

- Fig. 3D: a comparison should be made between WT and the triple CtoA mutant, not between EV and the CtoA mutant. And why was viral titer not determined in this experiment, like in Fig. 1?

Response: Thank you for this question. We transfected cells with either an empty vector or RNF144B-C48/80/219A-Myc plasmid for 18 hours, followed by infection with EMCV (MOI = 1) for the indicated time points. Viral titers in the cell supernatant were determined during the infection using the TCID₅₀ method. A comparison between the wild-type and the triple C-to-A mutant showed that the mutant significantly diminished its ability to enhance virus replication. Please refer to the updated Fig. 3D for details.

Fig. 3D

(D) HEK293T cells were transfected with empty vector, RNF144B-myc or C48/80/219A-Myc plasmid. After 18 h, cells were infected with EMCV (MOI= 1). n = 3 biological replicates. The cells supernatant was harvested and analyzed by TCID₅₀ (50 % Tissue Culture Infective Dose) assay. Statistical significance was determined by two-tailed unpaired Student' s t-test. "ns" indicates no significant difference, **P = 0.0034 (RNF144B), **P = 0.00210 (C48/80/219A).

- The authors indicate that the ubiquitin smears below 135 kDa (i.e. below the molecular weight of MDA5) may be due to unanchored ubiquitin chains. Covalent and non-covalently bound ubiquitin chains can be distinguished by using a particular lysis buffer, i.e. 1% SDS eliminates unanchored chains and leaves only covalently bound ubiquitin chains. From the materials and methods, I understand that the authors have used a 1% SDS-based lysis buffer for their ubiquitination experiments, hence the smearing below 135 kDa cannot be due to unanchored chains. It therefore remains very puzzling why there is so much smearing below 135 kDa. It may be useful to do the reverse IP (IP of HA-ubiquitin, immunoblot for FLAG-MDA5). Or, even better, to do an IP of endogenous MDA5 (as done in S2I for mouse RNF144b KO cells). Perhaps the smearing below 135 kDa is an overexpression artefact?

Response: We greatly appreciate this question and valuable suggestion. We performed an IP of endogenous MDA5 to detect the ubiquitination of MDA5 during virus infection. The immunoprecipitation result demonstrated that RNF144B

facilitated the ubiquitination of endogenous MDA5. The majority of the smearing below 135 kDa disappeared, suggesting that it was an overexpression artifact. However, a small portion of smearing below 135 kDa still remains, which might be due to partially degraded proteins. Please refer to Fig. EV2J for details.

(J) A549 cells transfected with RNF144B-myc. After 18 h, cells were infected with EMCV (MOI= 1) for indicated time points. Cell lysates were subjected to immunoprecipitation with anti-MDA5 antibody followed by immunoblot analysis with indicated antibodies.

- Line 286: "it did not induce a global increase in tonic IFN signaling". Based on the ISG15 plot, I would say that basal ISGs are modestly upregulated in the KO cells and that the conclusion of the authors is incorrect. On the other hand, the fact that the KO cells are equally susceptible to HSV infection suggests that differences in tonic signaling are minor. I suggest that the authors:

o 1) analyse a few additional ISGs by qPCR analysis (as done for ISG15) to obtain a more global picture of alterations in ISG expression in the KO cells. Irrespective of the outcome of this analysis, the results will be worthwhile to incorporate and clarify to the reader what happens with basal ISG expression in these cells.

Response: Thank you for the suggestion. We examined the mRNA expression levels of additional ISGs (IFIT1, IFIT2, IFIT3 and MX1) in both WT and RNF144B KO cells. The findings showed modestly increase in basal IFN and ISGs expression in RNF144B KO cells. Please refer to Fig. EV5A for details.

Fig. EV5A

(A) Rnf144b^{+/+} and Rnf144b^{-/-} MEFs cell lysates were analyzed for ISGs mRNA level by qPCR.

2) rephrase the text to make a less strong statement on tonic IFN signaling.

Response: Thank you for your suggestion. We have rephrased the text about statement on tonic IFN signaling. Please find the revision at line 287-295. And we have discussed in discussion (line 395-398).

- What do the authors demonstrate in Fig. S4K? There is no referral to this figure in the text. The authors mention this figure in their rebuttal letter, and indicate that this experiment was done to test whether recombinant type I IFN signaling is affected by the loss of RNF144B. However, this is not a proper readout to test this, as TBK1 is upstream of the IFNAR. Instead, the authors should have tested Phospho-STAT1 or ISG transcript levels after recombinant IFN stimulation. Now, going back to the experiment displayed in S4K, this seems a very important experiment with a clear result. Is the increased phosphorylation of TBK1 (in absence of any viral ligand!) upon IFN stimulation in RNF144b KO cells a result of MDA5 stabilization and therefore increased autoactivation in these cells? Or is this a direct effect of RNF144B on TBK1 as reported by Zhang et al. 2019? The first point could be addressed by testing whether the increase in P-TBK1 disappears upon knockdown/knockout of MDA5 in RNF144b KO cells.

Response: Thank you for your suggestion. We apologize for not mentioning Fig. S4K in the text. We assessed Phospho-STAT1 levels by Western blot and ISG transcript levels by qPCR. Upon recombinant IFN stimulation, Phospho-STAT1 levels increased in *Rnf144b*^{-/-} MEFs compared to wild-type MEFs (see Fig. EV5C). Additionally, mRNA expression levels of *Ifit1*, *Ifit2*, and *ISG15* were elevated in MEFs from *Rnf144b* KO mice compared to their WT counterparts (see Fig. EV5D).

Furthermore, we evaluated p-TBK1 levels following MDA5 knockdown in RNF144b KO cells. We transfected siRNAs targeting MDA5 into *Rnf144b*^{-/-} MEFs. The RNF144b KO-induced increase in p-TBK1 levels upon EMCV infection was reversed by MDA5 knockdown (see Fig. EV5E). These findings suggest that the increased phosphorylation of TBK1 upon IFN stimulation in RNF144b KO cells depends on MDA5. Additionally, as previously reported, RNF144B's regulatory role on TBK1 is not dependent on its E3 ligase activity. However, our results show that RNF144B's regulation of MDA5 and IFN production is dependent on its E3 ligase activity. Taken together, the enhancement of IFN signaling upon RNF144b deficiency is dependent on its regulation of MDA5 rather than a direct effect on TBK1.

Fig. EV5C-5D

(C) Western blot analysis of the indicated signaling proteins in MEFs from Rnf144b^{+/+} or Rnf144b^{-/-} mice were treated with IFN for the indicated time periods.

(D) qPCR analysis of ISGs mRNA level in MEFs from Rnf144b^{+/+} or Rnf144b^{-/-} mice were treated with IFN for the indicated time periods.

Fig. EV5E

(E) siRNAs targeting MDA5 or NC were transfected into Rnf144b^{+/+} or Rnf144b^{-/-} MEFs cells and infected with EMCV (MOI= 1) for 12 h. Cell lysates were used for immunoblot analysis with the indicated antibodies.

Minor issues/textual issues:

- Fig. 2J: please explain what 'mut2' refers to.

Response: Mut2 refers to an RNF144B mutant in which five cysteine residues (C206, C211, C214, C219, and C222) in the RING2 domain were mutated to serine. We have included this information in the Fig. EV2E.

- Fig. 5K: please label panels as in 5J (indicating whether proteins are tagged with FLAG or HA, to clarify that 5K shows overexpressed proteins, rather than endogenous proteins).

Response: Thanks for your suggestion. We have labeled panels in 5K as in 5J.

- Line 220: MG132 is a proteasome inhibitor (not a protease inhibitor).

Response: It has been corrected.

- Line 210: 'this mutant' should be 'the K23/43R double mutant'.

Response: Fixed.

- Line 216: please include a concluding sentence.

Response: Thank you for the suggestion. A concluding sentence has been included. Please find in line 213-215.

- Fig. S5: eliminate panel E and F (unclear what the relevance of these figures is). It is also unclear what the panel on the right in S5A demonstrates. There is no referral in the text to this panel.

Response: Thank you for the suggestion. Panel S5E and S5F has been removed. Figure S5A (new Figure EV6A) was mentioned in line 301.

Referee #3

The authors have addressed all the major concerns. I have no further concerns and recommend the manuscript for publication.

Dr. Jing Zhang
LVRI
Yanchangbu, Xujiaping #1
Lanzhou, Gansu 730000
China

Dear Dr. Zhang,

I am very pleased to accept your manuscript for publication in the next available issue of EMBO reports. Thank you for your contribution to our journal.

Yours sincerely,

Referee #1:

The authors have addressed all of my previous concerns.

Referee #2:

The authors have thoroughly addressed all my previous issues. I have no further questions. The publication is suitable for publication in EMBO Reports.
